# Bernoulli's principle-mediated $Cl_2$ electrosynthesis

Zhihao Nie[1], Guoliang Xu[2], Jingjing Duan [2], Markus Antonietti [3] & Sheng Chen [1,3] ✉

Existing technologies for chlorine ($Cl_2$) synthesis are generally suffered from low productivity or high production cost. Guided by Bernoulli's principle, here we report an efficient yet cost-effective electrochemical system for $Cl_2$ electrosynthesis, which is composed of anodic chlorine evolution reaction (CER) connected to gas chamber by triple-phase gas diffusion layer. The key is to modulate gas diffusion layer by Bernoulli's principle, wherein the pressure difference at triple-phase boundary drives oriented $Cl_2$ migration directly into gas chamber, thus preventing the crossover of anodic/cathodic products. By further joining with a pH-tolerant catalyst, a standalone prototype device is built for high-rate $Cl_2$ production, operating at the Faradaic efficiencies of 96.3% ~ 87.6% in the current density range of 0.1 ~ 1.14 A cm$^{-2}$, having superior $Cl_2$ synthesis performance. Further technical-economic evaluations of our synthetic scheme demonstrate reduced $Cl_2$ production cost, saving 6.75% (1.17 million dollar per year) as comparison to conventional chlor-alkali design. We expect these findings offer broader opportunities to develop industrially production processes for other chemical commodities.

Chlorine ($Cl_2$) is one of the 100 most important chemical compounds in the world with diverse applications in disinfection goods production, pharmaceutical manufacture and wastewater treatment[1]. Up to 2024, the global production of $Cl_2$ was more than 90 million tons per year, its market value being over US $20 billion. However, an efficient yet cost-effective technology for $Cl_2$ production is still lacking.

The chemical oxidation method for $Cl_2$ synthesis was developed centuries ago by Berthollet (1789)[2] and Deacon (1868)[3], but is limited by low productivity. After 1900s, membrane-based chlor-alkali processes have become the dominant routes for industrial $Cl_2$ production, which are composed of anodic chlorine evolution reaction (CER)[4], cathodic hydrogen evolution reaction (HER)[5] in addition to membranes (like asbestos fleeces[6] or fluorine-containing ion exchange membranes[7]) for separating $Cl_2$ production at the anode, hydrogen ($H_2$) production at the cathode and hydroxyl ions (OH$^-$) in the electrolyte. In spite of their improved productivities, a major problem associated with these systems includes high operation costs arising from limited lifetime and low tolerance to contaminant ions of membranes[8,9]. Further, these membranes also increase system maintenance cost because of complicated configuration, as gas pressures in anodic and cathodic chambers must remain in balance, otherwise aggravating membrane degradation[10].

By contrast, a membrane-free chlor-alkali design would allow for simplified system configuration and durable operation, thus potentially delivering production cost savings (Fig. 1a, b). Yet, realizing such a design has met substantial challenges because of anodic/cathodic product crossover[11]:

$$\text{Anodic CER}: 2Cl^- - 2e^- \rightarrow Cl_2, E^o = 1.36V \ vs.\text{RHE} \quad (1)$$

$$\text{Cathodic HER}: 2H_2O + 2e^- \rightarrow H_2 + 2OH^-, E^o = 0V \ vs.\text{RHE} \quad (2)$$

[1]Key Laboratory for Soft Chemistry and Functional Materials, School of Chemistry and Chemical Engineering, Nanjing University of Science and Technology, Nanjing, China. [2]School of Energy and Power Engineering, Nanjing University of Science and Technology, Ministry of Education, Nanjing, China. [3]Max Planck Institute of Colloids and Interfaces, Potsdam, Germany. ✉e-mail: sheng.chen@njust.edu.cn

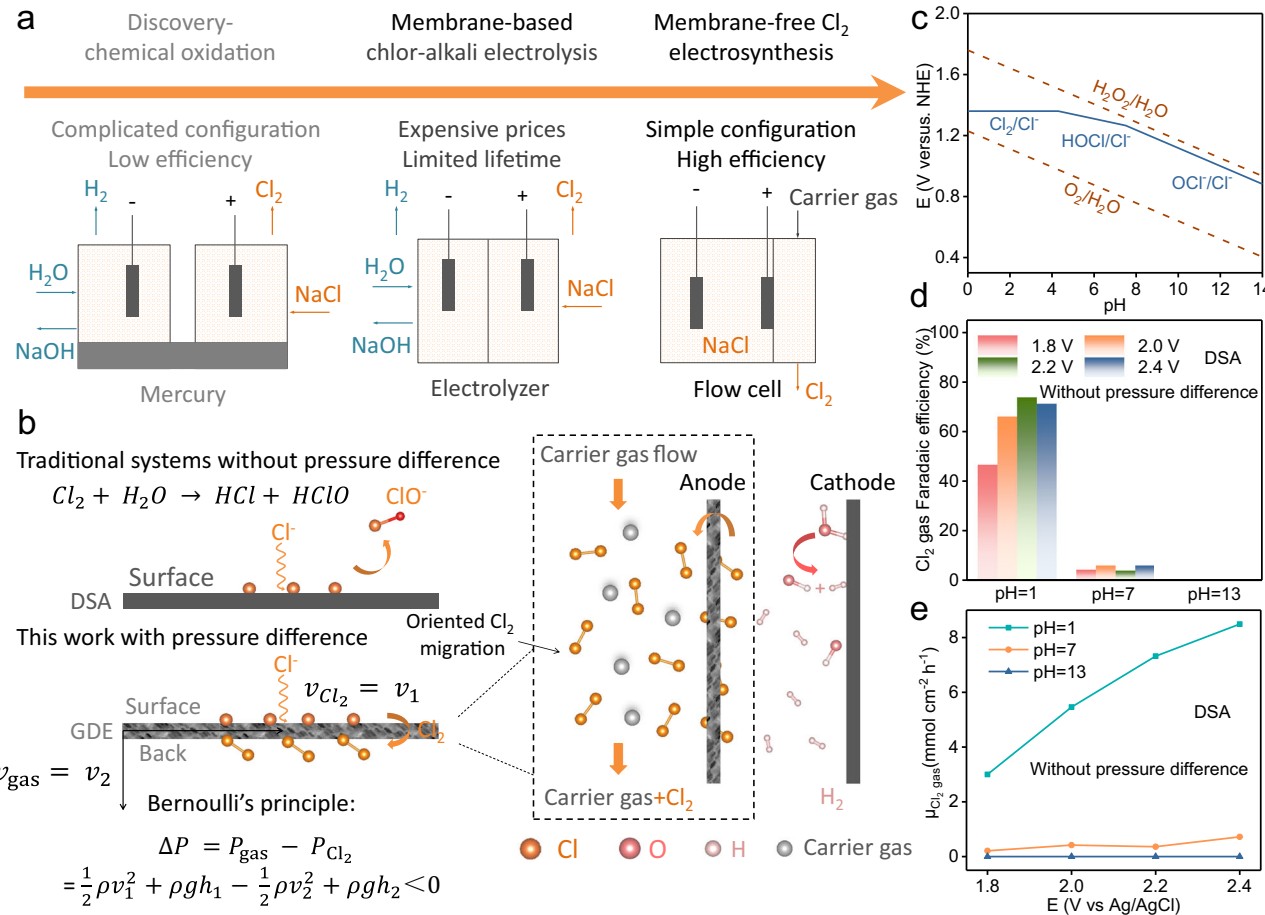

**Fig. 1 | Limitation of Pourbaix diagram for $Cl_2$ electrosynthesis. a** Overview of $Cl_2$ synthesis history. **b** The basic configuration of our electrochemical cells. **c** Pourbaix diagram of $Cl_2$ evolution reduction. **d, e** Experimental results of $Cl_2$ Faradaic efficiencies and yield rates by using a benchmark DSA electrode. Source data for (**c–e**) are provided as a Source Data file.

$$\text{Side OER} : 2H_2O \rightarrow O_2 + 4H^+ + 4e^-, E^o = 1.23V\ vs.\ \text{RHE} \quad (3)$$

Without a membrane, gaseous products of anodic $Cl_2$ and cathodic $H_2$ directly mix to form an $H_2/Cl_2$ mixture, especially at high current densities coupled with high reaction rates, which tends to explode upon ignition or light irradiation ($H_2/Cl_2$ explosion limit: 5%~87.5%)[12]. Further, hydroxyl ions ($OH^-$) from cathodic HER easily migrants to the anodic CER chamber that elevates pH (pH > 7), and according to Pourbaix diagram (Fig. 1c)[13], resulting in parasitic oxygen evolution reaction (OER) that consumes electrons otherwise for producing $Cl_2$[7]. In spite of several preliminary studies separating anodic/cathodic products by redox mediators (like Hg/NaHg[14] and $Na_{0.44}MnO_2$[6]), their $Cl_2$ productivities are prohibitively low, with the maximum electricity efficiency of 50% by considering a whole operating cycle. Further, problems associated with these systems involve production costs, complex configuration and limited lifetimes of redox mediators[15–17].

In this work, we report a strategy to address the stumbling problem of anodic/cathodic product crossover, which is achieved by taking advantage of "Bernoulli's principle", an energy conservation model firstly reported in 1738[18,19]. The local pressure of a streaming gas can be leveraged by tuning its speed, with the pressure being lower the higher the speed. Yet, the conventional Bernoulli's model only describes binary-phase systems. For gas-involving electrochemistry (like CER[7] and other redox reactions of $O_2$[20]/$CO_2$[21]/$N_2$[22]), this model

needs be revised to include triple-phase interfaces, as high-rate reactions occurs at the ternary boundaries of gas/aqueous electrolytes/catalysts. Guided by the updated model, here we have conducted numerical simulations for the local pressure of $Cl_2$ at triple-phase boundary (Please see Figs. 1b, 5c, d and Supplementary Video 1). By manipulating the flow rates of liquids (0 ~ 8.4 mL min⁻¹) and carrier gases (0 ~ 80 mL min⁻¹), a substantial pressure difference can be achieved (0 ~ 116.3 × 10⁻³ Pa). This offers the opportunity to promote oriented migration of $Cl_2$ away from reaction interfaces (i.e., $H_2/OH^-$ at cathode/anode), which can not only prevent $H_2/Cl_2$ crossover but also circumvent the Pourbaix diagram problems, leading to enhanced activities. Finally, our strategy has been demonstrated for solving key challenging problems in other chemical/catalytic reactions like water splitting ($2H_2O \rightarrow 2H_2 + O_2$, Supplementary Fig. 79).

## Results and discussion

### Limitation of Pourbaix diagram

To illustrate typical limitations of Pourbaix diagram in CER (Fig. 1b), the benchmark dimensionally stable anode (DSA, Supplementary Fig. 1), as developed half a century ago[23], has been integrated into a flow cell to evaluate its activities. Firstly, the LSV of DSA in 5 M NaCl (pH = 1) were measured, showing a current density of 10 mA cm⁻² at an overpotential of 80 mV (Supplementary Fig. 2), which is consistent to the literature[5]. Nevertheless, by elevating the pH from 1 to 13 with the DSA catalyst, $Cl_2$ yield rates and Faradaic efficiencies (FE) change significantly. At the potential of 1.8 V, the $Cl_2$ gas Faradaic efficiencies are 46.6%, 4.3% and

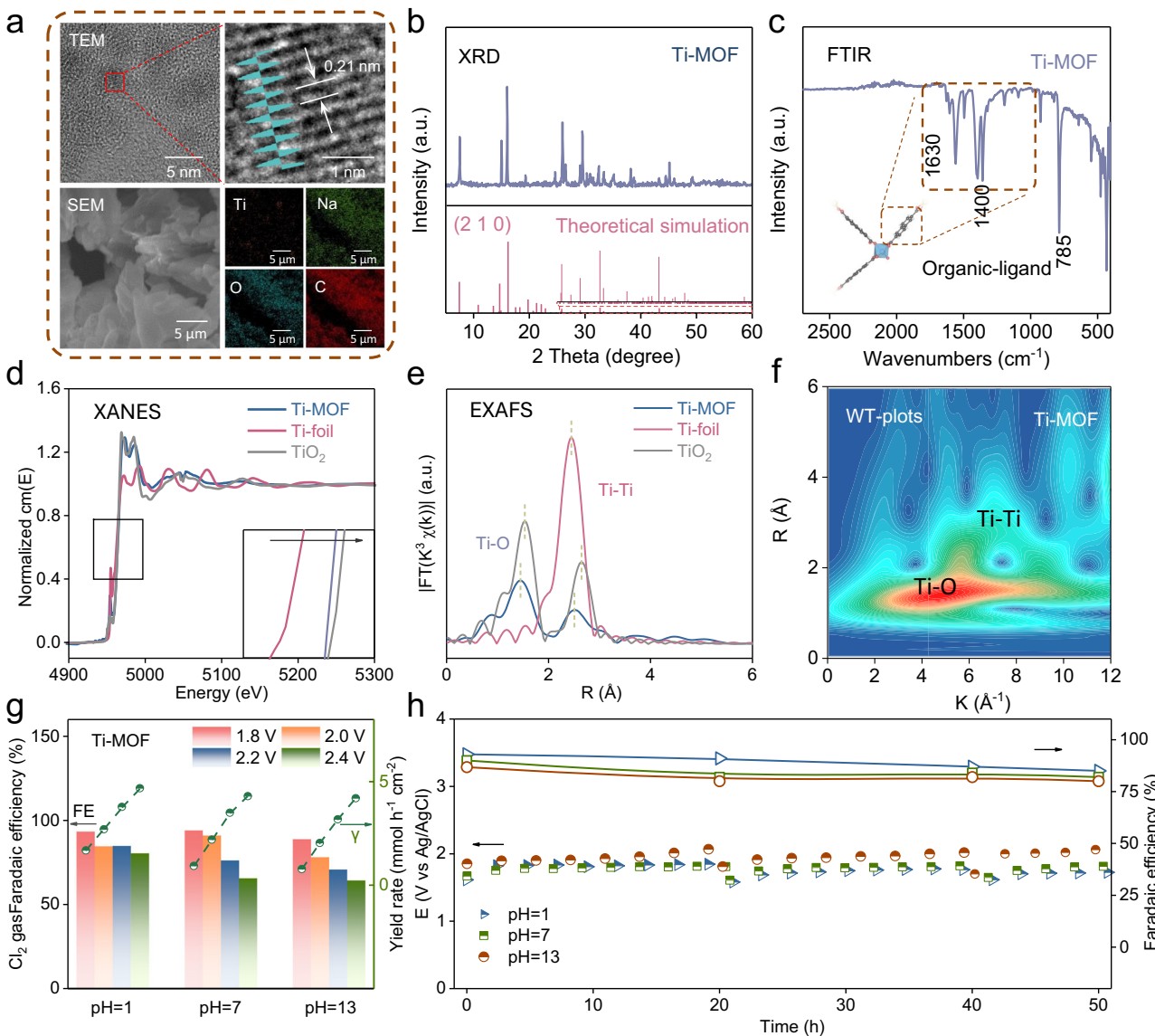

**Fig. 2 | Catalyst and system design for Cl$_2$ electrosynthesis beyond Pourbaix diagram. a** TEM and enlarge the image of the Ti-MOF catalyst. The blue wavy line form is the lattice spacing after FTT. **b** X-ray diffractometer (XRD) of Ti-MOF (In order to make the peaks ranging from 25 to 60° in the theoretical simulation more clearly, they were enlarged by a factor of 10). **c** Fourier Transform infrared spectroscopy (FTIR) of Ti-MOF. **d** X-ray absorption near edge structure (XANES) spectra of Ti-MOF, Ti-foil and TiO$_2$. **e** Extended X-ray absorption fine structure (EXAFS) and fitting curves spectra of Ti-MOF, Ti-foil and TiO$_2$. **f** The corresponding wavelet transform (WT)-EXAFS contour plots of Ti-MOF. **g** Cl$_2$ Faradaic efficiencies and yield rates at different pHs with Ti-MOF for Cl$_2$ evolution reduction. **h** Stability test at different pHs for Ti-MOF at 100 mA cm$^{-2}$. Source data for (**b**–**h**) are provided as a Source Data file.

0% at pH = 1, 7 and 13, respectively. This illustrates the above-discussed pH effects on catalysts. Analogously, the Cl$_2$ gas yield rates are 3, 0.21, and 0 mmol h$^{-1}$ cm$^{-2}$ at pH = 1, 7, and 13, respectively. The above phenomenon is consistent to the Pourbaix diagram applied to CER (Fig. 1c), which shows the Cl$^-$/Cl$_2$ couple in the pH < 4.3 range, HOCl/Cl$^-$ couple in the range of 4.3 ~ 7.5 and OCl$^-$/Cl$^-$ couple at pH > 7.5. Therefore, Cl$_2$ prefers to be produced in an acidic environment, while it easily hydrolyzes to HOCl and OCl$^-$ in neutral and alkaline conditions. Accordingly, it requires to develop pH-tolerant strategies and catalysts to circumvent the limitation of the Pourbaix diagram[24].

## pH-universal Cl$_2$ electrosynthesis

We have prepared a titanium-based metal-organic framework (Ti-MOF) catalyst by a facile two-step procedure, starting by the coordinate assembly of Ti nodes and 2,6-naphthoic ligand, followed by low-temperature calcination at 450 °C to improve its crystallinity.

Morphological characterizations show Ti-MOF comprised of nanosheet structures with Ti, O, C as the main components, where a small portion of sodium originates from the organic ligand precursor (please see the experiment sections). Close examination of a Ti-MOF nanosheet reveals its well-defined crystallinity with the adjacent lattice spacing of 0.21 nm (Fig. 2a and Supplementary Fig. 3), which agrees well with X-ray diffraction (XRD) containing a series of characteristic peaks corresponding to metal-organic framework architectures with dominant (2 1 0) crystal face (Fig. 2b and Supplementary Table 1)[25]. Further, the successful synthesis of Ti-MOF has been demonstrated by the Fourier transform infrared spectra (FTIR, Fig. 2c), showing characteristic vibrations of 1400–1630 cm$^{-1}$ of naphthalene ring from the organic ligand (Supplementary Figs. 4–6) and the vibrational peak at 785 cm$^{-1}$ from Ti-O bonding[26,27]. Notably, Ti-MOF can survive in pH-universal conditions, owing to the robust Ti-O bonding within the framework[28–30].

To further unravel the coordination environment inside Ti-MOF, we have conducted analyses of X-ray absorption near-side structure (XANES) and extended X-ray absorption fine structure (EXAFS) spectroscopy (Fig. 2d, e, Supplementary Figs. 7, 8). The Ti-MOF has presented XANES absorption edge energy between Ti-foil and $TiO_2$ (4962.95 *vs.* 4961.22 and 4963.46 eV), indicating its valence state of Ti in the range of 0 ~ +4. Different from Ti foil bearing only Ti-Ti coordination (distance: 2.89 Å), the Fourier transform EXAFS spectrum of Ti-MOF shows featured shells of Ti-Ti (3.12 Å) and Ti-O (1.92 Å) coordination. The Ti-Ti distance (3.12 Å) corresponds to interlayer interaction between Ti atoms in the form of multi-nuclear titanium clusters[31]. The Ti-O distance (1.92 Å) is significantly shorter as comparison to the Ti-C bond distance (2.14 Å, Supplementary Figs, 9, 10). Further, the coordination number of Ti foil is quantified as 6 (for Ti-Ti shell), while that of Ti-MOF is 4.05 for Ti-Ti shell (Supplementary Table 2). This result agrees well with XPS results (Supplementary Figs. 11, 12) as well as the wavelet transform (WT) diagrams (Fig. 2f).

It is known that WT analysis can serve as a qualitative supplement to EXAFS fitting, which can resolve overlapping coordination shells and validate scattering paths. Consequently, quantitative structural parameters of WT analysis have been derived from r-space fitting. As comparison to its precursor without calcination, the Ti-O coordination number of Ti-MOF only shows slight decline from 3.98 to 3.36, indicating calcinations treatment does only weakly affect at least the local metal-organic framework architecture. Ti atom can theoretically coordinate with six other atoms in the form of a hexacoordinated architecture, which represents the global symmetry constraints for Ti atoms. On the other hand, the coordination number of Ti-O in Ti-MOF has been experimentally determined by XAS (Fig. 2d, e and Supplementary Figs. 7, 8), where the average number is 3.98 (before low-temperature calcination) or 3.36 (final structure after calcination). This phenomenon indicates the evaporation of some organic ligands during the calcination process, which can expose more Ti active sites for electrochemical reaction.

Subsequently, we start the electrochemical performance evaluation for different samples. The gas diffusion electrode (GDE) substrate has shown low $Cl_2$ Faradaic efficiencies (Supplementary Fig. 13). The activities of Ti-MOF have been evaluated by tuning the pressure difference according to Bernoulli's principle (Fig. 2g, h and Supplementary Figs. 14–29, 30a–d). On one side of the GDE is the microporous layer loaded with Ti-MOF sample and the electrolyte circulated through the liquid chamber with a fixed rate of 8.4 mL min⁻¹, while on the other side is the gas chamber with air flow rate of 80 mL min⁻¹, which generates a pressure difference of $116.3 \times 10^{-3}$ Pa at the three-phase boundary according to the Bernoulli's equation (Fig. 2g, Supplementary Video 2 and Supplementary Fig. 30e). Without pressure difference, Ti-MOF has demonstrated diminished activities with elevated pH values, *i.e.*, Faradaic efficiency of 72.8% (yield rate of 2.28 mmol cm⁻² h⁻¹) at pH = 1, 5.8% (yield rate of 0.21 mmol cm⁻² h⁻¹) at pH = 7 and 0% (yield rate of 0 mmol cm⁻² h⁻¹) at pH = 13. With pressure difference, the activities of Ti-MOF show seldom fluctuations in the pH range of 1 ~ 13 at applied potential of 1.8 ~ 2.4 V. Specifically at the potential of 1.8 V with the Ti-MOF catalyst, the $Cl_2$ Faradaic efficiencies are 93.3, 94, and 88.8% at the pH of 1, 7, and 13, respectively.

For reliable compare with benchmark DSA, we have further loaded the main components ($RuO_2$ and $TiO_2$) of DSA on the gas diffusion electrode (GDE, Supplementary Fig. 31). By applying interfacial pressure difference, the activities increased significantly, *i.e.*, Faradaic efficiency of 86.3% (yield rate of 3.42 mmol cm⁻² h⁻¹) at pH = 1, 88% (yield rate of 3.3 mmol cm⁻² h⁻¹) at pH = 7 and 81.4% (yield rate of 3.3 mmol cm⁻² h⁻¹) at pH = 13. To facilitate comparison across diverse pH conditions, all of the potentials have been converted to RHE (Supplementary Fig. 32 and Supplementary Table 3). Furthermore, to investigate the dissolution of $Cl_2$ under different pH conditions, Supplementary Fig. 33 shows that when the pH becomes alkaline, More $Cl_2$ will dissolve in the electrolyte. Based on above data, we consider the pH is not a highly influential parameter anymore. In other words, we can move out of the limits of the Pourbaix diagram to produce $Cl_2$ efficiently across a wide range of pH conditions.

Notably, the CER activity of Ti-MOF is further validated by other experimental data. For example, we have investigated the effect of calcination temperatures of Ti-MOF on CER performances (150, 250, 350, 450, and 550 °C; Supplementary Figs. 34–37). The $Cl_2$ Faradaic efficiencies of Ti-MOF increase with calcination temperatures, starting from 38.6% at 150 °C to reach the maximum value (88.8%) at 450 °C, and finally decrease to 77.7% at 550 °C. Analogously, the $Cl_2$ Faradaic efficiencies are 38.6%, 66.6%, 64.7%, 88.8% and 77.7% at the temperatures of 150, 250, 350, 450, and 550 °C, respectively. Importantly, the Ti-MOF also exhibits strong stability in universal pH conditions (Fig. 2h). Over the period of 50 hrs at pH = 1, 7, 13, the Ti-MOF electrode has shown little morphology and structure decay after stability test (Supplementary Figs. 38–40).

## Standalone prototype device

Next, a standalone prototype device has demonstrated for membrane-free chlor-alkali process (Supplementary Video 3 and Supplementary Figs. 41–43). Figure 3a shows the schematic view of the prototype device, which is composed of a liquid chamber for cathodic HER/anodic CER, a gas chamber for carrier gas and external pumps driving NaCl electrolyte and gas flow circulating through the system. The prototype device is present in the form of a cuboid in the size of $24 \times 22 \times 12$ cm³ (Fig. 3b and Supplementary Video 3). Manipulated by triple-phase Bernoulli's principle, the pressure difference has formed that takes as-produced $Cl_2$ into gas chamber, as evidenced by KI titration solutions with brown $I_2$ formed during the operation (the top right corner of Fig. 3b). In addition, gas chromatography demonstrates the successful separation of $Cl_2$ and $H_2$ in the membrane-free prototype device (the bottom right corner of Fig. 3b and Supplementary Figs. 44, 45). Finally, The $H_2$ generated at the cathode is collected and measured by gas chromatography, gas-phase results (Supplementary Fig. 46) demonstrate that $H_2$ is discharged from the reaction system along with the electrolyte.

Next, we have evaluated the performances of the prototype device with fixed NaCl electrolyte rate (8.4 mL min⁻¹) and tunable gas flow rates in the range of 0 ~ 80 mL min⁻¹. It shows the $Cl_2$ Faradaic efficiencies closely affiliated with flow rates (Fig. 3c). For Ti-MOF, the $Cl_2$ Faradaic efficiencies increase by elevating flow rates, for example, 17.6% at 0 mL min⁻¹ (yield rate: 0.228 mmol h⁻¹ cm⁻²), 32.25% at 10 mL min⁻¹ (yield rate: 0.48 mmol h⁻¹ cm⁻²), 45.12% at 20 mL min⁻¹ (yield rate: 0.6 mmol h⁻¹ cm⁻²) and 75.57% at 30 mL min⁻¹ (yield rate: 0.978 mmol h⁻¹ cm⁻²). This phenomenon can be explained by the triple-phase Bernoulli equation: low rates of gas flow only result in the small pressure difference ($0 \sim 7.26 \times 10^{-3}$ Pa), making it difficult to take away the as-generated $Cl_2$ at the three-phase boundary. While beyond 30 mL min⁻¹, the $Cl_2$ Faradaic efficiencies have reached the maximum value (75.8 ~ 82.6% and yield rate: 1.02 ~ 1.08 mmol h⁻¹ cm⁻²) in the range of 40 ~ 80 mL min⁻¹ with moderate pressure difference ($16.3 \sim 116.3 \times 10^{-3}$ Pa). Notably, other than air flow, controlled experiments have been conducted in the same condition by using $CO_2$ flow and a vacuum condition. As expected, the $Cl_2$ Faradaic efficiencies are comparable to the air counterpart, thus demonstrating the generality for our system (Fig. 3c, Supplementary Fig. 47 and Supplementary Table 4, 5).

We have found the catalyst of Ti-MOF an important contributor in the membrane-free chlor-alkali process, which even outperforms benchmark DSA for $Cl_2$ electrosynthesis. As showing in Fig. 3d, e, the $Cl_2$ gas Faradaic efficiencies for Ti-MOF is 94% at the potential of 1.8 V, which is 18 times the one of DSA under the same test conditions and set-up. Analogously, the $Cl_2$ gas yield rate of Ti-MOF is 3.4 times the one of DSA at a potential of 1.8 V. Further with Ti-MOF, the

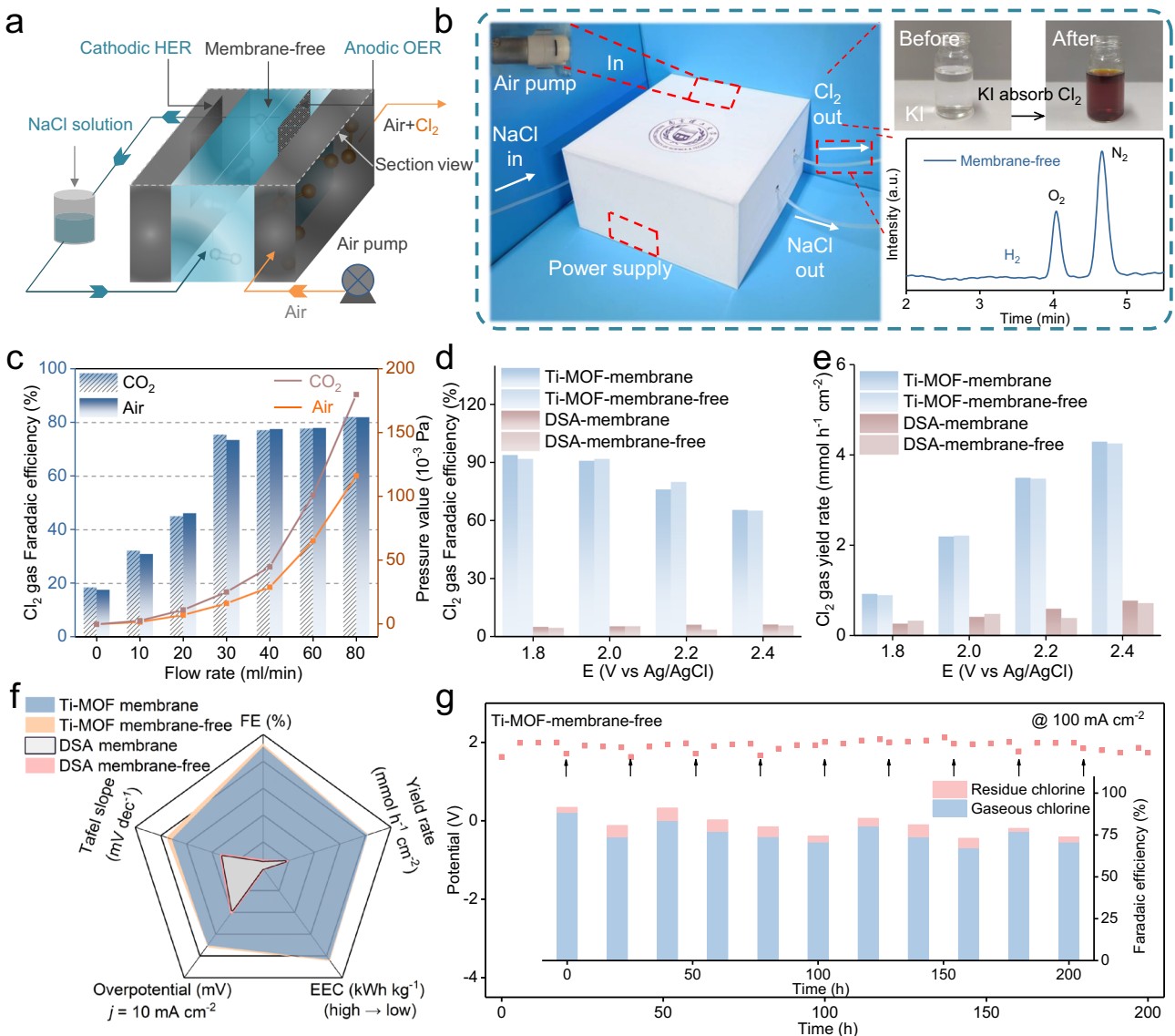

**Fig. 3 | Prototype device for Cl₂ electrosynthesis. a** The schematic diagram of the prototype device. **b** the optical picture of the prototype device operating before and after Cl₂ electrosynthesis, where the optical image shows a color change after Cl₂ absorption with KI, and gas chromatography shows seldom H₂ in the anodic gases. **c** The Cl₂ Faradaic efficiencies change with gas flow rates. **d**–**f** The Cl₂ Faradaic efficiencies (FE), yield rates and other activity comparisons for Ti-MOF and DSA in membrane-based/-free devices. **g** Stability test for Ti-MOF for 200 hours (at 100 mA cm⁻²). The arrows below the pompadour dots refer to flushing the electrodes every 20 h. Source data for (**b**–**g**) are provided as a Source Data file.

performances with and without membranes have been compared, which shows minor alternations in Cl₂ Faradaic efficiencies (94% *vs.* 92%) and yield rates (0.93 *vs.* 0.9 mmol h⁻¹ cm⁻² Fig. 3d, e and Supplementary Fig. 48). We have also compared the activities in terms of other indicators (like electricity consumption, overpotential and Tafel plot), all of which underpinning the excellent performances of Ti-MOF (Fig. 3f).

Finally, durability is a noteworthy merit for large-scale applications. As shown in Fig. 3g, the blue dots in the figure represent the potentials driven the long-term stability test of the prototype device, which shows little fluctuation for 200 h at a current density of 100 mA cm⁻². Analogously, the bar chart represents the overall Cl₂ Faradaic efficiencies, which can be divided into gaseous Cl₂ (blue) and the dissolved Cl₂ species (pompadour). The overall Cl₂ Faradaic efficiencies maintain stable over the whole 200-hr period (Supplementary Fig. 49), with little dissolved Cl₂ detected (pompadour, Faradaic efficiencies of 1.7% ~ 7.5%).

## Technical-economic evaluations

To further evaluate the performances under the operating conditions of the literature and industrial chlor-alkali electrolysis, the prototype device has worked under industrial operation conditions of 70 °C by collecting both gaseous Cl₂ and residue chlorine (denoted as active chlorine or AC)[7]. In spite of the "activity-selectivity trade-off" relationship from competitive OER (Supplementary Fig. 50), the productivity of 18.675 mmol h⁻¹ cm⁻² has been achieved with a Faradaic efficiency of 87.6% at a current density of 1.14 A cm⁻², which is competitive among all the chlorine synthesis technologies described in the literature. (Fig. 4a, b and Supplementary Table 6). In spite of the membrane-involving system achieving comparable activities under the same testing conditions, the separation membrane turned black under elevated temperature at high current densities (Supplementary Figs. 51, 52). Finally, the stability of the prototype device was tested under 1.14 A cm⁻² (Supplementary Fig. 53), which demonstrates durability for 40 hrs.

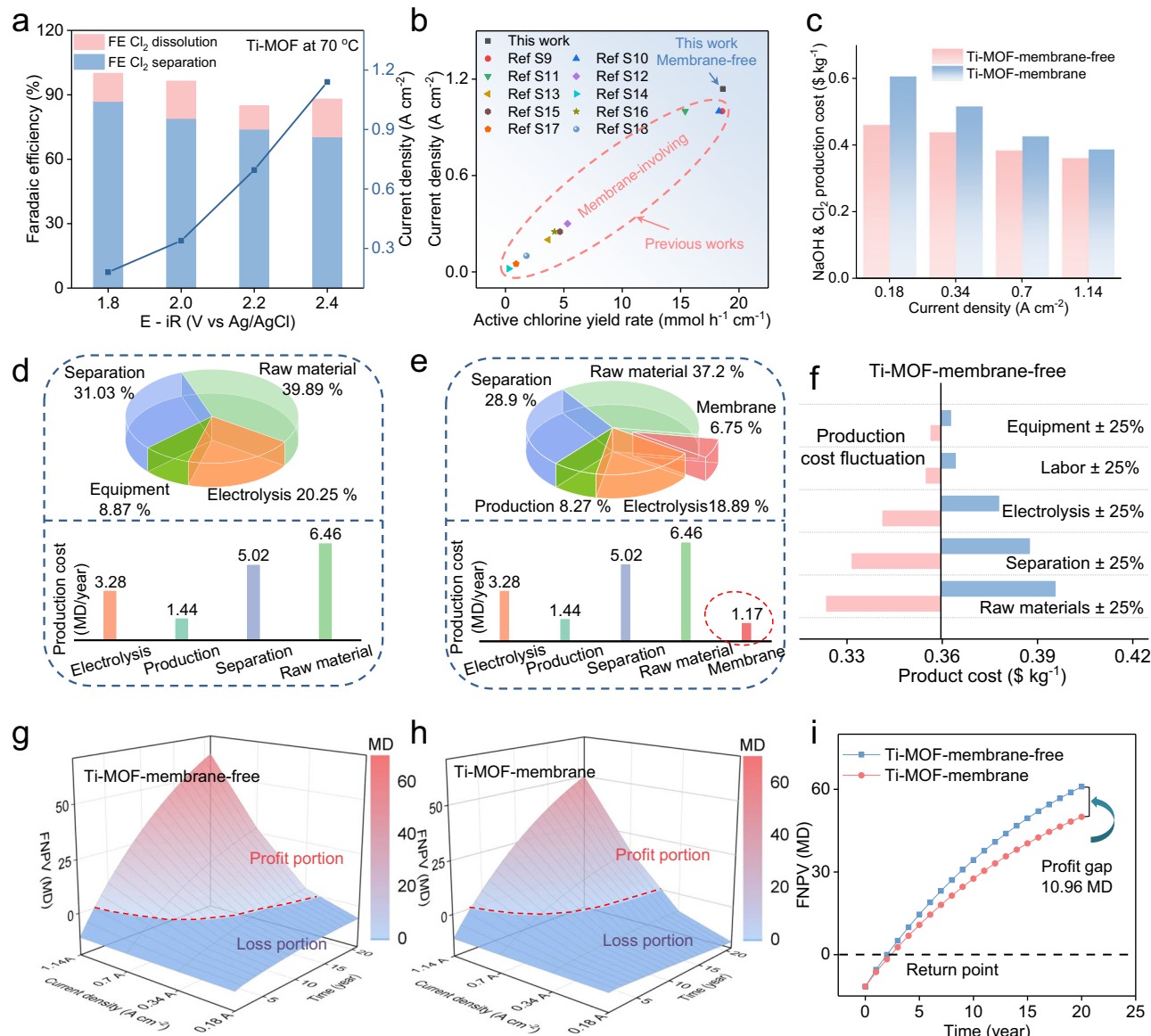

**Fig. 4 | Techno-economic analyses. a** The Faradaic efficiency and current density of active chlorine production ($Cl_2$ gas and residue chlorine in the solution, test compensation for $1\,\Omega$). **b** Active chlorine productivity compared with the literature. **c** The economic production cost of electrochemical production per kilogram of $Cl_2$ and NaOH for membrane-based and -free. **d, e** Production cost breakdown for different components. **f** Single factor analysis. **g, h** Schematic FNPV analyses. **i** Concrete FNPV analyses. Source data for (**a–i**) are provided as a Source Data file.

Consequently, we have analyzed the capital production cost over a 20-year production by techno-economics simulations, where the cost breakdown is presented as a function of current densities by normalizing both capital investment and operating cost (Fig. 4c, Supplementary Figs. 54–56 and Supplementary Tables 7–10)[32]. Overall, the capital cost of both membrane-free/based systems generally decreases with the elevated current densities from 0.18 to 1.14 A cm⁻² , while our membrane-free device has presented substantial cost saving as comparison to the membrane-based counterpart over the whole current density range. This result is consistent to single variable sensitivity analysis performed to examine the main parameters contributing to determining production cost (Fig. 4d, e). In both membrane-free/based systems, the capital cost is most susceptible to the variation in raw materials, followed by separation, electricity and production cost (labor and equipment), while the use of membrane has accounted for additional production cost of 6.75% (1.17 million dollar per year) in the membrane-based systems.

Consequently, the profit profile has been calculated from the difference between sales and cost after 25% tax deduction recorded as

financial net present value (FNPV)[20]:

$$FNPV = \Sigma_{(t=0)}^{n}(Cl - CO)_t \times (1+i)^{(-t)} \qquad (4)$$

where *Cl* is the present value of future cash flow (product revenue), *CO* is the present value of the original investment (capital and operating costs and 25% tax), *i* is the discount rates and *t* is the duration. The profits difference with/without the membrane has been intuitively described in Fig. 4g, h, where the red dotted line divides the boundary between loss and profit. Obviously, more sections of FNPV profit have been demonstrated for the Ti-MOF-membrane-free system as compared to the membrane-based counterpart, indicating the elimination of membrane can lead to economic savings. The above result is consistent to the payback duration profiles (Fig. 4i). In spite of both membrane-free/-based systems show similar payback duration within four years, the profit gap becomes significant over 20-year operation (60.44 *vs* 49.48 million dollars) originated from the removal of membrane in the system.

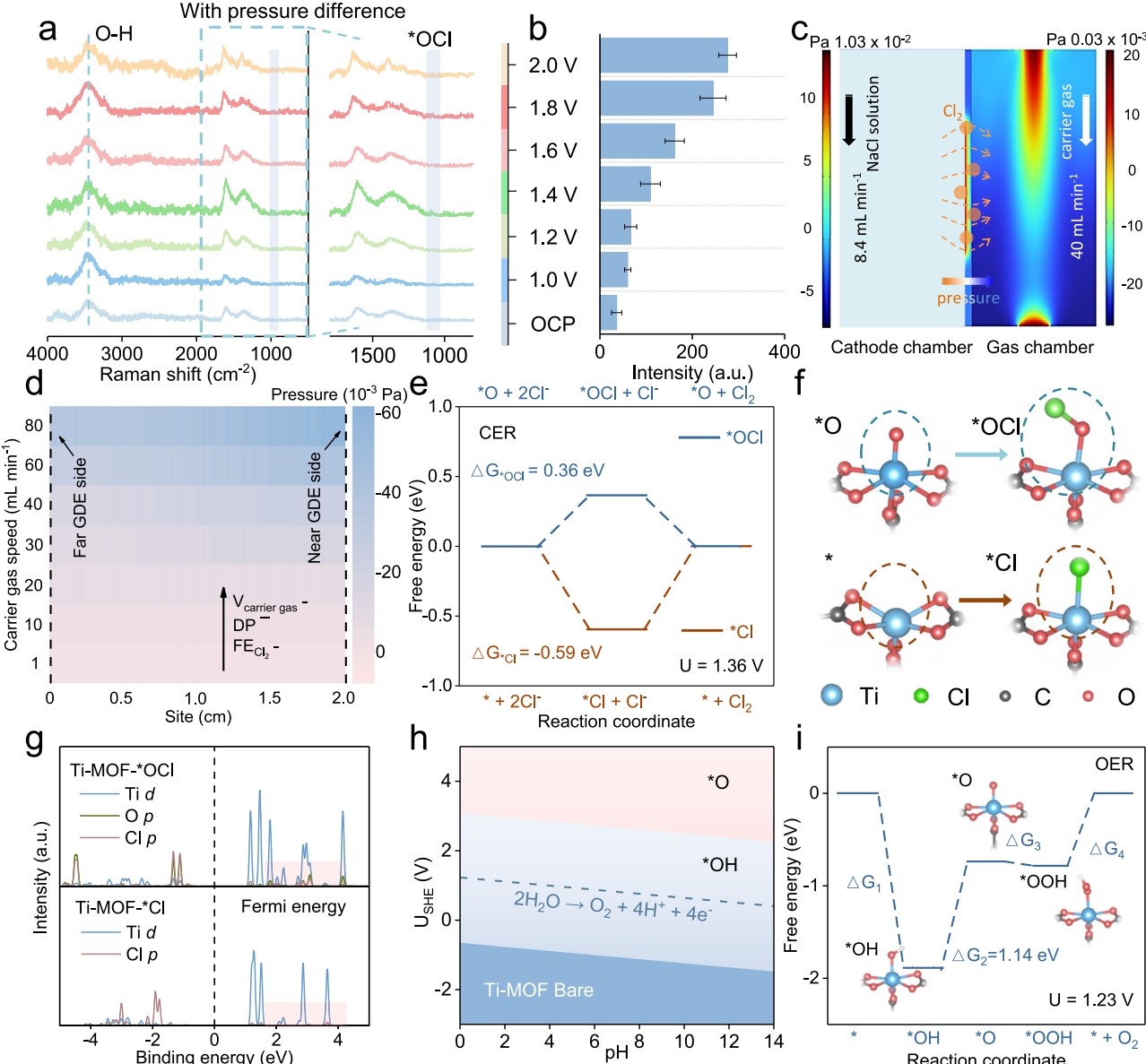

**Fig. 5 | Mechanism study. a** The *operando* Raman spectra. **b** Ti-*OCl Raman peak intensity. **c, d** Gas pressure distribution from finite element simulations. **d** Numerical quantification of pressures under different flow rates. **e, f** Density functional theory (DFT) calculations of free energy diagrams for two intermediate processes (*i.e.*, *O → *OCl and *→*Cl). **g** Density of states (DOS) profiles of Ti-MOF-*Cl and Ti-MOF-*OCl. **h** Calculated Pourbaix diagrams of Ti-MOF, showing the surface adsorption changes with applied potentials at different pHs. **i** The Gibbs free energy change of the side OER process. Source data for (**a–i**) are provided as a Source Data file.

Finally, we have made a corresponding comparison with DSA system with membranes, which is a benchmark process for current industrial chlor-alkaline process (Supplementary Fig. 57 and Supplementary Table 11). It shows that the incorporation of DSA significantly elevates NaOH/Cl₂ production costs from the industrial chlor-alkaline process, which underperform Ti-MOF-based membrane-free system in the range of 1~10 year's operation. After 11 years, the traditional DSA-membrane system begins to gain an advantage due to its lower power consumption. Overall, the production cost per kg of Cl₂/NaOH are comparable for the DSA-based membrane system and our Ti-MOF-based membrane-free system ($0.36768 *vs* $0.36069).

## Mechanism study

To unravel the mechanism for good performances, *operando* Raman spectra has been conducted to capture the short-lifetime intermediate species (*i.e.*, *OCl) in the CER process. The Raman measurements have been performed in a home-made flow-type cell equipped with a quartz

window that can detect the Raman signal from GDE at a 532 nm excitation wavelength. The applied potentials have been polarized from 1.0 to 2.0 V in 200 mV steps (Fig. 5a and Supplementary Fig 58). The *operando* Raman spectra has displayed a subset of vibrational data: the band at 1601 cm⁻¹ is assigned to the C = C stretching vibration of the benzene ring, while the band at 1374 cm⁻¹ corresponds to the symmetric stretching vibration of the carboxylate group (-COO⁻) in the MOF substrate.; the band at 3400 cm⁻¹ from H₂O; the potential-dependent band at 1093 cm⁻¹ from the Cl-O stretching mode of *OCl during CER process[33].

At the pressure difference of 116.3 × 10⁻³ Pa, the *operando* spectra begin to show the stretching modes of *OCl at -1.0 V. The peak intensity of this band increases continuously until reaching the maximum value at 2.0 V, indicating the generation and rapid accumulation of *OCl on the surface of the catalyst. Quantitatively, Fig. 5b and Supplementary Fig. 58b show the relative peak intensities of *OCl without/with pressure difference at different potentials, which agrees well with

the fact of $Cl_2$ produced at the electrode/electrolyte interfaces intermediately taken away by gas flow following triple-phase Bernoulli's principle, leading to decreased *OCl peak signal. To further verified above result, this presence of residue $Cl_2$ has been examined by local pH analyses performed in a RRDE system (Supplementary Fig. 59), which shows the increase of protons at electrode/electrolyte interfaces owning to the dissolution of $Cl_2$ in aqueous electrolytes ($Cl_2 + H_2O \rightarrow 2H^+ + Cl^- + ClO^-$)[20]. Therefore, it provides additional evidence of Bernoulli's principle contributing to separating *OCl species at elevated potentials, and according to Le Chatelier principle, promoting the rate-determining step (RDS) of *OCl-$Cl_2$ on the catalyst surfaces[34].

In order to understand more clearly the oriented $Cl_2$ migration following triple-phase Bernoulli's principle, we have conducted finite element simulations for the operating system (Figs. 5c, d and Supplementary Video 1). Figure 5c shows the pressure distribution map with pressure difference of $29 \times 10^{-3}$ Pa (Please see other pressure difference diagrams in Supplementary Figs 60–63), which shows the NaCl solution flowing at a constant rate of 8.4 mL $min^{-1}$ in the anode/cathode chamber (light blue area), and the carrier gas flowing at high rates (80 mL $min^{-1}$) but narrow areas from the inlet to outlet (warm color area). We can propose the as-generated $Cl_2$ at electrode/electrolyte interface would be driven by the negative pressure into gas chamber, which is then taken out by carrier gas away from the system. Notably, Fig. 5d presents the negative pressure generated by carrier gas at different flow rates and positions in the gas chamber. When the flow rate reaches 40 mL $min^{-1}$, the negative pressure generated by the flow rate begins to distinct change, which agree well with the experiment results (Fig. 3c) that $Cl_2$ selectivity begins to reach the plateau at the carrier gas flow rate exceeding 40 mL $min^{-1}$. So, we consider the interfacial pressure difference is the key to achieve pH-universal $Cl_2$ electrosynthesis.

Next, the reaction pathways of CER over Ti-MOF have been examined by theoretical calculations using density functional theory (DFT). According to XRD and XAFS results (Fig. 2b–f), the theoretical structure of Ti-MOF has been built on the basis of cluster model composed of Ti coordinated with 2,6-naphthalene (Supplementary Figs. 64–65, the structural information before and after optimization is presented in Supplementary Data 1). The reaction pathways of CER are proposed for Ti-MOF by using Gibbs free energy as a descriptor with potential correction of U = 1.36 V (Fig. 5e)[35]. Generally, one electron-transfer CER is composed of two cascade steps of the catalysts adsorbing chlorine-containing species, followed by the dissociation to $Cl_2$ for recovering the active centers[9]. Particularly in the formation of chlorine-containing species, two intermediates might appear according to their chlorine sources (Fig. 5f): the local modeling diagram for *O → *OCl pathway and the lower part of * → *Cl pathway[36,37]. As showing in Fig. 5e, the energy for *O → *OCl is 0.36 eV, while * → *Cl is −0.59 eV. However, the required energy for dissociation to $Cl_2$ is −0.36 eV for *OCl → *O + $Cl_2$ while 0.59 eV for *Cl → * + $Cl_2$. Consequently, the Ti-MOF prefers the *OCl reaction pathway (*O → *OCl→ *O + $Cl_2$) with an energy barrier of 0.36 eV than the *Cl pathway of 0.59 eV (* → *Cl→ * + $Cl_2$) under the same computation condition.

To understand the above results from an electronic perspective, further theoretical calculations have been performed for the density of states (DOS, Fig. 5g and Supplementary Figs. 66 – 67) and differential charge density (Supplementary Figs. 68–71) for Ti-MOF adsorbed by *OCl and *Cl intermediates. In the DOS profiles in Fig. 5g, Ti-MOF-*OCl has demonstrated higher and sharper peak intensity near the Fermi energy level as comparison to Ti-MOF-*Cl, which reflecting the stronger adsorption of *OCl on the Ti-MOF surface than *Cl counterpart[36]. This result is consistent with the larger overlap in the binding states of 0.85 ~ 3.59 eV between O p orbital and Cl p orbitals for Ti-MOF-*OCl than *Cl counterpart, which explains the large energy barrier for the formation of *OCl as comparison to *Cl (0.36 vs. − 0.59 eV)[9,38]. On the other hand, the subsequent intermediate step of dissociation to $Cl_2$ has

been examined by the three-dimensional schematic illustration of charge density electron/proton transfer occurs between Ti-MOF-*OCl in comparison to that of the *Cl counterpart[36]. Quantitative analyses shows the charge transfer of 0.2 eV for Cl→O inside Ti-MOF-*OCl while 0.48 eV for Cl→Ti inside Ti-MOF-*Cl. As a consequence, it requires small energy to break Cl-O binding of Ti-MOF-*OCl to generate $Cl_2$ as comparison to Cl-Ti of Ti-MOF-*Cl (− 0.36 vs. 0.59 eV) (Phonon spectrum information showed in Supplementary Tables 12−13).

The strong-resistant property of Ti-MOF for pH-universal CER has been explained (Fig. 5h), whose surface forming energy (SFE) representing the durability in different applied potentials and pH values. Without applied potentials (or negative SPF), the Ti-MOF material situates at the dark blue area characteristic of good stability in the pH range from 1 to 13. Next, the Ti-MOF has been polarized positively, which generate *OH at the local environment owning to competitive oxygen evolution (OER). The Ti-MOF can still maintain its structure in pH=1 ~ 13 (Please see the blue area). Further, the deprotonation of *OH to *O occurs at 1.8 − 2.4 V (vs. Ag/AgCl) that initialize the *OCl reaction pathway (*O → *OCl→ *O + $Cl_2$), and the Ti-MOF can still maintain its durability in all pH ranges (as shown in the pompadour area)[39]. Ti-MOF catalyst prefers to promote CER to side OER, as evidenced by its low energy barrier of 0.36 as comparison to 1.14 eV for OER under the same condition (Fig. 5i, Supplementary Figs. 72,73−78, intermediate model and properties and Phonon spectrum information showed in Supplementary Tables 14−16)[40].

Finally, we have demonstrated the generality of our configuration for water splitting (Supplementary Fig. 79a). In a similar electrolysis design, the system produces $H_2$ in 1 M KOH electrolyte, which is taken away by carrier gas (i.e., $CO_2$) from GDEs according to Bernoulli's principle. The gas chromatography data shows the characteristic peak of $H_2$ product during the test (Supplementary Fig. 79b). The $H_2$ Faradaic efficiency increases with flow rates and pressure difference (Supplementary Fig. 79c), and comparable for both membrane-based and -free configurations at different potentials (Supplementary Fi. 79d).

## Discussions

In this work, an efficient yet cost-effective $Cl_2$ electrosynthesis system has been developed. Mechanism studies reveal that a gas diffusion electrode design using the triple-phase Bernoulli's principle to remove the $Cl_2$ from the locus of generation allow to break the restriction of Pourbaix diagram, and superior $Cl_2$ productivity under pH-universal conditions are obtained. This has allowed assembling a standalone prototype membrane-free chlor-alkali device for producing $Cl_2$, where activities and stability with minor performance changes after prolonged operation could be demonstrated. This work not only provides clues for scalable $Cl_2$ production, but also has general implications by providing an additional dimension to manipulate versatile systems such as water splitting, urea electrolysis and ammonia electrolysis beyond classical rule.

## Methods
### Materials synthesis
Synthesis of pH-tolerant Ti-MOF catalyst. 2,6 naphthoic acid (5 g) and sodium hydroxide (1.85 g) were mixed in a water/ethanol solution (1/1 vol) and dried overnight. Next, a portion of above sample (2.916 g) was dispersed in a mixed solution of N,N-Dimethylformamide (28 mL) and methanol (3 mL), where tetraisopropyl titanate (1.153 g) was added with vigorous stirring, and heated at 150 °C for 24 hrs. The as-obtained product was finally calcined at 450 °C for 3 hrs at a heating rate of 5 °C $min^{-1}$ under argon atmosphere.

### Materials characterization
Transmission electron microscopy (TEM) was conducted on a JEOL JEM-F200 TEM. X-ray diffraction (XRD) was conducted on a SmartLab SE with Cu Kα radiation operating at 40 kV and 30 mA. X-ray

photoelectron spectroscopy (XPS) was collected on a Thermo Scientific ESCALAB 250Xi with Al Kα X-ray source. Fourier transform infrared (FT-IR) was conducted on a Nicolet iS20 FT-IR spectrometer. Ti K-edge X-ray absorption near edge structure (XANES) and extend X-ray absorption fine structure (EXAFS) experiments was conducted at Shanghai Synchrotron Radiation Facility. Gas was detected in the PANNA A91 Plus gas chromatography.

## Electrochemical analyses

Electrode fabrication. The dimensionally stable anode (DSA) was purchased commercially and used without further treatment (electrode area: $1 \times 1\,cm^2$). The Ti-MOF electrode was prepared by a spray-coating procedure: firstly, the catalyst ink was prepared by mixing Ti-MOF (5 mg), acetylene black (1 mg) and nafion (50 μL) in isopropyl alcohol (950 μL) with mild ultrasonication. Next, the ink (300 μL) was filled into a pipette gun and sprayed onto a gas diffusion electrode ($1 \times 1\,cm^2$) with the catalyst loading of $1.5\,mg\,cm^{-2}$. More details of materials, electrodes and reactor unit are shown in Supplementary Figs. 30a–d.

Electrochemical testing setup. As shown in Supplementary Fig. 30e, our test was conducted in a flow-type reaction device. The anode and cathode have faced each other, and the reference electrode is placed between the anode and cathode. The middle chamber is connected by the electrolyte, while the rightmost chamber allows the carrier gas to pass through. Both the electrolyte and the carrier gas enter through the lower liquid (gas) inlet and exit through the upper liquid (gas) outlet. All the tests were conducted in an air environment, and the room temperature was maintained at around 20 °C except for the heating test.

We have evaluated the performances of the prototype device with a fixed NaCl electrolyte rate ($8.4\,mL\,min^{-1}$) and tunable gas flow rates (Supplementary Video 2). The reaction chamber can hold an electrolyte volume of $1 \times 2 \times 2\,cm^3$. Since DSA is manufactured by coating an electrocatalytic metal oxide layer (such as $RuO_2$, $IrO_2$) on a titanium substrate, its structure is a solid titanium plate, so it is impossible to apply pressure difference treatment to the back of the electrode. Therefore, the effective components ($RuO_2$ and $TiO_2$) were used as a surrogate for DSA. In details, the $RuO_2$ and $TiO_2$ mixture with a mass ratio of 1:1 was prepared as ink and sprayed onto GDE under the same mass loading.

Electrochemical testing. All measurements were conducted with CHI 660 Electrochemical Analyzer in a flow-type cell using Ti-MOF as the working electrode, iridium-plated titanium oxide as the counter electrode, Ag/AgCl as the reference electrode in 5.0 M NaCl aqueous electrolytes (with pH adjusted by HCl or NaOH). Place the electrode to be calibrated and the standard reference electrode in a saturated potassium chloride solution, and record the open-circuit potential (OCP). Ensure that the potential remains within the ± 5 mV range. At this point, the reference electrode can be used. Due to the fact that the pH value of the electrolyte fluctuates with temperature, according to the test, the range of the pH value change of the electrolyte is less than ±0.5. All electrolytes should be placed in a well-ventilated and safe area, and the electrolyte should be used immediately after it is prepared. The polarization curve was collected using a potentiostatic method with a potential range from 1.8 V to 2.4 V (vs. Ag/AgCl) at a $100\,mV\,s^{-1}$ scan rate. All CV, LSV curves were corrected with iR compensation (90%). All compensated potentials are marked, and the uncompensated potentials are written normally. The impedance test was conducted using parameters of $10^5$ for high frequency and 0.01 for low frequency.

Electrochemical IR correct compensation and heating testing. All measurements were conducted with CHI 660 Electrochemical Analyzer connecting the high current amplifier in a flow-type cell using Ti-MOF as the working electrode, iridium-plated titanium oxide as the counter electrode, Ag/AgCl as the reference electrode in 0.1 M NaOH + 5.0 M NaCl aqueous electrolytes. The IR correct compensation is performed by manually entering the ohmic value of the compensation (1 Ω) into the electrochemical workstation software. The heating experiment is to heat the electrolyte in a water bath and then pass it into an electrolytic cell.

## Prototype device fabrication

The general structure of the membrane-free device is similar to a conventional liquid flow electrolyzer. However, the membrane-free device has made some adjustments in some details. As shown in Supplementary Fig. 42, after the removal of the proton membrane, there is no distinction between the anode chamber and the cathode chamber. However, due to the special structure of the flow cell, the chamber where the anode and cathode are located is referred to as the anode region and the cathode region. The electrolyte flows into the entire chamber from the bottom of the anode region at a rate of $8.4\,mL\,min^{-1}$. The direction of the liquid flow is mainly towards the anode region for filling, but also towards the cathode region (the multiple black arrows in the Supplementary Fig. 43). Finally, the electrolyte flows out from the upper part of the cathode region. The purpose of this is to create a pressure potential towards the cathode region for the hydrogen generated in the chamber, which can reduce the hydrogen diffusion in the flow cell to some extent. In order to more effectively reduce the movement of hydrogen in the entire electrolyzer, we attached a tape to the upper part of the gasket window, covering an area of about $0.5\,cm^{-2}$ (Supplementary Fig. 43a). Supplementary Fig. 43b uses a simplified schematic to explain the mechanism of covering this area. When $H_2$ forms at the Anode, it will float towards the upper part of the chamber due to its lower density. Most of the $H_2$ will be discharged with the liquid flow. However, there will be some $H_2$ that cannot be discharged in time, causing it to agitate in the chamber, which increases the risk of explosion due to contact with the generated $Cl_2$ and disturbing the electrode surface material. By making the above modifications, the membrane-free device has been assembled. Notably, to prevent electrolyte penetration into gas chamber, the counter electrode was engineered with the specialized design shown in Supplementary Fig. 80. The feasibility of this approach has been demonstrated through contact angle measurements and PTFE waterproof testing (Supplementary Fig. 81).

## The determination of $Cl_2$ product

$Cl_2$ is determined by the traditional KI titration method, which reacts with $Cl_2$ to change the colorless KI to brown $I_2$. Next, the above mixture was mixed with starch solution (1 mL, 1 wt%), and titrated with 0.1 M $Na_2S_2O_3$ solution. The end point of titration is determined by the observation of the blue-black color disappearing after the last drop of 0.1 M $Na_2S_2O_3$ into the absorption solution with no longer color change within half a minute. The relevant reaction equations are listed as follows:

$$Cl_2 + 2KI = I_2 + 2KCl \tag{5}$$

$$I_2 + 2Na_2S_2O_3 = 2NaI + Na_2S_4O_6 \tag{6}$$

The $Cl_2$ yield ($mol\,h^{-1}\,cm^{-2}$) and Faradaic efficiency (FE) for the $Cl_2$ production were calculated according to the equations:

$$Yield = \frac{0.1V_{Na_2S_2O_3}}{2t} \tag{7}$$

$$FE(\%) = \frac{0.1FV_{Na_2S_2O_3}}{It} \tag{8}$$

Where $V_{Na_2S_2O_3}$ is the $Na_2S_2O_3$ consumed by titration of $I_2$ in the total absorption solution (L), $t$ is reaction duration, $F$ is the Faraday constant (96485 C mol$^{-1}$) and $I$ is the applied steady current.

To ensure greater accuracy of the results, in addition to the KI titration method, the FAS-DPD titration method and the amperometric titration method were also employed. The specific methods are described in the Experimental method section of the Supplementary information. Moreover, the test results are shown in (Supplementary Fig. 82), indicating that the results obtained using the KI titration method in this study are reliable over a wide range. Therefore, in the lower part of Supplementary Fig. 82, we analyzed the error of Faraday efficiency. And in other parts, the data were selected to be repeatedly tested, and the more average values were presented.

## Standardization of $Na_2S_2O_3$

The accurate concentration of $Na_2S_2O_3$ was calibrated using the potassium dichromate method:

Weigh a certain mass (accurate weighing) of dried $K_2Cr_2O_7$, and dissolve it in a specific volume of deionized water.

Add a KI solution acidified with dilute sulfuric acid, shake well, then place the mixture in a sealed dark environment and let it stand for 3 – 5 min.

Titrate with the $Na_2S_2O_3$ solution to be calibrated until the solution turns pale yellow, then add 2 mL of starch indicator (the solution will turn dark blue at this point). Continue titrating until the blue color of the solution fades and does not revert within 30 seconds-this is the endpoint. Record the volume of $Na_2S_2O_3$ solution consumed. Repeat the operation 3 times in parallel and take the average concentration.

The accurate concentration of the $Na_2S_2O_3$ solution can be calculated from the titration results:

$$c(Na_2S_2O_3) = \frac{6 \times m(K_2Cr_2O_7) \times 1000}{M(K_2Cr_2O_7) \times V(Na_2S_2O_3)} \quad (9)$$

Among them, 6 represents the reaction molar ratio, m($K_2Cr_2O_7$) is the mass of $K_2Cr_2O_7$ weighed, M($K_2Cr_2O_7$) = 294.18 g mol$^{-1}$, and V($Na_2S_2O_3$) is the volume of the $Na_2S_2O_3$ solution used for the titration.

## Calibration of gas chromatography

Five mixed standard gases containing different concentrations of $CH_4$, $C_2H_6$, $C_2H_4$, $C_3H_8$, and CO (however, these gases are not involved in this work, so no analysis of carbon-containing gases will be conducted), as well as $H_2$, $O_2$, and $N_2$ were used to calibrate the relationship between concentration and gas phase signal.

In the most recent calibration, we used standard gases with $H_2$ contents of 1.98, 0.0044, 5.9167, 3.843, and 0.02 mol mol$^{-1}$. The peak areas measured by the instrument were 1078.87, 2.38, 3223.48, 2105.32, and 12.23 µV*s, respectively. From these data, the relationship between $H_2$ concentration and peak area in the gas phase was calculated as: c = 0.00183*s − 0.0024. When conducting the $H_2$ measurement and creating the standard curve, the key criterion is that the standard curve of $H_2$ must consist of at least five standard gas calibration curves, and the $R^2$ value of this curve should be greater than or equal to 0.9999. When measuring $H_2$ in the gas bag, since $H_2$ is lighter, it is necessary to thoroughly shake the gas bag and maintain an intake time of more than 30 s. This will yield more accurate data.

The calculations or data for the molar amounts or yields of each gas is as follows:

Since we have conducted a quantitative analysis of $H_2$, the calculation formula for the Faraday efficiency of $H_2$ is as follows:

$$FE_{H2} = \frac{Q_g}{Q_{total}} \times 100\% = \frac{nzF}{Q_{total}} \times 100\% = \frac{(0.0018s - 0.0024) \times v_{carrier\ gas} \times T \times zF}{Q_{total} \times V_m}$$
$$(10)$$

In the formula: s represents the measured area of $H_2$ in the gas phase (µV s),$v_{carrier\ gas}$ is the gas flow rate during the reaction process (mL min$^{-1}$), T is the reaction time (s), z is the number of transferred electrons, F is the Faraday constant, F = 96485 C mol$^{-1}$, $V_m$ is the molar volume of the gas, which is 24.5 L mol$^{-1}$ at normal temperature and pressure.

## Density function theory (DFT) computation methods

Structure optimization. First-principles calculations were performed by using MedeA-VASP based plane wave set approach[41,42]. The electron-ion interactions were described by the density function method, and the electron-electron exchange correlations were described by the Perdew-Burke-Ernzerhof (PBE) functional[43,44]. The Ti-MOF model framework has been built on the basis of experiment results and literature, including: XRD patterns (Fig. 2b)[28], FTIR (Fig. 2c)[27]and X-ray absorption spectroscopy (XAS, Fig. 2d, e and Supplementary Tables 1–2).

For structure optimization, the RMM-DIIS update algorithm was used. We used low precision for efficient computation and normal precision for accurate computation. The convergence is 0.06 and 0.05 for efficiency and accuracy computation. Both efficiency and accuracy computation use 400 eV planewave cutoff. A Gaussian smearing with a width of 0.05 eV was also utilized. For the surface Brillouin zone integration, a $1 \times 1 \times 1$ Monkhorst−Pack kpoint mesh was used. The convergence criteria for electronic self-consistent iteration and ionic relaxation were set to $10^{-4}$ eV and $10^{-5}$ eV. Hubbard-U correction method (LSDA + U) was carried out to improve the description of highly correlated Ti with the value of U = 2.5 eV set to standard LDA or GGA.

## Computational details

Based on Bernoulli's principle for fluid dynamics, the local pressure of gas flowing through a conduit decreases as its flow velocity increases[45]. Consequently, the gas pressure within the backside region of the electrode in the gas chamber diminishes with elevated flow rates. This hydrodynamic relationship is further operationalized in our simulations through Darcy law:

$$u = -\frac{k}{\mu} \nabla p \quad (11)$$

which provides a concrete framework for modeling such systems. Given that the gas-liquid flow within the chambers can be regarded as laminar, the Darcy law offers an appropriate theoretical foundation for these calculations.

Theoretically, Bernoulli principle a basic rule in fluid dynamics illustrating the relationship between the surface velocity and pressure. In the present work, two fluids have presented at the three-phase boundary, carrier gas flow and electrolyte flow, and according to Bernoulli principle that can be illustrated as:

$$P_1 + \frac{1}{2}\rho_1 V_1^2 + \rho_1 g h_1 = P_2 + \frac{1}{2}\rho_2 V_2^2 + \rho_2 g h_2 \quad (12)$$

where 1 represents carrier gas flow and 2 represents electrolyte flow. So, the interfacial pressure difference ($\nabla p$) can be illustrated as:

$$\nabla p = \frac{1}{2}\rho_1 V_1^2 + \rho_1 g h_1 - \frac{1}{2}\rho_2 V_2^2 - \rho_2 g h_2 \quad (13)$$

Quantitively, the pressure difference has been calculated by software (Fig. 5c, d, Supplementary Video 1), showing the pressure difference in the range of 0 - 116.3 × 10$^{-3}$ Pa with the carrier gas flow of 0 - 80 mL min$^{-1}$ and electrolyte flow of 8.4 mL min$^{-1}$. Therefore, when high-velocity carrier gas flow and low electrolyte flow have been

applied to the three-phase boundary, it will generate a large localized pressure gradient. This can promote $Cl_2$ bubbles migration away from electrolyte flow, and into carrier gas (Please see Supplementary Video 2).

## Model description

Complete model with anode gas chamber, gas diffusion layer, catalytic layer and electrolyte chamber is set up at the steady state with two-dimensional (2D) finite element approaches. Free software (like OpenFOAM, FreeCAD and SALOME) is applied to solve the finite element analysis and simulation problem. In order to analyze the influence of anode carrier gas flow rate on the overall pressure and gas flow state of the system, which is shown in Supplementary Fig. 60, following sections, including flow equations and mass transfer in porous medium are applied.

## Mass Transport in the porous media

Darcy law and mass transfer equations are used to describe the gas transport in CDL and CCL:

$$u = \frac{\kappa}{\mu}\nabla p \tag{23}$$

$$\kappa = \frac{dp^2}{180}\frac{\varepsilon^3}{(1-\varepsilon)^2} \tag{24}$$

Where J is the flux of carried gas (kg/(m^2*s)), ρ is the density (kg/(m^3)), μ is Dynamic viscosity (Pa*s), $D^{eff}$ is the effective diffusion coefficient ($m^2$/s), κ is the permeability ($m^2$) of the medium, dp is particle diameter (m) and ε is the porosity of porous medium.

## Mass Transport in the anode gas chamber and anode electrolyte chamber

The $CO_2$ in the gas chamber and the liquid in the electrolyte chamber can be regarded as laminar flow, so it can be described by the following equation:

$$\rho(\mu\nabla)\mu = \nabla(-pI + K) - (\mu\kappa^{-1})\mu + F \tag{25}$$

$$\rho\nabla\mu = 0 \tag{26}$$

$$K = \mu(\nabla u + (\nabla u)^T) - \frac{2}{3}\mu(\nabla u)I \tag{27}$$

Where u is the speed (m/s) of gas or liquid.

## *Operando* Raman spectroscopic characterizations

*Operando* Raman was conducted on CHI1140C electrochemical workstation equipped with a HORIBA LabRAM HR Evolution Raman spectrometer with 532 nm solid laser as an excitation source. A self-made flow-type cell with a quartz window was used to detect the Raman signal from the catalyst electrode (prepared by loading catalysts onto GDE). Pt foil and Ag/AgCl electrode were used as the counter and reference electrodes, respectively. The concentration of NaCl electrolyte was reduced to 1 M to reduce the interference of $Cl^-$ on the Raman signal.

## Data availability

All relevant data that support the findings of this study are presented in the manuscript and supplementary information file. The experimental datasets associated with this work are provided in the form of Source Data files. Source data are provided in this paper.

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

## Acknowledgements

The authors acknowledge the financial and equipment support from National Natural Science Foundation (Grant No. 52572232 and 52376193), Jiangsu Natural Science Foundation (Grant No. BK 20230097) and the BL20U1, BL17B and BL14W1 beamline at the Shanghai Synchrotron Radiation Facility.

## Author contributions

S.C. supervised the project and designed the experiments. Z.H.N., G.L.X., and J.J.D. performed experiments and theoretical calculations. All authors discussed the results and assisted with the paper preparation.

## Funding

## Competing interests

S.C. has filed a Chinese provisional patent application (no. 202610016731.8) based on this work. The other authors declare no competing interests.
