## [Transparent Peer Review file · Nature Communications]

Bernoulli's principle-mediated Cl₂ electrosynthesis

Corresponding Author: Professor sheng Chen

Version 0:

Reviewer comments:

Reviewer #1

(Remarks to the Author)

This manuscript by Nie et al. present a breakthrough in chlorine (Cl₂) electrosynthesis by integrating Bernoulli's principle with a pH-tolerant Ti-MOF catalyst. The novel triple-phase gas diffusion electrode design enables oriented Cl₂ migration via pressure differences, eliminating membrane requirements and preventing H₂/Cl₂ crossover. The manuscript is well-organized, clearly presented, and the results are convincing. However, there are still some questions need to be illustrated clearly. Also, the conclusion needs to be further revised under solid evidence. Below are specific critiques to address these issues.

1. The abstract states Faradaic efficiencies of 96.3%~87.6% within 0.1~1.14 A cm⁻², yet fails to clarify why the efficiency decreases with increasing current density.
2. The working potential of CER is 1.8 V-2.4 V in this manuscript, which is relatively high compared to other literatures. Is this system really energy-efficient?
3. While Bernoulli's principle explains Cl₂ migration via carrier gas, the separation mechanism for Cl₂ and carrier gas is unaddressed. Does the mixture of carrier gas with Cl₂ cause Cl₂ to be impure?
4. Grammar mistake: please revise recurrent misuse of "despite of" to "in spite of" or "despite".
5. Please correct the anodic CER equation in introduction: The electron transfer number should be 2 not 1.
6. Replace "mPa" into "MPa" throughout the manuscript.
7. Grammar mistake in introduction: subject-verb agreement error of "circumvents" → "circumvent".
8. In figure 2a, lattice spacing of 0.2 nm is not clear, which should be redone to better prove the (210) crystal face.
9. Why in Figure 2e there exist Ti-Ti and Ti-O, but in Figure 2f authors only mention Ti-O?
10. Why the coordination number of Ti-O in Ti-MOF is 3.183, but authors speculate Ti node is octahedrally coordinated? What's the relationship between the two?
11. In Figure 3b, gas chromatography doesn't detect H₂. Where does the H₂ go? Is H₂ be collected?
12. The plateau in Cl₂ Faradic efficiency beyond 40 mL min⁻¹ gas flow is not explained mechanistically in Figure 3c. Please clarify whether this stems from gas-liquid equilibrium constraints or kinetic bottlenecks.
13. In Figure 3d-e, the control experiment of DSA-membrane should be added.
14. In Fig. 3g, the "dissolved Cl₂ species" (brown) color scheme is inconsistent with the legend. Please replot color.
15. The techno-economic analysis assumes industrial operation at 70°C. Justify this temperature is along with actual chlor-alkali industry standards with literature citations.
16. Why Cl₂ is counted as "cost" in Figure 4c?
17. In technical-economic evaluations section, the profits difference is described in Figures 4 g-h not Figures 5 g-h. The same goes for payback duration profiles.
18. In Figure 5a, there seems no change in *OCl peak at all. Please reexamine experimental conditions.
19. Fig. 5h exhibits mismatched sector colors between graphical representation and textual description.
20. The authors utilized Bernoulli's principle to regulate gas and liquid flow rates, forming a pressure difference of 116.3 mPa to promote Cl₂ outflow. However, it remains unclear whether this pressure difference remained stable during the reaction process. Furthermore, complex factors inherent to electrochemical reactions could significantly alter the interfacial pressure, such as gas product evolution and interfacial mass transfer. Critically, the authors determined this value solely through calculation using the Bernoulli equation, without providing experimental evidence to substantiate that this minute pressure difference was the key factor enabling efficient Cl₂ synthesis.
21. The authors proposed the *OCl pathway in the CER process, where the Cl⁻ adsorbs on the Ti-O site. However, the outermost electrons shell of Cl⁻ is already full, and it's extremely difficult and unreasonable for Cl⁻ to adsorb onto the Ti-O

site. Please provide a reasonable explanation.

22. In Figure 5a, the intensity of the Raman characteristic peaks corresponding to Ti-*OCl didn't exhibit an obvious dependence on applied potentials. In addition, no significant correlation can be identified between the peak intensity in Figure 5a and the intensity analysis diagram in Figure 5b. Please optimize the experimental conditions to obtain more conclusive results.

23. The experimental data and supplementary videos could only confirm the absence of H₂ in the flowing gas, and cannot determine whether H₂ and Cl₂ are mixed within the reactor. Please design experiments to prove the successful separation of the generated H₂ and Cl₂.

24. Please provide the stability experiments at the current density of 1.14 A cm⁻².

Reviewer #2

(Remarks to the Author)

This study presents an innovative membrane-free electrochemical system for chlorine (Cl₂) production, leveraging Bernoulli's principle to drive Cl₂ migration via a triple-phase gas diffusion layer. The work demonstrates impressive Faradaic efficiencies (up to 96.3%) and stability, with potential cost savings over conventional chlor-alkali processes. While the concept is compelling and the results are promising, several key issues require clarification to strengthen the manuscript's rigor and impact.

(1) Given the harsh electrochemical conditions (high current densities, extreme pH), surface reconstruction or phase changes of the MOFs are inevitable. The authors should address this critical aspect more thoroughly.

(2) If the Ti-MOF catalyst undergoes surface reconstruction under operational conditions, the initial structure used for calculations may not reflect the true active phase.

(3) The authors should include the k-space of the XAFS data in the supplementary information. In addition, an R-space range of 1 to 6 Å is sufficient to capture the main coordination shells of the materials. The EXAFS fitting results in Supplementary Table 1 should include standard deviations or confidence intervals for each fitted parameter.

(4) While wavelet transform analysis can provide additional insights into the EXAFS data, its utility in this context is limited. The structural information obtained from WT may not be highly accurate. Given this, the authors should either provide a more detailed justification for including WT analysis or consider omitting it if it does not significantly contribute to the understanding of the catalyst's structure.

(5) The authors should include photographs of the internal construction of the prototype device. In addition, a critical aspect that remains unclear is how this membrane-free device ensures water does not leak into the adjacent gas chamber.

Reviewer #3

(Remarks to the Author)

The authors have reported a chlorine evolution system based on Bernoulli principle, which promotes oriented migration of Cl₂ away from reaction interfaces. This has not only prevented H₂/Cl₂ crossover but also circumvented the Pourbaix diagram problems, leading to significantly enhanced activities. By further joining with a pH-tolerant catalyst, a membrane-free prototype device has been built for high-rate Cl₂ production. Further, the strategy has been demonstrated for solving key challenging problems in other chemical/catalytic reactions like water splitting (2H₂O → 2H₂ + O₂). Overall, the strategy is interesting, and the manuscript is well organized. It can be published in Nature Communications after addressing the following concerns.

1. In Figure 3c, the authors have tested Cl₂ evolution in membrane-free prototype device using CO₂ and air carrier gases. However, only the Faradic efficiency of air carrier has been provided. Please provide the relevant data for CO₂ carrier gas.

2. The author discussed the activities of Cl₂ electrosynthesis on gas diffusion electrode-supported catalysts. However, the performances of gas diffusion electrode substrate have not been provided.

3. In terms of mechanism (Figures 5), why did the authors choose *Cl and *OCl as reaction intermediates? What is the reason of choosing these two intermediates for DFT calculates?

4. Some experimental details need to be provided, such as the optical picture of the synthetic and electrochemical testing processes.

5. For DFT calculations (Figures 5f), the author should provide more details on building the slab models. Why they use the cluster model for theoretical calculations?

6. The authors have claimed Cl₂ migration to gas chamber by pressure difference according to Bernoulli's principle. However, the description is too simple. More in-depth discussion should be given, both from theory and experiments prospect.

7. Some typos and errors have been noted, particularly in the Reference section and the fourth paragraph of Introduction section. A careful recheck of the whole manuscript is recommended.

Reviewer #4

(Remarks to the Author)

In this manuscript, the authors reported a membrane-free electrochemical system for Cl₂ synthesis with a titanium-based metal-organic framework (Ti-MOF) anode. Under the guidance of Bernoulli's principle, Cl₂ could permeate through the gas diffusion electrode and be directly separated from the anode by the carrier gas on the other side during the reaction. Although the Cl₂ separation strategy is of significance, the manuscript lacks sufficient evidences to support the conclusions. Meanwhile, the comparison of the activity data was not scientific. Therefore, this manuscript is not recommended for publication in Nature Communications.

Specific comments are as follows.

1. The author's identification of the signal at 1.92 Å as the Ti-O scattering path was unconvincing without reference samples with Ti-O and Ti-C scattering pathways (such as TiO₂, TiC₂, etc.). In addition, the authors are required to clarify why the Ti-Ti coordination in Ti-MOF (2.89 Å) was significantly smaller than that in the reference Ti foil (3.11 Å).
2. Line 90, the authors mentioned the well-defined crystallinity of Ti-MOF nanosheet, which was inconsistent with an amorphous structure revealed by the TEM image in Figure 2a. Moreover, the authors are suggested to index the sharp diffraction peaks (25-30 degree) in the XRD pattern (Figure 2b).
3. The FTIR identification of organic ligands in Figure 2c should be confirmed by reference samples.
4. Line 108-113, Page 6, the final structure of Ti-MOF did not align with the provided evidence. The coordination number of Ti-O was 3.183, inconsistent with the metal-oxygen-layers of a hexacoordinated TiO₆ units.
5. The authors were required to provide the complete calculation process of the pressure difference at the triple-phase interface in the main text, as well as the revised triple-phase Bernoulli formula.
6. The applied potentials for all the activity experiments in the three-electrode system were referenced against the Ag/AgCl and had not been converted to the potentials relative to the more commonly used reversible hydrogen electrode (RHE). Therefore, it was unreasonable to investigate the effect of pH variation on activity at the same potential (vs. Ag/AgCl), as the potential (vs. RHE) would change with pH under these conditions.
7. The authors compared the Cl₂ yield rate of Ti-MOF with that of the most recently reported catalysts. However, when achieving nearly similar Cl₂ generation rates, Ti-MOF required a higher current density and exhibited a lower Faraday efficiency, which did not meet the claimed "record activities". Note that the overpotential at 10 mA cm⁻² or 100 mA cm⁻² is a common descriptor for chlorine evolution activity. The authors should provide the activity data of the electrocatalysts in the text and conduct a systematic comparison with the state-of-the-art materials, especially under different pH conditions.
8. The authors claimed to observe Raman signals of Ti-*OCl in the operando Raman spectra, but the signal-to-noise ratio of the provided Raman spectra is too low to substantiate their conclusions.
9. The model constructed in DFT calculations demonstrated the atomically dispersed Ti atoms. The authors need to further clarify whether Ti-MOF was a single-atom catalyst, as there was no relevant evidence or description in the main text. Furthermore, the author constructed a cluster model terminated by sodium ions without considering periodic boundary conditions, which seems to be an unreasonable approach, as both the XRD and HRTEM results demonstrated the periodic crystalline structure of Ti-MOF.
10. It was meaningless for the authors to merely compare whether the presence of a membrane affected the production costs of NaOH and Cl₂ with Ti-MOF as the anode. The authors should compare the costs of the products in a membrane-free system with Ti-MOF as the anode versus a traditional membrane system with DSA as the anode.
11. Is pH-universal Cl₂ electrosynthesis attributed to the use of Ti-MOF or the strategy of chlorine gas separation?
12. It is still quite confusing whether the pH tolerance is brought by the characteristic of Ti-MOF or the pressure difference (namely the three-phase set up). Obviously, the DSA exhibits very poor CER performance at pH=7 and 13 in a traditional three-electrode set up. However, when testing the CER activity of Ti-MOF, the authors tuned the pressure difference which is not performed for the scenario of DSA. The authors should provide the experimental conditions clearly in the Method Section/Figure Illustration. Furthermore, if Ti-MOF is the main contributor for the pH tolerance, what property brings this benefit?
13. The authors are suggested to provide solid data source during technical-economic evaluations. Important parameters include the price of electricity, NaCl, water, membrane, etc. As the American manufacturing industry is mentioned (Page 7, Supporting Information), we hypothesize this Cl₂ production factory to be set up in the U.S. Based on the data on Global-Petrol-Prices., the industrial electricity price in the U.S. is estimated at \$0.15/kWh in 2024, much higher than the \$0.03/kWh used in the authors' estimates.
14. The oxygen evolution reaction (OER) chemical equation (in Line 43, Page 3) was incorrect. Generally, OER involves a four-electron transfer rather than a six-electron transfer.
15. The term "Cl₂ yield rates" in Line 128-130, Page 7 was incorrect and should be replaced with "the Cl₂ Faradaic efficiencies".
16. There are many incorrect expressions and figure numbers in the main text. Such as "...the yield rate of Ti-MOF is 2.6 times the one of DSA..." in Line 161, Page 8, which should be change to "...the Cl₂ yield rate of Ti-MOF is 2.6 times the one of DSA...".

Reviewer #5

(Remarks to the Author)

In this article, the authors designed and prepared an efficient yet cost-effective electrochemical system for Cl₂ electrosynthesis. It modulates gas diffusion layer by Bernoulli's principle, wherein the pressure difference at triple-phase boundary drives oriented Cl₂ migration directly into gas chamber, thus preventing the crossover of anodic/cathodic products and reaching a pH-tolerant CER. This article provides excellent and comprehensive guidance on design and development of electrocatalytic chlorine evolution devices and effectively reduces industrial costs. However, there are still many problems in terms of structural characterization and mechanism exploration. In a whole, this work cannot be accepted for becoming suitable for this journal.

Q1: The author points out that "a Ti-MOF nanosheet reveals its well-defined crystallinity with the adjacent lattice spacing of 0.2 nm, which agrees well with XRD containing a series of characteristic peaks corresponding to metal-organic framework architectures with dominant (210) crystal face" (Line 90-92). However, in fact, the so-called lattice spacing were not observed in Figure. 2a, and XRD (Figure. 2b) can only correspond to the three peaks before 20°, but basically did not match peaks after 20°. Therefore, a Fourier-transform TEM image or SAED image should be exhibited to support this view. At the same time, XRD needed to be verified for correctness.

Q2: The data from XAS in Figure. 2d-e and Supplementary Table 1 contradicts the author's description of catalysts structure in the article. The author points out that "The Ti node is octahedrally coordinated" (Line 111-112). But the EXAFS-fitting indicated a CN of 3.183 for Ti-O (Line 103 in Supplementary Table 1). It's necessary to clarify the error values towards each parameter in Supplementary Table 1 and explain this contradictory phenomenon.

Q3: The author used operando Raman spectroscopy to verify the intermediates in the reaction process and observed potential-dependent *OCl signals, but in reality, no corresponding signal was observed at 1077 cm⁻¹ in either Figure 5a or Supplementary Figure 41a-b.

Q4: Could the authors clearly describe the correlation of Bernoulli principle in its reaction process? It seems that there is no connection between them, and the mechanism study part has not been well reflected.

Version 1:

Reviewer comments:

Reviewer #1

(Remarks to the Author)

Most of my concerns have been well addressed in this revision. Still, there are two issues to be solved.

1. The influence of pH on the solubility of Cl₂ should be considered. An environment with low pH (acidic) favors the existence of Cl₂ in its molecular form (where physical dissolution dominates), while a high pH (alkaline) environment promotes chemical reactions of Cl₂, significantly increasing its apparent total solubility.

2. Why is the difference between Ti-MOF-membrane and Ti-MOF-membrane-free in Figure 3d so negligible? The performance disparity between the membraned and membrane-free configurations appears marginal and requires further clarification.

In Fig. 3f, the colors corresponding to the four catalysts are indistinguishable. Please recreate this figure with clearer and more distinct visual differentiation.

Reviewer #2

(Remarks to the Author)

The authors have comprehensively addressed all of my previous concerns. The revisions are satisfactory, and the manuscript now presents a complete and valuable piece of work that meets the high standards of Nature Communications.

Reviewer #3

(Remarks to the Author)

The revised version can be accepted.

Reviewer #4

(Remarks to the Author)

The authors have addressed most of my concerns, but there are still some minor issues should be solved before publication.

1. The authors attributed the lower coordination number of Ti-O (3.36) in Ti-MOF compared to that of the theoretical TiO₆ units to the difference in contributions from O at various positions (i.e., internal O contributes 1, while surface O contributes 0.5, Figure R2). They should also consider another possibility that the lower coordination number may be related to the presence of abundant oxygen vacancies in the Ti-O framework. Considering the fact of the lower coordination number, the authors are suggested to revise the theoretical model to better reflect the actual situation.

2. Were the performance tests of DSA and Ti-MOF in Figure 3d and e conducted in the same prototype device shown in Figure 3b? Was there a pressure difference in both cases? Why did DSA exhibit such poor chlorine production performance (with a Faraday efficiency for chlorine gas below 10%)?

3. The authors should assign the vibrational peaks of H₂NDC at 1668 and 1415 cm⁻¹ and of 2,6 naphthoic 2Na at 786 cm⁻¹.

4. The author needs to provide the formulas used for calculating the economic costs, rather than only presenting the final calculation results.

5. Why the current densities used in the three tables, namely Supplementary Table 9-11: Ti-MOF-membrane -free for CER, Ti-MOF-membrane for CER, and DSA-membrane for CER, were different? Would the difference in current density affect the calculation of economic analysis?

Reviewer #5

(Remarks to the Author)

The data presentation in the manuscript has several shortcomings that need to be addressed before the results can be considered reliable:

1) The description of Cl₂ measurement is not clear or sufficiently detailed. Critical information such as reagents details including companies, purity, and prepared concentration, full titration procedure, and standardization of the Na₂S₂O₃ solution is missing. Without these details, the reproducibility and reliability of the results cannot be assessed.

2) The KI–thiosulfate titration method is known to introduce significant errors, especially due to interferences and endpoint subjectivity. The manuscript does not provide any indication of experimental errors, replicate measurements, or data variability. The use of the FAS-DPD titration method would have been more suitable for this study, as it is widely regarded as more accurate and specific for chlorine analysis.

3) There are several well-established, high-accuracy techniques for chlorine quantification (e.g., amperometric titration, ion chromatography, or membrane-covered amperometric analyzers). The manuscript should validate the reported chlorine yields with at least one of these methods to strengthen confidence in the results.

4) The description of GC analysis for H₂, O₂, and N₂ is incomplete. Essential details such as the instrument type, detector used, column specifications, calibration procedures, and standards are not reported. Furthermore, the manuscript does not include calculations or data for the molar amounts or yields of each gas, which are crucial for evaluating the performance of the system.

Version 2:

Reviewer comments:

Reviewer #1

(Remarks to the Author)

All my concerns have been well addressed in this revision. Congratulations!

Reviewer #4

(Remarks to the Author)

The authors have addressed all my concerns. I recommend its publication at this version.

Reviewer #5

(Remarks to the Author)

The comments are addressed. Suggest to accept for publication

List

Response to Reviewer #1	Page 2
Response to Reviewer #2	Page 37
Response to Reviewer #3.....	Page 53
Response to Reviewer #4	Page 68
Response to Reviewer #5	Page 112

We would like to thank the Editor's invitation to revise the manuscript. We are also appreciated of the reviewer #1 for his/her constructive comments to improve the quality of this work.

After receiving the reviewer's comments, we have spent ten intensive weeks in order to answer the concerns raised by the astute reviewer. We believe we have shifted the work to the next level of proof and gained a much deeper understanding of the system. Please see the following point-by-point response to the reviewer #1's comments. Thank you and best wishes,

Kind regards

The authors

Response to Reviewer #1

Original Comment: *This manuscript by Nie et al. present a breakthrough in chlorine (Cl₂) electrosynthesis by integrating Bernoulli's principle with a pH-tolerant Ti-MOF catalyst. The novel triple-phase gas diffusion electrode design enables oriented Cl₂ migration via pressure differences, eliminating membrane requirements and preventing H₂/Cl₂ crossover. The manuscript is well-organized, clearly presented, and the results are convincing. However, there are still some questions need to be illustrated clearly. Also, the conclusion needs to be further revised under solid evidence. Below are specific critiques to address these issues.*

Original Comment 1. *The abstract states Faradaic efficiencies of 96.3%~87.6% within 0.1~1.14 A cm⁻², yet fails to clarity why the efficiency decreases with increasing current density.*

Response: Thanks for your kind reminding. Such a phenomenon is known as “activity-selectivity trade-off” relationship, which has been widely observed in many previous studies (like Nature Communications 2020, 11, 412; Science 2023, 380, 727).

In our system, the membrane-free chlor-alkali process involves anodic chlorine evolution reaction (CER) coupled with cathodic hydrogen evolution (HER), which is complicated by side oxygen evolution reaction (OER) as follows:

With the current densities increasing from 0.1 to 1.14 A cm⁻², the applied voltage rises from 1.8 to 2.4 V. This has intensified side reaction (such as OER), which consume electrons otherwise for CER, thereby reducing the Faradaic efficiency for Cl₂ production.

To provide evidence for above hypothesis, we have conducted additional experiment of determining O₂ byproduct from side OER. At different applied potentials, O₂ was collected and measured by gas chromatography (GC, Supplementary Figure 49). We have annotated the GC peak areas of O₂ as 40.36 at 1.8 V, which has doubled at 2.4 V (*i.e.*, 88.8), respectively. This result confirms an increase in O₂ levels with rising applied potentials, thereby reducing overall efficiency of Cl₂ production.

Based on above results, the following figure/sentence has been added/ revised in updated manuscript:

Supplementary Figure 49. GC data of O₂ under 1.8V and 2.4V conditions.

“Page 9, Line 193-194: In spite of “activity-selectivity trade-off” relationship from competitive OER (Supplementary Figure 49) ...”

“Supplementary information, Line 484-488: We have conducted additional experiment of determining O₂ byproduct from side OER. At different applied potentials, O₂ was collected and measured by gas chromatography (GC, Supplementary Figure 49). We have annotated the GC peak areas of O₂ as 40.36 at 1.8 V, which has doubled at 2.4 V (*i.e.*, 88.8), respectively. This result confirms an increase in O₂ levels with rising applied potentials, thereby reducing overall efficiency of Cl₂ production.”

Original Comment 2. *The working potential of CER is 1.8 V-2.4 V in this manuscript, which is relatively high compared to other literatures. Is this system really energy-efficient?*

Response: Thanks for your useful comment. Generally, the energy efficiency of an electrochemical system is calculated according to the following equations.^{Adv Mater 2016, 28, 3423}

$$\varepsilon_{\text{energy}} = \frac{E_{\text{eq}}}{E_{\text{eq}} + \eta} \times \varepsilon_{\text{Faradaic}}$$

where E_{eq} is the equilibrium potential and η refers to the overpotential. Based on this equation, one can see that the high energy efficiency arises from the combination of both a low overpotential and a high Faradaic efficiency.

In our system, the Faradaic efficiencies locate in the range of 96.3%~87.6%, so energy efficiency is largely determined by overpotentials. In Figures 2,3 of the original manuscript, the working potential of CER has been presented versus Ag/AgCl reference electrode, which is difficult to determine overpotentials (η). For reliable comparison, we have conducted linear sweep voltammetry (LSV), and plotted these data versus RHE (Supplementary Figures 14,32). We have further determined the overpotentials at different pHs (148, 156 and 404 mV for pH=1, 7 and 13) according to LSV. *ChemElectroChem* 2020, 7, 1448; *Nature* 2023, 617, 519 When compared with literature (Supplementary Table 3), our system demonstrates competitive overpotentials among all non-noble metals which is even comparable to some noble metal benchmarks. This result demonstrated our system is energy-efficient.

Based on above results, the following figure/sentence/table has been added into updated manuscript:

Supplementary Figure 32. Electrochemical performance data of Ti-MOF by potential conversion to reversible hydrogen electrode (RHE). a-c, LSV. d-f, Faradaic efficiency and gas production rate.

Supplementary Table 3. The comparison of the overpotential (10 mA cm⁻²) of CER.

Electrocatalysts	Overpotential (mV)	Reaction conditions	References
RuO _x /2D TiO _x	90	1 M pH=1	29
Ru-O ₄	30	1 M pH=1	30
CoO _x Cl _y	100	0.5 M pH=2	31
RCON-H	89 (1 A cm ⁻²)	5 M pH=2	11
MoO _x @IrO ₂ -Ta ₂ O ₅	30	4 M pH=2	32
Pt ₁ /p-NC@CNTs	40	1M 0.1 HClO ₄	33
Ir ₁ O ₆	71.8	4 M pH=2	7
RuO ₂ /TiO ₂	220	4 M pH=3	34
Ni(Co/Mn)Sb ₂ O _x	450	4 M pH=2	24
Co ₃ O ₄ /FTO	200	Saturated pH=3	35
Ti _{0.35} V _{0.35} Sn _{0.25} Sb _{0.05} -oxide	987	5 M NaCl pH≈2	25
Ti-MOF	148	5 M NaCl pH=1	This work
	156	5 M NaCl pH=7	
	404	5 M NaCl pH=13	

“Page 7, Line 135-136: To facilitate comparison across diverse pH conditions, all of the potentials have been converted to RHE (Supplementary Figure 32 and Supplementary Table 3).”

Ref 7. Wang, J. et al. Engineering the Coordination Environment of Ir Single Atoms with Surface Titanium Oxide Amorphization for Superior Chlorine Evolution Reaction. *J. Am. Chem. Soc.* **146**, 11152–11163 (2024).

Ref 24. Moreno-Hernandez, I. A., Brunschwig, B. S., Lewis, N. S. Crystalline nickel, cobalt, and manganese antimonates as electrocatalysts for the chlorine evolution reaction. *Energy Environ. Sci.* **12**, 1241–1248 (2019).

Ref 25. Mirzaei Alavijeh, M. et al. A selective and efficient precious metal-free electrocatalyst for chlorine evolution reaction: An experimental and computational study. *Chem. Eng. J.* **421**, 127785 (2021).

Ref 29. Ji, J. et al. Ruthenium Oxide Clusters Immobilized in Cationic Vacancies of 2D Titanium Oxide for Chlorine Evolution Reaction. *Small Struct.* **5**, 2300240 (2023).

Ref 30. Liu, Y. et al. Electrosynthesis of chlorine from seawater-like solution through single-atom catalysts. *Nat. Commun.* **14**, 2475 (2023).

Ref 31. Xiao, M. et al. Self-adaptive amorphous CoO_xCl_y electrocatalyst for sustainable chlorine evolution in acidic brine. *Nat. Commun.* **14**, 5356 (2023).

Ref 32. Zhan, Y. et al. Efficient electrocatalytic chlorine evolution of MoO_x modified $\text{IrO}_2\text{-Ta}_2\text{O}_5$ and its application in the seawater electrolysis. *J. Electroanal. Chem.* **992**, 119283 (2025).

Ref 33. Quan, L. et al. Atomic Pt-N₄ Sites in Porous N-Doped Nanocarbons for Enhanced On-Site Chlorination Coupled with H₂ Evolution in Acidic Water. *Adv. Funct. Mater.* **33**, 2307643 (2023).

Ref 34. Menzel, N., Ortel, E., Mette, K., Kraehnert, R., Strasser, P. Dimensionally Stable Ru/Ir/TiO₂-Anodes with Tailored Mesoporosity for Efficient Electrochemical Chlorine Evolution. *ACS Catal.* **3**, 1324–1333 (2013).

Ref 35. Zhu, X. et al. Co₃O₄ nanobelt arrays assembled with ultrathin nanosheets as highly efficient and stable electrocatalysts for the chlorine evolution reaction. *J. Mater. Chem. A* **6**, 12718–12723 (2018).

Original Comment 3. *While Bernoulli's principle explains Cl₂ migration via carrier gas, the separation mechanism for Cl₂ and carrier gas is unaddressed. Does the mixture of carrier gas with Cl₂ cause Cl₂ to be impure?*

Response: Thanks for your kind suggestion. Yes, Cl₂ and carrier gas (like CO₂) has together evolved out from the gas chamber of our system, which requires further post separation. Yet, because of their different chemical and physical properties, the separation of Cl₂ and carrier gas is a well-developed and low-cost process. For example, Li and co-workers has demonstrated the facile separation of Cl₂

and CO₂ according to their density differences and solubility disparities in NaClO solution, achieving high-purity Cl₂ (as shown in the following Figure R1).^{Nature 2023, 617, 519}

[FIGURE REDACTED]

Figure R1. The **Figure S1** of the literature *Nature* 2023, 617, 519. The details of the separation of CO₂ and Cl₂ are elaborated in the text, providing a basis for the separation of gases in this paper.

Original Comment 4. *Grammar mistake: please revise recurrent misuse of “despite of” to “in spite of” or “despite”.*

Response: Thanks for your kind reminding. The grammar mistake has been corrected in updated manuscript as follows:

“Page 3, Line 33: **In spite of** their improved productivities...”

“Page 4, Line 49-50: **In spite of** several preliminary studies separating anodic/cathodic products by redox mediators (like Hg/NaHg¹⁴ and Na_{0.44}MnO₂⁶) ...”

“Page 9, Line 196-197: **In spite of** membrane-involving system achieving comparable activities under the same testing condition...”

“Page 10, Line 221: **In spite of** both membrane-free/-based systems show similar payback duration within four years...”

Original Comment 5. Please correct the anodic CER equation in introduction: The electron transfer number should be 2 not 1.

Response: Thanks for your kind reminding. The mistake has been corrected in introduction as follows:

“Page 3, Line 41: Anodic CER: $2\text{Cl}^- - 2\text{e}^- \rightarrow \text{Cl}_2$, $E^0=1.36 \text{ V vs. RHE}$ (1) ...”

Original Comment 6. Replace “mPa” into “MPa” throughout the manuscript.

Response: We are sorry for making the confusion. We have replaced all “mPa” with “ 10^{-3} Pa ” as follows:

Figure 3. Prototype device for Cl₂ electroynthesis. c, The Cl₂ Faradaic efficiencies change with gas flow rates.

“Page 4, Line 62: ...substantial pressure difference can be achieved (0~116.3 $\times 10^{-3} \text{ Pa}$).”

“Page 7, Line 123-124: ...which generates a pressure difference of 116.3 $\times 10^{-3} \text{ Pa}$ at the three-phase boundary according to the Bernoulli's equation.”

“Page 8, Line 166-167: ...low rates of gas flow only result in the small pressure difference (0~7.26 $\times 10^{-3} \text{ Pa}$).”

“Page 8, Line 169-170: ...the maximum value (75.8~82.6% and yield rate: 1.02~1.08 $\text{mmol h}^{-1} \text{ cm}^{-2}$) in the range of 40~80 mL min^{-1} with moderate pressure difference (16.3~116.3 $\times 10^{-3} \text{ Pa}$).”

“Page 11, Line 241: At the pressure difference of 116.3 $\times 10^{-3} \text{ Pa}$...”

“Page 12, Line 255: Figure 5c shows the pressure distribution map with pressure difference of 29 10^{-3} Pa ...”

Original Comment 7. Grammar mistake in introduction: subject-verb agreement error of “circumvents” → “circumvent”.

Response: Thanks for your kind reminding. The grammar mistake has been corrected in updated manuscript as follows:

“Page 4, Line 63-64: ...which can not only prevent H₂/Cl₂ crossover but also **circumvent** the Pourbaix diagram problems.”

Original Comment 8. In figure 2a, lattice spacing of 0.2 nm is not clear, which should be redone to better prove the (210) crystal face.

Response: Following the reviewer’s kind suggestion, we have redone the TEM characterization, and further determine lattice spacing as 0.21 nm, corresponding to (210) crystal face of Ti-MOF catalyst.

Based on above results, the following figures and sentences have been added/revised in updated manuscript:

Figure 2. Catalyst and system design for Cl₂ electrosynthesis beyond Pourbaix diagram. a, TEM and enlarge the image of Ti-MOF catalyst. The blue wavy line form is the lattice spacing after FFT.

Supplementary Figure 3. The dislocation analyses for measuring crystal plane spacing of Ti-MOF.

“Page 5, Line 87-89: Close examination of a Ti-MOF nanosheet reveals its well-defined crystallinity with the adjacent lattice spacing of 0.21 nm (Figure 2a, Supplementary Figure 3) ...”

“Supplementary information, Line 173-178: In the main text, Figure 2a (inset) demonstrates clear lattice fringe with interplanar spacing of 0.21 nm. This result is consistent to Supplementary Figure 3, which validates the crystallinity through Fast Fourier Transform (FFT)-processed dislocation analyses and quantified interplanar spacing measurements. More specifically, the lattice region within red box in Figure 2a (main text) was FFT-processed and measured across six consecutive intervals (total distance: 1.2748 nm), yielding a mean spacing of 0.21 nm.”

Original Comment 9. *Why in Figure 2e there exist Ti-Ti and Ti-O, but in Figure 2f authors only mention Ti-O?*

Response: We are sorry for not explaining it clearly. Following the reviewer’s kind suggestion, we have further examined Ti-Ti interaction of Ti-MOF in EXAFS profile, where the Ti-Ti distance (3.12 Å) corresponds to interlayer interaction between Ti atoms in the form of multi-nuclear titanium clusters. *ACS Mater Lett* 2021, 3, 64

Based on above results, the following sentence and reference have been added into updated manuscript:

“Page 6, Line 101-103: The Ti-Ti distance (3.12 Å) corresponds to interlayer interaction between Ti atoms in the form of multi-nuclear titanium clusters.³¹

Ref 31. Sun, Y. et al. Large Titanium-Oxo Clusters as Precursors to Synthesize the Single Crystals of Ti-MOFs. *ACS Mater. Lett.* 3, 64-68 (2020).

Original Comment 10. *Why the coordination number of Ti-O in Ti-MOF is 3.183, but authors speculate Ti node is octahedrally coordinated? What’s the relationship between the two?*

Response: We are sorry for not explaining it clearly. “Ti node is octahedrally coordinated” means one Ti atom can theoretically coordinated with six other atoms in the form of octahedra architecture, which represents the global symmetry constraints for Ti atoms. On the other hand, the average coordination number of Ti-O in Ti-MOF has been experimentally determined by X-ray absorption near-side structure (XANES, Figures 2d, e, Supplementary Figures 7–8), where the average number

is 3.98 (before low-temperature calcination) or 3.36 (after calcination). To illustrate the relationship between the two, we have built a structural model for Ti-MOF as follows (Figure R2):

As shown in Figure R2, the spatial configuration within the Ti-O cluster is in the form of octahedra containing of 16 Ti atoms and 96 O atoms with a ratio of 1/6 (Figure R2). However, when counting the actual coordination number of Ti-O for XANES, the contribution of each atom has to be taken into account, for example, the coordination of O atoms would be considered according to their local environment, which is 1 (internal O atom) or 0.5 (surface O atom). We can see that 48 O atoms are internal and 48 O atoms are on the surface. Accordingly, the number of O atoms coordinated to Ti is calculated as: $48 + 48 \times 0.5 = 72$, resulting in a Ti/O ratio of 1:4. This result is close to our experimentally XANES measured value (3.98).

Further, to improve its crystallinity, the Ti-MOF has been subject to calcination at 450 °C for 3 hrs. Such a calcination treatment has not altered Ti-MOF architecture, which is still in the form of octahedra, as confirmed by TEM (Figure 2a), XRD (Figure 2b), FT-IR (Figure 2c). While X-ray absorption near-side structure (XANES) confirms the coordination number of Ti-O has decreased to 3.36 (Supplementary Table 2). This phenomenon indicates the evaporation of some organic ligands during calcination process, which can expose more Ti active sites for electrochemical reaction.

Based on above results, the following figure/sentence has been added/revised in updated manuscript:

“Page 6, Line 111-117: Ti atom can theoretically coordinate with six other atoms in the form of hexacoordinated architecture, which represents the global symmetry constraints for Ti atoms. On the other hand, the coordination number of Ti-O in Ti-MOF has been experimentally determined by XAS (Figures 2d, e, Supplementary Figures 7–8), where the average number is 3.98 (before low-temperature calcination) or 3.36 (final structure after calcination). This phenomenon indicates the evaporation of some organic ligands during calcination process, which can expose more Ti active sites for electrochemical reaction.”

Figure R2. The periodic structure of Ti-MOF-before.

Supplementary Table 2. Bond length and coordination of Ti-MOF-before and after calcination

Material	shell	CN	R(Å)	ΔE_0 (eV)	σ^2 (10^{-3}Å^2)	R factor
Ti-foil	Ti-Ti	6.24±0.21	2.89±0.002	-7.05±0.18	0.006±0.0005	0.0084±0.00224
TiO ₂	Ti-C	4.15±0.05	1.95±0.003	-1.55±0.26	0.006±0.0001	0.014±0.0012
	Ti-Ti	2.05±0.05	3.06±0.0004		0.0040±0.00025	
TiC	Ti-O	6.03±0.08	2.14±0.00039	-3.47±0.01	0.0002±0.0001	0.0095±0.00035
	Ti-Ti	11.93±0.14	3.05±0.0002		0.0009±0.0001	
Ti-MOF (before)	Ti-O	3.98±0.04	1.90±0.0005	-7.04±0.10	0.016±0.0001	0.0034±4.1×10 ⁻³
	Ti-Ti	4.05±0.002	3.10±10 ⁻⁷		0.019±1.5×10 ⁻⁵	
Ti-MOF	Ti-O	3.36±0.11	1.92±0.002	-4.21±0.21	0.011±0.0009	0.013±0.0043
	Ti-Ti	4.05±0.09	3.12±0.00008		0.018±0.00064	

Original Comment 11. *In Figure 3b, gas chromatography doesn't detect H₂. Where does the H₂ go? Is H₂ be collected?*

Response: We are sorry for explaining it clearly. Figure 3b only show the gas chromatography (GC) profile of anodic products (*i.e.*, Cl₂ and carrier gas), while H₂ is produced from the cathodic part. Therefore, the absence of H₂ in Figure 3b provides evidence of no gas crossover in our membrane-free system.

To further determine H₂, we have collected the cathodic products, and characterized by gas chromatography (GC). As shown in Supplementary Figure 45, the H₂ signal can be detected by GC.

Supplementary Figure 45. Gas chromatography (GC) profile of cathodic products.

“Page 8, Line 158-160: Finally, The H₂ generated at the cathode is collected and measured by gas chromatography, gas-phase results (Supplementary Figure 45) demonstrate that H₂ is discharged from the reaction system along with the electrolyte.”

Original Comment 12. *The plateau in Cl₂ Faradic efficiency beyond 40 mL min⁻¹ gas flow is not explained mechanistically in Figure 3c. Please clarify whether this stems from gas-liquid equilibrium constraints or kinetic bottlenecks.*

Response: We are sorry for not explaining it clearly. We consider this phenomenon stemming from kinetic bottlenecks, *i.e.*, the competition between Cl₂ dissolution in aqueous electrolyte (driven by gas-liquid equilibrium) and Cl₂ migration to gas chamber (driven by Bernoulli's principle). We have given the response as follows:

Firstly, Cl₂ is a soluble gas in water (0.79 g per 100 g of water), and according to Pourbaix diagram (Figure 1c), rapidly hydrolyzes to produce hypochlorous acid (HClO) and hypochlorous acid which then dissociates into hypochlorite ion (OCl⁻) and hydrogen ion (H⁺):

We have calculated the mass of Cl₂ theoretically produced per second with Faradaic efficiency of Cl₂ generation assumed to be 100% at 100 mA cm⁻²:

$$m = \frac{I \times T \times M}{n \times F}$$

where m represents the mass of Cl₂, I is the current density, T is time, M is the molar mass of Cl₂ (71 g mol⁻¹), n is the number of transferred electrons, and F is the Faraday constant (96485 C mol⁻¹).

Based on above formula, it is estimated Cl₂ generation rate as 0.0000368 g cm⁻² s⁻¹. By comparison with the Cl₂ solubility in water (0.79 g per 100 g of water), most of the produced Cl₂ would be dissolved and dissociated into OCl⁻/Cl⁻ couple driven by gas-liquid equilibrium.

On the other hand, we have calculated the driven force of Cl₂ migration to gas chamber according to Bernoulli's principle (Supplementary Table 4). By calibration into electrode area, the generate pushing weight of Cl₂ is in the range of 0~0.16 mg cm⁻² for 0~30 mL min⁻¹, and 0.28~1.14 mg cm⁻² for 40~80 mL min⁻¹. By dividing Cl₂ theoretical productivity (0.0000368 g cm⁻² s⁻¹), it can be concluded that the at least seven folds of Cl₂ migration driven force can overcome the Cl₂ dissolution in aqueous electrolyte driven by gas-liquid equilibrium.

Based on above results, the following table/sentence has been added/revised in updated manuscript:

Supplementary Table 4. The pressure difference generated by air at different flow rates as carrier gas and the magnitude of the driving force produced by this pressure difference.

Air flow velocity/mL min ⁻¹	Pressure/mPa	Generate pushing weight/mg cm ⁻²
0	0	0
10	1.817141784	0.017807989

20	7.268567137	0.071231958
30	16.35427606	0.160271905
40	29.07426855	0.284927832
60	65.41710423	0.641087621
80	116.2970742	1.139711327

“Supplementary information, Line 733-754:

In Figure 3c, the plateau in Cl₂ Faradaic efficiency beyond 40 mL min⁻¹ gas flow stems from kinetic bottlenecks, *i.e.*, the competition between Cl₂ dissolution in aqueous electrolyte (driven by gas-liquid equilibrium) and Cl₂ migration to gas chamber (driven by Bernoulli’s principle).

Firstly, Cl₂ is a soluble gas in water (0.79 g per 100 g of water), and according to Pourbaix diagram (Figure 1c), rapidly hydrolyzes to produce hypochlorous acid (HClO) and hypochlorous acid which then dissociates into hypochlorite ion (OCl⁻) and hydrogen ion (H⁺):

We have calculated the mass of Cl₂ theoretically produced per second with Faradaic efficiency of Cl₂ generation assumed to be 100% at 100 mA cm⁻²:

$$m = \frac{I \times T \times M}{n \times F}$$

where m represents the mass of Cl₂, I is the current density, T is time, M is the molar mass of Cl₂ (71 g mol⁻¹), n is the number of transferred electrons, and F is the Faraday constant (96485 C mol⁻¹). Based on above formula, it is estimated Cl₂ generation rate as 0.0000368 g cm⁻² s⁻¹. By comparison with the Cl₂ solubility in water (0.79 g per 100 g of water), most of the produced Cl₂ would be dissolved and dissociated into OCl⁻/Cl⁻ couple driven by gas-liquid equilibrium.

On the other hand, we have calculated the driven force of Cl₂ migration to gas chamber according to Bernoulli’s principle (Supplementary Table 4). By calibration into electrode area, the generate pushing weight of Cl₂ is in the range of 0~0.16 mg cm⁻² for 0~30 mL min⁻¹, and 0.28~1.14 mg cm⁻² for 40~80 mL min⁻¹. By dividing Cl₂ theoretical productivity (0.0000368 g cm⁻² s⁻¹), it can be concluded that the at least seven folds of Cl₂ migration driven force can overcome the Cl₂ dissolution in aqueous electrolyte driven by gas-liquid equilibrium.”

Original Comment 13. In Figure 3d-e, the control experiment of DSA-membrane should be added.

Response: Following the reviewer’s comment, we have added the control experiment of DSA-membrane as follows:

Figure 3. Prototype device for Cl₂ electroynthesis. c, The Cl₂ Faradaic efficiencies change with gas flow rates. d-f, The Cl₂ Faradaic efficiencies (FE), yield rates and other activity comparisons for Ti-MOF and DSA in membrane-based/-free devices.

“Page 9, Line 175-180: As showing in Figures 3d, 3e, the Cl₂ gas Faradaic efficiencies for Ti-MOF is 94% at the potential of 1.8 V, which is 18 times the one of DSA under the same test conditions and set-up. Analogously, the Cl₂ gas yield rate of Ti-MOF is 3.4 times the one of DSA at a potential of 1.8 V. Further with Ti-MOF, the performances with and without membranes have been compared, which shows minor alternations in Cl₂ Faradaic efficiencies (94% vs. 92%) and yield rates (0.93 vs. 0.9 mmol h⁻¹ cm⁻² Figures 3d, 3e and Supplementary Figure 47).”

Original Comment 14. In Fig. 3g, the “dissolved Cl₂ species” (brown) color scheme is inconsistent with the legend. Please replot color.

Response: Thanks for your kind reminding. We have rechecked color of dissolved Cl₂ species, which is “pompadour”. So, we have corrected the relevant descriptions in updated manuscript as follows:

“Page 9, Line 186-187: ...which can be divided into gaseous Cl₂ (blue) and the dissolved Cl₂ species (pompadour)...”

“Page 9, Line 188-189: ...with little dissolved Cl₂ detected (pompadour, Faradic efficiencies of 1.7% ~ 7.5%) ...”

“Page 27, Line 538-539: ...The arrows below the pompadour dots refer to flushing the electrodes every 20 hours.”

Original Comment 15. *The techno-economic analysis assumes industrial operation at 70 °C. Justify this temperature is along with actual chlor-alkali industry standards with literature citations.*

Response: Thanks for your useful comment. Yes, as pointed out by the reviewer, actual chlor-alkali process is temperature sensitive, and has been operated under elevated temperatures, where maintaining the right temperature is essential for maximizing product quality, ensuring worker safety, and optimizing energy consumption. This has been usually achieved by air- or water-cooling systems. The operation temperature can be up to 90 °C (as shown in Figure R3).^{Nature 2023, 617, 519} In this work, we have examined the influence of Cl₂ Faradaic efficiencies at different temperatures (55~90 °C, Supplementary Figure 50), and determine 70 °C as the optimized condition for system operation.

Based on above results, relevant figure/sentence has been revised/added into updated manuscript:

[FIGURE REDACTED]

Figure R3. **a**, The polarization curves of catalysts of NCOOH, NCOOH with *iR* correction, RCON-H and DSA with *iR* correction at 90 °C. **g** Stability testing of NCOOH under constant current and constant voltage conditions.

Supplementary Figure 50. Electrochemical data of Ti-MOF tested in 0.1 M NaOH+5 M NaCl.

a, CV curves at different temperatures under IR correct compensation at 1Ω. **b**, Cl₂ gas Faradic efficiencies and current density at different temperatures under IR correct compensation at 1Ω.

“Page 9, Line 192-193: ...the prototype device has worked under industrial operation conditions of 70 °C by collecting both gaseous Cl₂ and residue chlorine (denoted as active chlorine or AC).”

Ref 7. Yang, J. et al. CO₂-mediated organocatalytic chlorine evolution under industrial conditions. *Nature* 617, 519–523 (2023).

Original Comment 16. Why Cl₂ is counted as “cost” in Figure 4c?

Response: We are sorry for not explaining it clearly. In Figure 4c, the “cost” refers to the “production cost” of Cl₂ from our chlor-alkali system. We have revised all of the relevant sections in updated manuscript and supplementary information as follows:

“Page 3, Line 39: ...thus potentially delivering **production** cost savings (Figure 1a-b).”

“Page 4, Line 51-52: ...problems associated with these systems involve **production** costs...”

“Page 10, Line 210-211: ...while the use of membrane has accounted for additional **production** cost of **6.75% (1.17 million dollar per year)** in the membrane-based systems.”

Figure 4. Techno-economic analyses. **a**, The Faraday efficiency and current density of active chlorine production (Cl₂ gas and residue chlorine in the solution). **c**, The economic production cost of electrochemical production per kilogram of Cl₂ and NaOH for membrane-based and -free. **d-e**, Production cost breakdown for different components. **f**, Single factor analysis.

Supplementary Figure 53. Techno-economic analyses for Ti-MOF systems with/without membrane at 0.7 A cm^{-2} . a, FNPV analyses. b, Cost distribution percentages in an operating cycle. c-d, Production cost distribution in an operating cycle.

Supplementary Figure 54. Techno-economic analyses for Ti-MOF systems with/without membrane at 0.34 A cm^{-2} . a, FNPV analyses. b, Cost distribution percentages in an operating cycle. c-d, Production cost distribution in an operating cycle.

Supplementary Figure 55. Techno-economic analyses for Ti-MOF systems with/without membrane at 0.18 A cm^{-2} . a, FNPV analyses. b, Cost distribution percentages in an operating cycle. c-d, Production cost distribution in an operating cycle.

Original Comment 17. *In technical-economic evaluations section, the profits difference is described in Figures 4 g-h not Figures 5 g-h. The same goes for payback duration profiles.*

Response: Thanks for the kind reminding. We have rechecked the relevant figures and descriptions in manuscript, and corrected the typos/errors as follows:

“Page 10, Line 216-217: The profits difference with/without membrane has been intuitively described in **Figures 4g, h...**”

“Page 10, Line 220-221: The above result is consistent to the payback duration profiles (**Figure 4i**) ...”

Original Comment 18. *In Figure 5a, there seems no change in *OCl peak at all. Please reexamine experimental conditions.*

Response: Thanks for your kind reminding. We have re-tested the *operando* Raman data for five times and provide the responses as follows:

Generally, Raman spectra is a scattering signal with intrinsic weak intensity. When used to study the structural characteristics of a bulk material, obvious signal peaks can be seen. This is due to the structural crystal lattice that contributes to overall vibration of the bulk material, leading to enhanced scattered signals.

While in this work, the *operando* Raman is used to probe the transient surface-adsorbed species (*OCl intermediates) during electrocatalysis. The scattered Raman signals are mainly focused on bond vibrations of adsorbed species on catalyst surfaces, which are known to be very weak as comparison to structural crystal lattices in bulk materials. Consequently, the *operando* Raman signals for catalytic reactions often demonstrate characteristically low signal-to-noise ratios in the literatures (like ORR, NRR, CRR). Nat Commun 2024, 15, 4157; Angew Chem Int Ed 2025, 64, e202414202; J Am Chem Soc 2024, 146, 11152
Our *operando* Raman signals are comparable to the above literature.

To confirm the accuracy of the experimental results, we have further performed five *operando* Raman tests. We directly determined the positions and intensities of the peaks using Raman spectrometer software, and updated the *operando* Raman data with error bars (Figures 5a-b and Supplementary Figure S7). By comparing the Raman vibration at different electrolysis potentials, we conclude the characteristic *OCl Raman peak at approximately 1093 cm^{-1} (inside the blue transparent frame), and the intensity of *OCl peak changes consistently with applied potentials, reaching maximum intensity at 2.0 V vs Ag/AgCl (Figure 5b). Further, the data demonstrate significantly greater accumulation of *OCl intermediate without pressure difference. This observation suggests Bernoulli principle leveraging our reaction system that generates an immediate pressure gradient, enabling continuous removal of Cl_2 product and thereby enhancing chlorine evolution kinetics. All of these results provide the validity of our *operando* synthesis condition.

Based on above results, relevant figure/sentence has been revised/added into updated manuscript:

Figure 5. Mechanism study. **a**, The *operando* Raman spectra. **b**, Ti-*OCl Raman peak intensity.

Supplementary Figure 57. The *operando* Raman spectra without pressure difference. **a**, *operando* Raman spectra. **b**, Ti-*OCl Raman peak intensity.

“Page 11-12, line 235-240): The applied potentials have been polarized from 1.0 to 2.0 V in 200 mV steps (Figure 5a, Supplementary Figure 57). The *operando* Raman spectra has displayed a subset of vibrational data: the band at 1601 cm^{-1} is assigned to the C=C stretching vibration of the benzene ring, while the band at 1374 cm^{-1} corresponds to the symmetric stretching vibration of the carboxylate group (-COO⁻) in the MOF substrate.; the band at 3400 cm^{-1} from H₂O; the potential-dependent band at 1093 cm^{-1} from the Cl-O stretching mode of *OCl during CER process.³³”

“Supplementary information, line 530-550):

Generally, Raman spectra is a scattering signal with intrinsic weak intensity. When used to study

the structural characteristics of a bulk material, obvious signal peaks can be seen. This is due to the structural crystal lattice that contributes to overall vibration of the bulk material, leading to enhanced scattered signals.

While in this work, the *operando* Raman is used to probe the transient surface-adsorbed species (*OCl intermediates) during electrocatalysis. The scattered Raman signals are mainly focused on bond vibrations of adsorbed species on catalyst surfaces, which are known to be very weak as comparison to structural crystal lattices in bulk materials. Consequently, the *operando* Raman signals for catalytic reactions often demonstrate characteristically low signal-to-noise ratios in the literatures (like ORR, NRR, CRR).^{7,19,28} Our *operando* Raman signals are comparable to the above literature.

To confirm the accuracy of the experimental results, we performed five *operando* Raman tests. We directly determined the positions and intensities of the peaks using Raman spectrometer software, and updated the *operando* Raman data with error bars (Figures 5a-b and Supplementary Figure 57). By comparing the Raman vibration at different electrolysis potentials, we conclude the characteristic *OCl Raman peak at approximately 1093 cm^{-1} (inside the blue transparent frame), and the intensity of *OCl peak changes consistently with applied potentials, reaching maximum intensity at 2.0 V vs Ag/AgCl (Figure 5b). Further, the data demonstrate significantly greater accumulation of the *OCl intermediate under zero pressure differential conditions. This observation suggests that the Bernoulli principle leveraged in our reaction system generates an immediate pressure gradient, enabling continuous removal of Cl_2 product and thereby enhancing chlorine evolution kinetics. All of these results provide the validity of our *operando* synthesis condition.”

Ref 7. Wang, J. et al. Engineering the Coordination Environment of Ir Single Atoms with Surface Titanium Oxide Amorphization for Superior Chlorine Evolution Reaction. *J. Am. Chem. Soc.* **146**, 11152–11163 (2024).

Ref 19. Huang, Q. et al. Single-zinc vacancy unlocks high-rate H_2O_2 electrosynthesis from mixed dioxygen beyond Le Chatelier principle. *Nat. Commun.* **15**, 4157 (2024).

Ref 28. Quan, L., Zhao, X., Yang, L. M., You, B., Xia, B. Y. Intrinsic Activity Identification of Noble Metal Single-Sites for Electrocatalytic Chlorine Evolution. *Angew. Chem., Int. Ed.* **64**, 202414202 (2024).

Original Comment 19. *Fig. 5h exhibits mismatched sector colors between graphical representation and textual description.*

Response: We are sorry for the mistake. We have made corrections as follows:

“Page 13, Line 294-295: ...Ti-MOF material situates at the **dark blue** area characteristic of good stability in the pH range from 1 to 13”

“Page 13, Line 296-297: The Ti-MOF can still **maintain** its structure in pH=1~13 (Please see the **blue** area).”

“Page 14, Line 298-299: ...and the Ti-MOF can still **maintain** its durability in all pH ranges (as shown in the **pompadour** area).”

Original Comment 20. *The authors utilized Bernoulli's principle to regulate gas and liquid flow rates, forming a pressure difference of 116.3 mPa to promote Cl₂ outflow. However, it remains unclear whether this pressure difference remained stable during the reaction process. Furthermore, complex factors inherent to electrochemical reactions could significantly alter the interfacial pressure, such as gas product evolution and interfacial mass transfer. Critically, the authors determined this value solely through calculation using the Bernoulli equation, without providing experimental evidence to substantiate that this minute pressure difference was the key factor enabling efficient Cl₂ synthesis.*

Response: Thanks for your useful comment. We fully understand the reviewer's concern about the presence, stability and function of interfacial pressure difference at three-phase boundary. We have given the response as follows:

i) Firstly, as pointed out by the astute reviewer, the value of interfacial pressure difference is very small (milli-Pa level), and there are many factors influencing the pressure difference, such as gas product evolution (Cl₂ and O₂ byproduct) and interfacial mass transfer (like electrolyte). So, it is very difficult to directly monitor the pressure difference experimentally. Although there is already significant advance in *operando* characterization techniques (like *operando* Raman, FT-IR and XPS). There are seldom reports of *operando* manometer for measuring gas pressures. Therefore, we would like to provide some indirect evidence for the presence, stability and function of interfacial pressure difference.

ii) The presence/function of interfacial pressure difference can be confirmed by both theoretical simulations and experimental phenomenon of Cl_2 evolution. Theoretically, our COMSOL simulation based on Bernoulli's principle (Figure 5c-d) shows the pressure difference in the range of 0 ~ 116.3 mPa with the carrier flow of 0~80 mL min^{-1} , which drives as-generated Cl_2 migration to gas chamber. This prediction is well aligned with experimental results: by tuning the carrier flow rate from 0 to 80 mL min^{-1} , the Faradaic efficiencies of Cl_2 gas products increases from 18.5% to 82.1% (Figure 3c). Moreover, mechanism study has been conducted by *operando* Raman spectra for chlorine evolution reaction (CER) with/without interfacial pressure difference (Figure 5a and supplementary Figure 57). The decreased *OCl peak signal unambiguously confirm the role of interfacial pressure difference, *i.e.*, Cl_2 produced at the electrode/electrolyte interfaces intermediately taken away by gas flow following Bernoulli's principle.

iii) Further, the stability of interfacial pressure difference can be revealed by stability test of chlorine evolution reaction (CER) with interfacial pressure difference. Ti-MOF electrode exhibits strong stability in universal pH conditions (pH=1, 7, 13) with little morphology and structure decay after stability test (Figure 2h; Supplementary Figures 37–39). Further, the prototype membrane-free device shows long-term stability for Cl_2 electrosynthesis with little fluctuation for 200 hrs at a current density of 100 mA cm^{-2} (Figure 3g). Even at extremely high current density of 1.14 A cm^{-2} , our system can still demonstrate excellent stability for 40 hrs (Supplementary Figure 52). All of above results confirm the interfacial pressure difference stable for Cl_2 electrosynthesis.

Figure 2h, Stability test at different pHs for Ti-MOF at 100 mA cm^{-2} .

Figure 3. Prototype device for Cl₂ electroynthesis. **c**, The Cl₂ Faradaic efficiencies change with gas flow rates. **g**, Stability test for Ti-MOF for 200 hours (at 100 mA cm⁻²). The arrows below the pompadour dots refer to flushing the electrodes every 20 hours.

Figure 5. Mechanism study. **a**, The *operando* Raman spectra. **c-d**, Gas pressure distribution from COMSOL simulations.

Supplementary Figure 37. Characterization of Ti-MOF on GDE after 50-hrs CER test in 5 M NaCl at pH=1. a, XRD. b, FTIR.

Supplementary Figure 38. Characterization of Ti-MOF on GDE after 50-hrs test in 5 M NaCl at pH=7. a, XRD. b, FTIR.

Supplementary Figure 39. Characterization of Ti-MOF on GDE after 50-hrs test in 5 M NaCl at pH=13. a, XRD. b, FTIR.

Supplementary Figure 52. Ti-MOF stability experiments at the current density of 1.14 A cm^{-2} .

Supplementary Figure 57. The *operando* Raman spectra without pressure difference. a, *operando* Raman spectra. b, Ti-*OCl Raman peak intensity.

“Supplementary information, line 597-623):

As shown in Supplementary Figures 59-62, the value of interfacial pressure difference is very small (milli-Pa level), and there are many factors influencing the pressure difference, such as gas product evolution (Cl_2 and O_2 byproduct) and interfacial mass transfer (like electrolyte). So, it is very difficult to directly monitor the pressure difference experimentally. Although there is already significant advance in *operando* characterization techniques (like *operando* Raman, FT-IR and XPS). There are seldom reports of *operando* manometer for measure pressures yet. Therefore, we would like to provide some indirect evidence for the presence, stability and function of interfacial pressure difference.

Firstly, the presence/function of interfacial pressure difference can be confirmed by both theoretical simulations and experimental phenomenon of Cl₂ evolution. Theoretically, our COMSOL simulation based on Bernoulli's principle (Figure 5c-d) shows the pressure difference in the range of 0 ~ 116.3 10⁻³ Pa with the carrier flow of 0~80 mL min⁻¹, which will drive as-generated Cl₂ migration to gas chamber. This prediction is well aligned with experimental results: by tuning the carrier flow rate from 0 to 80 mL min⁻¹, the Faradaic efficiencies of Cl₂ gas products increases from 18.5% to 82.1% (Figure 3c). Moreover, mechanism study has been conducted by *operando* Raman spectra for chlorine evolution reaction (CER) with/without interfacial pressure difference (Figure 5a and supplementary Figure 57). The decreased *OCl peak signal unambiguously confirm the role of interfacial pressure difference, *i.e.*, Cl₂ produced at the electrode/electrolyte interfaces intermediately taken away by gas flow following Bernoulli's principle.

Further, the stability of interfacial pressure difference can be revealed by stability test of chlorine evolution reaction (CER) with interfacial pressure difference. Ti-MOF electrode exhibits strong stability in universal pH conditions (pH=1, 7, 13) with little morphology and structure decay after stability test after 50 hrs (Figure 2h; Supplementary Figures 37–39). The prototype device shows long-term stability for Cl₂ electrosynthesis with little fluctuation for 200 hrs at a current density of 100 mA cm⁻² (Figure 3g). Even at extremely high current density of 1.14 A cm⁻², our system can still demonstrate excellent stability for 40 hrs (Supplementary Figure 52). All of above results confirm the interfacial pressure difference stable for Cl₂ electrosynthesis.”

Original Comment 21. *The authors proposed the *OCl pathway in the CER process, where the Cl⁻ adsorbs on the Ti-O site. However, the outermost electrons shell of Cl⁻ is already full, and it's extremely difficult and unreasonable for Cl⁻ to adsorb onto the Ti-O site. Please provide a reasonable explanation.*

Response: We are sorry for explaining it clearly. To answer this question, we have retrieved relevant literature and conducted theoretical simulations, and given the response as follows:

Firstly, Cl-O bond is covalent in nature that can be formed by adsorbing Cl to oxygen species, as demonstrated in the common chemical of HClO (Figure R4). Therefore, there are many studies discussing the *OCl intermediate formation during CER process. As a typical example, Zhang and

co-workers have detected by *in situ* Raman spectrum both *Cl and *OCl intermediates on Ir-based active sites (Figure R5). *J Am Chem Soc* 2024, 146, 11152 They have discussed the origin of these intermediates from the perspective of engineering coordination environment (Ir₁O₄ and Ir₁O₆). On another study, the authors have attributed the key of forming *OCl intermediate to appropriate adsorption energy caused by potential differences in electrochemical reaction processes (Figure R6). *ACS Catal* 2018, 8, 9034

Inspired by above studies, we have conducted theoretical calculations of charge transfer profiles (Supplementary Figures 67, 69), where the increase in Ti's electron transfer number indicates that the bonded O acquires electrons (1.89 vs. 1.808 eV). Subsequently, the elevated electron density around O enhances its orbital overlap with Cl (Figure 5g), as demonstrated by distinct p-p orbital overlap between O and Cl (green and red lines). This will prompt formation of chemical bonds of Cl-O, and consequently *OCl intermediate during CER process.

Based on above discussions, we have added/revised relevant sections in updated manuscript:

Figure R4 The electronic structural formula of HClO.

[FIGURE REDACTED]

Figure R5. CER mechanism studies. (c) Free energy diagrams of the CER process over Ir₁O₄ and Ir₁O₆. (d, e) In situ Raman spectrum of (d) Ir₁O₄ and (e) Ir₁O₆ collected at CER conditions. (*J. Am. Chem. Soc.* 2024, 146, 11152)

[FIGURE REDACTED]

Figure R6. Surface Pourbaix diagrams for (a) IrO₂ and (b) RuO₂. The bridge and cus sites of adsorption on rutile (110) oxides have been denoted by the superscripts b and c, respectively (ACS Catalysis, 2018, 8, 9034).

Supplementary Figure 69. Charge transfer profiles of Ti-MOF. a, 3D differential charge transfer of Ti-MOF. b, 2D differential charge transfer of Ti-MOF.

“Page 12-13, Line 272-273: ...the local modeling diagram for *O → *OCl pathway and the lower part of * → *Cl pathway.^{36,37}”

Ref 36. Wang, J. et al. Engineering the Coordination Environment of Ir Single Atoms with Surface Titanium Oxide Amorphization for Superior Chlorine Evolution Reaction. *J. Am. Chem. Soc.* 146, 11152–11163 (2024).

Ref 37. Sumaria, V., Krishnamurthy, D., Viswanathan, V. Quantifying Confidence in DFT Predicted Surface Pourbaix Diagrams and Associated Reaction Pathways for Chlorine Evolution. *ACS Catal.* 8, 9034–9042 (2018).

“Supplementary information, Line 655-661: The Cl-O bond is covalent in nature that can be formed by adsorbing Cl to oxygen species, as demonstrated in the common chemical of HClO. Inspired by

this structure, we have conducted theoretical calculations of charge transfer profiles, where the increase in Ti's electron transfer number indicates that the bonded O acquires electrons (1.89 vs. 1.808 eV). Subsequently, the elevated electron density around O enhances its orbital overlap with Cl (Figure 5g), as demonstrated by distinct p-p orbital overlap between O and Cl (green and red lines). This will prompt formation of chemical bonds of Cl-O, and consequently *OCl intermediate during CER process.”

Original Comment 22. *In Figure 5a, the intensity of the Raman characteristic peaks corresponding to Ti-*OCl didn't exhibit an obvious dependence on applied potentials. In addition, no significant correlation can be identified between the peak intensity in Figure 5a and the intensity analysis diagram in Figure 5b. Please optimize the experimental conditions to obtain more conclusive results.*

Response: Thanks for your constructive comment. Following the reviewer's kind suggestion, we have re-tested the *operando* Raman for five times, and re-examined the relevant peak intensities at different applied potentials. Please see the updated data and descriptions in response to **Original Comment 18** (above).

Original Comment 23. *The experimental data and supplementary videos could only confirm the absence of H₂ in the flowing gas, and cannot determine whether H₂ and Cl₂ are mixed within the reactor. Please design experiments to prove the successful separation of the generated H₂ and Cl₂.*

Response: We are sorry for explaining it clearly. The experimental data (Figure 3b) and Supplementary Video S3 only show the gas chromatography (GC) profile of anodic products, which illustrate the signals of Cl₂ and carrier gas. The absence of H₂ in Figure 3b provides evidence of no H₂ crossover in our membrane-free system.

On the other hand, to verify the production of H₂ in membrane-free system, we have collected the cathodic products, and characterized by gas chromatography (GC). As shown in Supplementary Figure 45, the H₂ signal can be detected by GC.

Based on above results, we can conclude the successful separation of the generated H₂ and Cl₂ in our membrane-free system. Accordingly, the following figure/sentence has been added/revised in updated manuscript:

Supplementary Figure 45. Gas chromatography (GC) profile of cathodic products.

“Page 8, Line 158-160: Finally, The H_2 generated at the cathode is collected and measured by gas chromatography, gas-phase results (Supplementary Figure 45) demonstrate that H_2 is discharged from the reaction system along with the electrolyte.”

Original Comment 24. Please provide the stability experiment at the current density of $1.14 A cm^{-2}$.

Response: Thank you for your constructive comment. We have conducted stability test at $1.14 A cm^{-2}$, and given the response as follows:

Supplementary Figure 52. Ti-MOF stability experiments at the current density of $1.14 A cm^{-2}$.

“Page 10, Line 199-200: Finally, the stability of prototype device was tested under $1.14 A cm^{-2}$ (Supplementary Figure 52), which demonstrate excellent durability for 40 hrs.”

We would like to thank the Editor's invitation to revise the manuscript. We are also appreciated of the reviewer #2 for his/her constructive comments to improve the quality of this work.

After receiving the reviewer's comments, we have spent ten intensive weeks in order to answer the concerns raised by the astute reviewer. We believe we have shifted the work to the next level of proof and gained a much deeper understanding of the system. Please see the following point-by-point response to the reviewer #2's comments. Thank you and best wishes,

Kind regards

The authors

Response to Reviewer #2

Original Comment: *This study presents an innovative membrane-free electrochemical system for chlorine (Cl₂) production, leveraging Bernoulli's principle to drive Cl₂ migration via a triple-phase gas diffusion layer. The work demonstrates impressive Faradaic efficiencies (up to 96.3%) and stability, with potential cost savings over conventional chlor-alkali processes. While the concept is compelling and the results are promising, several key issues require clarification to strengthen the manuscript's rigor and impact.*

Original Comment 1. *Given the harsh electrochemical conditions (high current densities, extreme pH), surface reconstruction or phase changes of the MOFs are inevitable. The authors should address this critical aspect more thoroughly.*

Response: Thanks for your kind reminding. We have conducted a series of characterizations for Ti-MOF before and after test, and given the following response:

1) Firstly, the Ti-MOF was characterized by XRD (Supplementary Figure 16c). After loading Ti-MOF on GDE (gas-diffusion electrode) substrate, there is a large carbon peak around 26°. However, many crystal plane peaks belonging to Ti-MOF still exist (15.9°, 18.1°, 31.7° and 45.5°). Even after extreme acidic/base conditions and long-term stability tests (Supplementary Figures 20c, 23c and 48a), these crystal plane peaks still exist and show little change.

2) Secondly, we have conducted Fourier transform infrared spectroscopy analysis (Supplementary Figure 16b). Many vibration signals exist after Ti-MOF loaded on GDE substrate: Ti-O (629 cm⁻¹), organic-ligant (C=C 1604 cm⁻¹, C-H 979, 1061 cm⁻¹), and C-O (1147, 1207 cm⁻¹). *J Am Chem Soc* 2021, 143, 1107. These peaks show insignificant change after extreme acidic/base condition or long-term stability tests (Supplementary Figures 20b, 23b and 48b).

3) Further, XPS was conducted for both before and after electrochemical tests (Supplementary Figure 19). The Ti 2p of Ti-MOF was detected before and after reaction. In the O 1s spectrum, Ti-O (lattice oxygen), C=O and C-O were detected. Compared with the characteristic peaks before test, the binding energies of all subsequent peaks changed insignificantly. Moreover, in C 1s spectrum, C=C, C-C, C-O and π - π^* peaks were detected. The positions of these peaks rarely changed, which indicates Ti-

MOF maintains good reaction stability.

Based on above experimental data, we conclude that Ti-MOF is a stable metal-organic framework material. One of the reasons is that the voltage applied to chlorine evolution reaction is positive, which makes it difficult to further oxidize Ti-MOF, as it already in a relatively high oxidation state.^{*Matter Chem Phy 2015, 151, 133*} Similar to previous literature, Ti-based MOF materials are often used as stable active materials in catalytic oxidation reactions.^{*J Ind Eng Chem 2024, 137, 606*} Secondly, the strong Ti-O coordination interaction inside Ti-MOF also stabilizes the structure, especially the Ti-MOF has been subjected to a low-temperature calcination to enhance its crystallinity (Figure 2b) in our work.^{*Matter 2020, 2, 440*} Therefore, we believe that Ti-MOF exhibits excellent stability under harsh electrochemical conditions.

Based on above discussions, the following sections have been added/revised in updated manuscript:

Supplementary Figure 16. The characterizations of Ti-MOF on gas diffusion electrode (GDE) before chlorine evolution reaction (CER). b, FTIR. c, XRD.

Supplementary Figure 20. Characterization of Ti-MOF on GDE after CER in 5 M NaCl at pH=13. **b**, FTIR. **c**, XRD.

Supplementary Figure 23. Characterization of Ti-MOF on GDE after CER in 5 M NaCl at pH=1. **b**, FTIR. **c**, XRD.

Supplementary Figure 48. Characterization of Ti-MOF on GDE after 200-hrs CER test in 5 M NaCl. **a**, XRD. **b**, FTIR.

Supplementary Figure 19. XPS spectra of Ti-MOF on GDE before and after CER in 5 M NaCl at pH=13. **a**, Ti 2p, **b**, Na 1s, **c**, C 1s, **d**, O 1s.

“Supplementary information, Line 261-278: To explore the changes of Ti-MOF before and after the reaction, we conducted a series of studies on it before and after the reaction:

1) Firstly, the Ti-MOF was characterized by XRD (Supplementary Figure 16c). After loading Ti-MOF on GDE (gas-diffusion electrode) substrate, there is a large carbon peak around 26° . However, many crystal plane peaks belonging to Ti-MOF still exist (15.9° , 18.1° , 31.7° and 45.5°). Even after extreme acidic/base conditions and long-term stability tests (Supplementary Figures 20c, 23c and 48a), these crystal plane peaks still exist and show little change.

2) Secondly, we have conducted Fourier transform infrared spectroscopy analysis (Supplementary Figure 16b). Many vibration signals exist after Ti-MOF loaded on GDE substrate: Ti-O (629 cm^{-1}), organic-ligand (C=C 1604 cm^{-1} , C-H 979 , 1061 cm^{-1}), and C-O (1147 , 1207 cm^{-1}).²² These peaks show insignificant change after extreme acidic/base condition or long-term stability tests

(Supplementary Figures 20b, 23b and 48b).

3) Further, XPS was conducted for both before and after electrochemical tests (Supplementary Figure 19). The Ti 2p of Ti-MOF was detected before and after reaction. In the O 1s spectrum, Ti-O (lattice oxygen), C=O and C-O were detected. Compared with the characteristic peaks before test, the binding energies of all subsequent peaks changed insignificantly. Moreover, in C 1s spectrum, C=C, C-C, C-O and π - π^* peaks were detected. The positions of these peaks rarely changed, which indicates Ti-MOF maintains good reaction stability.”

“Supplementary information, Line 472-479: Based on above experimental data, we conclude that Ti-MOF is a stable metal-organic framework material. One of the reasons is that the voltage applied to chlorine evolution reaction is positive, which makes it difficult to further oxidize Ti-MOF, as it already in a relatively high oxidation state.²⁶ Similar to previous literature, Ti-based MOF materials are often used as stable active materials in catalytic oxidation reactions.²⁷ Secondly, the strong Ti-O coordination interaction inside Ti-MOF also stabilizes the structure, especially the Ti-MOF has been subjected to a low-temperature calcination to enhance its crystallinity (Figure 2b) in our work.⁵ Therefore, we believe that Ti-MOF exhibits excellent stability under harsh electrochemical conditions.”

Ref 22. Lim, H. W. et al. Rational Design of Dimensionally Stable Anodes for Active Chlorine Generation. *ACS Catal.* **11**, 12423–12432 (2021).

Ref 26. Rasmi, K. R., Vanithakumari, S. C., George, R. P., Mallika, C., Kamachi Mudali, U. Development and performance evaluation of nano platinum coated titanium electrode for application in nitric acid medium. *Mater. Chem. Phys.* **151**, 133–139 (2015).

Ref 27. Pirkarami, A., Javanmard, A., Ghasemi, E. A highly-stable and low-energy-barrier photoelectrocatalyst (NiFe-LDH@Ti-MOF) for water splitting at high current densities via breaking the atomic structure symmetry. *J. Ind. Eng. Chem.* **137**, 606–618 (2024).

Original Comment 2. *If the Ti-MOF catalyst undergoes surface reconstruction under operational conditions, the initial structure used for calculations may not reflect the true active phase.*

Response: Thanks for your kind reminding. As discussed in **response to Comment 1** (Above), the Ti-MOF catalyst has not undergone significant surface reconstruction. So, the initial structure used

for calculations is OK. And the calculation results are still valid in this paper.

Original Comment 3. *The authors should include the k-space of the XAFS data in the supplementary information. In addition, an R-space range of 1 to 6 Å is sufficient to capture the main coordination shells of the materials. The EXAFS fitting results in Supplementary Table 1 should include standard deviations or confidence intervals for each fitted parameter.*

Response: We are appreciated of the reviewer for his/her helpful suggestions. We have made corresponding changes, and given the response as follows:

i) The k-space XAFS data for all relevant edges have been included in updated manuscript as Supplementary Figures 9c, 10.

ii) We agree that the R-range of 1–6 Å captures primary coordination shells of the analyses. Therefore, all of R-space figures have been replotted (Figure 2e; Supplementary Figures 7, 8, 9b, d and 10).

iii) We have added standard deviations into EXAFS fitting results in Supplementary Table 2.

Figure 2. Catalyst and system design for Cl₂ electrosynthesis beyond Pourbaix diagram. e, Extended X-ray absorption fine structure (EXAFS) and fitting curves spectra of Ti-MOF, Ti-foil and TiO₂.

Supplementary Figure 7. Extended X-ray absorption fine structure (EXAFS) and data fitting.

a, Ti-foil. b, Ti-MOF. c, TiO_2 .

Supplementary Figure 8. Synchrotron characterizations of Ti-MOF before calcination. a, X-ray absorption near edge structure (XANES). b, d, Extended X-ray absorption fine structure (EXAFS). c, Wavelet transform plots (WT-plots).

Supplementary Figure 9. Synchrotron radiation of TiC. a, XANES spectra of reference TiC. b,d EXAFS spectra of reference TiC and data fitting. c, the k-space XAFS data for edges of reference TiC.

Supplementary Figure 10. The k-space XAFS data for edges for different samples. a, Ti-foil. b, TiO₂. c, Ti-MOF before calcination. d, Ti-MOF.

Supplementary Table 2. Bond length and coordination of Ti-MOF-before and after calcination

Material	shell	CN	R (Å)	ΔE_0 (eV)	σ^2 (10^{-3} \AA^2)	R factor
Ti-foil	Ti-Ti	6.24±0.21	2.89±0.002	-7.05±0.18	0.006±0.0005	0.0084±0.00224
TiO ₂	Ti-C	4.15±0.05	1.95±0.003	-1.55±0.26	0.006±0.0001	0.014±0.0012
	Ti-Ti	2.05±0.05	3.06±0.0004		0.0040±0.00025	
TiC	Ti-O	6.03±0.08	2.14±0.00039	-3.47±0.01	0.0002±0.0001	0.0095±0.00035
	Ti-Ti	11.93±0.14	3.05±0.0002		0.0009±0.0001	
Ti-MOF (before)	Ti-O	3.98±0.04	1.90±0.0005	-7.04±0.10	0.016±0.0001	0.0034±4.1×10 ⁻⁵
	Ti-Ti	4.05±0.002	3.10±10 ⁻⁷		0.019±1.5×10 ⁻⁵	
Ti-MOF	Ti-O	3.36±0.11	1.92±0.002	-4.21±0.21	0.011±0.0009	0.013±0.0043
	Ti-Ti	4.05±0.09	3.12±0.00008		0.018±0.00064	

“Supplementary information, Line 719-724: **Supplementary note:** CN: coordination number; R: bond lengths between central atoms and surrounding coordination atoms; ΔE_0 : the difference of the zero kinetic energy value between the sample and theoretical model; σ^2 : Debye-Waller factor to account for both thermal and structural disorders; R factor is used to measure the goodness of the fitting. All the data in the figure are the average results after five repeated fittings, and the standard deviation of each average value is added to indicate the relative deviation of the value.”

“Page 6, Line 98-100: The Ti-MOF has presented XANES absorption edge energy **between Ti-foil**

and TiO₂ (4962.95 vs. 4961.22 and 4963.46 eV), indicating its valence state of Ti in the range of 0 ~ +4. Different from Ti foil bearing only Ti-Ti coordination (distance: 2.89 Å) ...”

“Page 6, Line 103-104: The Ti-O distance (1.92 Å) is significantly shorter as comparison to Ti-C bond distance (2.14 Å, Supplementary Figures 9–10).”

Original Comment 4. *While wavelet transform analysis can provide additional insights into the EXAFS data, its utility in this context is limited. The structural information obtained from WT may not be highly accurate. Given this, the authors should either provide a more detailed justification for including WT analysis or consider omitting it if it does not significantly contribute to the understanding of the catalyst's structure.*

Response: We are sorry for not explaining the key function of WT analysis clearly. We would like to give the response as follows:

Firstly, WT analysis can serve as a key method to resolve ambiguous structures. In the present study, it has been used to distinguish the overlapping contributions of Ti-O shell (1.8–2.0 Å)^{*Nat Commun* 2024, 15, 7650} and Ti-Ti shell (2.8–3.5 Å):^{*Phil Rev B* 1981, 23, 3781} *i*) the WT intensity maximizes at $k = 4\text{--}6 \text{ \AA}^{-1}$ (Ti-O) and $k = 6\text{--}8 \text{ \AA}^{-1}$ (Ti-Ti) providing independent validation of the atomic pair assignments for fitting models; *ii*) WT analysis maps the EXAFS signals in the k-space (energy) and R-space (distance), thus providing a two-dimensional representation. This can clearly separate these overlapping shells based on their scattering amplitudes and phase differences. The high spatial resolution is very helpful in confirming the coexistence of multiple coordination environments, which is crucial for understanding the structural evolution of the catalyst under reaction conditions.

Secondly, WT analysis can serve as a powerful qualitative tool for probing coordination environments in the present study (Supplementary table 2). When processing and analyzing images, WT better mimics the human visual system's perception mechanisms, aligning closely with human visual cognition patterns. Therefore, it enables more effective capture and processing of fine details and edge information within images, resulting in processed visuals that appear more natural and sharper. This enhancement facilitates researchers' accurate extraction of information from images, thereby improving the precision and reliability of image-based analysis.

On the other hand, as pointed out by the reviewer, WT analysis also has some limitations. For example, it cannot be not used solely for structure determination, but rather as a supplementary method that enriches the understanding of local structure of a catalyst. Also, the structural information obtained from WT may not be highly accurate. Nevertheless, these limitations have not affected the main conclusion of WT analysis in the present study.

Based on above discussions, we would like to keep WT analysis data in the updated manuscript, and give explanations as follows:

“Page 6, Line 107-109: It is known that WT analysis can serve as a qualitative supplement to EXAFS fitting, which can resolve overlapping coordination shells and validate scattering paths. Consequently, quantitative structural parameters of WT analysis have been derived from r-space fitting.”

“Supplementary information, Line 207-222: WT analysis can serve as a key method to resolve ambiguous structures. In the present study, it can be used to distinguished the overlapping contributions of Ti-O shell (1.8–2.0 Å)¹⁹ and Ti-Ti shell (2.8–3.5 Å):²⁰ *i*) the WT intensity maximizes at $k = 4\text{--}6 \text{ \AA}^{-1}$ (Ti-O) and $k = 6\text{--}8 \text{ \AA}^{-1}$ (Ti-Ti) providing independent validation of the atomic pair assignments for fitting models; *ii*) WT analysis maps the EXAFS signals in the k-space (energy) and R-space (distance), thus providing a two-dimensional representation. This can clearly separate these overlapping shells based on their scattering amplitudes and phase differences. The high spatial resolution is very helpful in confirming the coexistence of multiple coordination environments, which is crucial for understanding the structural evolution of the catalyst under reaction conditions.

Secondly, WT analysis can serve as a powerful qualitative tool for probing coordination environments in the present study (Supplementary table 2). When processing and analyzing images, WT better mimics the human visual system’s perception mechanisms, aligning closely with human visual cognition patterns. Therefore, it enables more effective capture and processing of fine details and edge information within images, resulting in processed visuals that appear more natural and sharper. This enhancement facilitates researchers’ accurate extraction of information from images, thereby improving the precision and reliability of image-based analysis.”

Ref 19. Huang, Q. et al. Single-zinc vacancy unlocks high-rate H₂O₂ electrosynthesis from mixed dioxygen beyond Le Chatelier principle. *Nat. Commun.* **15**, 4157 (2024).

Ref 20. Stern, E. A. Kim, K. Thickness effect on the extended-x-ray-absorption-fine-structure amplitude. *Phys. Rev. B* **23**, 3781-3787 (1981).

Original Comment 5. *The authors should include photographs of the internal construction of the prototype device. In addition, a critical aspect that remains unclear is how this membrane-free device ensures water does not leak into the adjacent gas chamber.*

Response: Thanks for your helpful comment. We have given the response as follows:

1) We have provided photographs of the internal construction of prototype device as Supplementary Figure 40. Specifically, Supplementary Figure 40a shows a photograph of working electrode assembly, with white PTFE hydrophobic membrane laminated on the backside. Supplementary Figure 40b presents the assembled anode/gas chamber unit, where the carrier gas flow path is indicated by orange arrows (inlet/outlet). Supplementary Figures 40c, d display the reaction chamber diagram.

2) In terms of the water leakage concern, we consider the multilayered waterproofing design of GDE/PTFE robustly preventing aqueous phase penetration into the gas chamber. More specifically, the gas diffusion electrode (GDE) maintained near-constant hydrophobicity on its gas-facing side (149° pre-reaction vs 148° post-reaction; Supplementary Figure 79a-b), confirming its effectiveness as the barrier against electrolyte penetration. Secondly, the hydrophobicity has been further reinforced by the PTFE partition layer (Supplementary Figure 40a), which exhibits persistent hydrophobicity on both interfaces: GDE-facing side ($152^\circ \rightarrow 150^\circ$) and gas chamber side ($153^\circ \rightarrow 151^\circ$) as shown in Supplementary Figure 79c-f. As a consequence, the minimal variations of contact angle ($\Delta\theta \leq 2^\circ$) demonstrate robust waterproof integrity throughout operation.

To further verify above hypothesis, we have conducted long-term barrier integrity and mass transfer resistance of PTFE membrane in H-cell experiment (Supplementary Figure 80). Deionized water (DIW) and yellow FeCl_3 solution were physically separated by the PTFE membrane in adjacent chambers. We can see that no color change in the DIW sample after one week, which reveals the effective liquid isolation function of PTFE.

Based on above discussions, the following pictures/data/sentences have been added into updated manuscript:

Supplementary Figure 40. a, working electrode assembly. b, anode/gas chamber unit. c, reaction chamber. d, Overview of reactor configuration.

Supplementary Figure 79. Contact Angle test. a-b, GDE before and after testing. c-d, rough and smooth PTFE surfaces before testing. e-f, rough and smooth PTFE surfaces after testing.

Supplementary Figure 80. Test of PTFE water resistance in H-cell. a-c, observe the water resistance of PTFE from different perspectives.

“Page 8, Line 149-150: Next, a standalone prototype device has demonstrated for membrane-free chlor-alkali process (Supplementary Video S3, Supplementary Figures 40-42).”

“Page 17, Line 370-373: Notably, to prevent electrolyte penetration into gas chamber, the counter electrode was engineered with the specialized design shown in Supplementary Figure 79. The feasibility of this approach has been demonstrated through contact angle measurements and PTFE waterproof testing (Supplementary Figure 80).”

“Supplementary information, Line 436-439: Supplementary Figure 40a shows a photograph of working electrode assembly, with white PTFE hydrophobic membrane laminated on the backside. Supplementary Figure 40b presents the assembled anode/gas chamber unit, where the carrier gas flow path is indicated by orange arrows (inlet/outlet). Supplementary Figures 40c, d display the reaction chamber diagram.”

“Supplementary information, Line 702-715:

In terms of the water leakage concern, we consider the multilayered waterproofing design of GDE/PTFE robustly preventing aqueous phase penetration into the gas chamber. More specifically,

the gas diffusion electrode (GDE) maintained near-constant hydrophobicity on its gas-facing side (149° pre-reaction vs 148° post-reaction; Supplementary Figures 79a-b), confirming its effectiveness as the barrier against electrolyte penetration. Secondly, the hydrophobicity has been further reinforced by the PTFE partition layer (Supplementary Figure 40a), which exhibits persistent hydrophobicity on both interfaces: GDE-facing side (152°→150°) and gas chamber side (153°→151°) as shown in Supplementary Figures 79c-f. As a consequence, the minimal variations of contact angle ($\Delta\theta\leq 2^\circ$) demonstrate robust waterproof integrity throughout operation.

To further verify above hypothesis, we have conducted long-term barrier integrity and mass transfer resistance of PTFE membrane in H-cell experiment (Supplementary Figure 80). Deionized water (DIW) and yellow FeCl₃ solution were physically separated by the PTFE membrane in adjacent chambers. We can see that no color change in the DIW sample after one week, which reveals the effective liquid isolation function of PTFE.”

We would like to thank the Editor's invitation to revise the manuscript. We are also appreciated of the reviewer #3 for his/her constructive comments to improve the quality of this work.

After receiving the reviewer's comments, we have ten intensive weeks in order to answer the concerns raised by the astute reviewer. We believe we have shifted the work to the next level of proof and gained a much deeper understanding of the system. Please see the following point-by-point response to the reviewer #3's comments. Thank you and best wishes,

Kind regards

The authors

Response to Reviewer #3

Original Comment: *The authors have reported a chlorine evolution system based on Bernoulli principle, which promotes oriented migration of Cl₂ away from reaction interfaces. This has not only prevented H₂/Cl₂ crossover but also circumvented the Pourbaix diagram problems, leading to significantly enhanced activities. By further joining with a pH-tolerant catalyst, a membrane-free prototype device has been built for high-rate Cl₂ production. Further, the strategy has been demonstrated for solving key challenging problems in other chemical/catalytic reactions like water splitting ($2\text{H}_2\text{O} \rightarrow 2\text{H}_2 + \text{O}_2$). Overall, the strategy is interesting, and the manuscript is well organized. It can be published in Nature Communications after addressing the following concerns.*

Original Comment 1. *In Figure 3c, the authors have tested Cl₂ evolution in membrane-free prototype device using CO₂ and air carrier gases. However, only the Faradaic efficiency of air carrier has been provided. Please provide the relevant data for CO₂ carrier gas.*

Response: Thanks for your kind reminding. We have conducted experiments, and added the relevant data for CO₂ carrier gas into Figure 3c as follows:

Figure 3c, The Cl₂ Faradaic efficiencies change with gas flow rates.

Original Comment 2. *The author discussed the activities of Cl₂ electrosynthesis on gas diffusion electrode-supported catalysts. However, the performances of gas diffusion electrode substrate have not been provided.*

Response: Thanks for your constructive comment. We have conducted comparison experiments for gas diffusion electrode substrate, and added the data as Supplementary Figure 13 as follows:

Supplementary Figure 13. Electrochemical tests of GDE substrate in 5 M NaCl at pH=7. a, CV. b, Cl₂ Faradic efficiencies and current densities from 1.8 to 2.4 V (vs. Ag/AgCl). c, LSV. d, Tafel slope from LSV.

“Page 6, Line 118-119: Subsequently, we start electrochemical performance evaluation for different samples. The gas diffusion electrode (GDE) substrate has shown low Cl₂ Faradaic efficiencies (Supplementary Figure 13).”

Original Comment 3. *In terms of mechanism (Figures 5), why did the authors choose *Cl and *OCl as reaction intermediates? What is the reason of choosing these two intermediates for DFT calculates?*

Response: Thank you for useful comment. These two intermediates have been selected for CER according to both literature and experiment results. In details,

- i) In the literature, there are many studies discussing the *OCl and/or *Cl intermediate formation during CER process. As a typical example, Zhang and co-workers have detected by *in situ* Raman spectrum both *Cl and *OCl intermediates on Ir-based active sites (Figure R1). *J Am Chem Soc* 2024, 146, 11152 They have discussed the origin of these intermediates from the perspective of engineering coordination environment (Ir₁O₄ and Ir₁O₆). On another study, the authors have attributed the key of forming *OCl intermediate to appropriate adsorption energy caused by potential differences in electrochemical reaction processes (Figure R2). *ACS Catal* 2018, 8, 9034
- ii) Further, in this work, we have experimentally observed *OCl intermediate through *in-situ* Raman experiments (Figure 5a-b and Supplementary Figure 57). Therefore, we start the DFT calculations from *Cl and *OCl in CER process. Based on above discussions, we have added relevant explanations in updated manuscript.

[FIGURE REDACTED]

Figure R1. CER mechanism studies. (c) Free energy diagrams of the CER process over Ir₁O₄ and Ir₁O₆. (d, e) In situ Raman spectrum of (d) Ir₁O₄ and (e) Ir₁O₆ collected at CER conditions.

[FIGURE REDACTED]

Figure R2. Surface Pourbaix diagrams for (a) IrO₂ and (b) RuO₂. The bridge and cus sites of adsorption on rutile (110) oxides have been denoted by the superscripts b and c, respectively.

Figure 5. Mechanism study. **a**, The *operando* Raman spectra. **b**, Ti-*OCl Raman peak intensity.

Supplementary Figure 57. The *operando* Raman spectra without pressure difference. **a**,

operando Raman spectra. **b**, Ti-*OCl Raman peak intensity.

“Page 11, line 236-240): The *operando* Raman spectra has displayed a subset of vibrational data: the band at 1601 cm^{-1} is assigned to the C=C stretching vibration of the benzene ring, while the band at 1374 cm^{-1} corresponds to the symmetric stretching vibration of the carboxylate group ($-\text{COO}^-$) in the MOF substrate.; the band at 3400 cm^{-1} from H_2O ; the potential-dependent band at 1093 cm^{-1} from the Cl-O stretching mode of *OCl during CER process.³³”

“Supplementary information, Line 26-35:

In the literature, there are many studies discussing the *OCl and/or *Cl intermediate formation during CER process. As a typical example, Zhang and co-workers have detected by *in situ* Raman spectrum both *Cl and *OCl intermediates on Ir-based active sites.⁷ They have discussed the origin of these intermediates from the perspective of engineering coordination environment (Ir_1O_4 and Ir_1O_6). On another study, the authors have attributed the key of forming *OCl intermediate to appropriate adsorption energy caused by potential differences in electrochemical reaction processes.⁸ Further, in this work, we have experimentally observed *OCl intermediate through *in-situ* Raman experiments (Figure 5a-b). Therefore, we start the DFT calculations from *Cl and *OCl) in the CER process.”

Ref 7. Wang, J. et al. Engineering the Coordination Environment of Ir Single Atoms with Surface Titanium Oxide Amorphization for Superior Chlorine Evolution Reaction. *J. Am. Chem. Soc.* **146**, 11152–11163 (2024).

Ref 8. Sumaria, V., Krishnamurthy, D., Viswanathan, V. Quantifying Confidence in DFT Predicted Surface Pourbaix Diagrams and Associated Reaction Pathways for Chlorine Evolution. *ACS Catal.* **8**, 9034–9042 (2018).

Original Comment 4. *Some experimental details need to be provided, such as the optical picture of the synthetic and electrochemical testing processes.*

Response: We have added relevant pictures for synthetic products and electrochemical testing electrodes/cells as follows:

Supplementary Figure 30. Optical pictures of the synthetic processes and electrode preparation.

a-b, Optical pictures of Ti-MOF-before and Ti-MOF. **c-d,** electrode have an effective active area of $1 \times 1 \text{ cm}^2$. **e,** Electrolysis reaction schematic diagram.

“Page 15, Line 329-330: More details of materials, electrodes and reactor unit are shown in Supplementary Figures 30a-d.”

Original Comment 5. For DFT calculations (Figures 5f), the author should provide more details on building the slab models. Why they use the cluster model for theoretical calculations?

Response: We are appreciated of your useful comment. We have given the response as follows:

(1) Details on building the slab models: The modeling framework has been built on the basis of experiment results including: XRD patterns (Figure 2b), FTIR (Figure 2c) and X-ray absorption spectroscopy (XAS, Figure 2d-e) and supplementary structural analyses (Supplementary Tables 1–2). Firstly, based on XRD, we determined that Ti-MOF is a relatively well-crystallized substance, which enables us to refer to some relevant models from previous literature. *Matter* 2020, 2, 440 Secondly, the results of FTIR indicated that there are incorporated organic ligands in Ti-MOF. *J Am Chem Soc* 2021, 143, 1107 Finally, XAS showed that the coordination number of Ti and O in Ti-MOF is 3.36.

(2) Reasons for using cluster model. Firstly and most importantly, the aim of DFT calculations is to explain the chlorine evolution reaction (CER) mechanism on Ti-MOF electrode. Based on our calculation results from cluster model (Figures 5e-i), the detailed CER mechanism has been successfully unravelled, *i.e.*, CER pathway by calculating Gibbs free energy change of reaction intermediates, as well as Pourbaix diagrams showing the surface adsorption changes with applied potentials at different pHs. Further, we have unraveled the high CER selectivity by comparison with Gibbs free energy change of side OER process.

As with many other reports, another reason for using cluster model is limited computational resource. In this study, we have focused on a representative portion of Ti-MOF material, and aimed to investigate the mechanism, particularly reaction intermediate and pathways. Actually in the literature, there are enormous state-of-the-art studies employing a cluster model for MOF-based materials: *i*) The cluster model of Au₆ combined with Zr₆O₈ was built to study the formation of CH₄ by carbon dioxide reduction (Figure R3) (*Angew Chem Int Ed* 2025, e202500269); *ii*) The cluster model of cMOF combined with Cu₂O was established to conduct a series of theoretical calculations (Figure R4). The combination of the calculation results and the experimental results verified the reaction path and explained the material advantages after the combination of cMOF (100) and Cu₂O (*Nano Energy* 141, 2025, 111077); *iii*) Due to the large scale of the modeling system, the author also extracted a part of MOF for study (Figure R5). The structural data such as bond length and bond angle

are studied, and the properties of the MIL-100(Fe) small cluster structure and its adsorption capacity for NO are studied more thoroughly (*Adv Mater* 2024, 36, 2403053); *iv*) due to the large size of the MAF-203 system, the authors adopted a smaller cluster model to simulate the molecular dynamics process of MAF-203 (Figure R6). Detailed data were given for the positions of atoms and bond lengths in the system (*J Am Chem Soc* 2025, 147, 5, 4595).

[FIGURE REDACTED]

Figure R3. Reaction coordinates in the pathway of CO₂ reduction over Au₆/Zr₆O₈. (*Angew. Chem. Int. Ed.* 2025, e202500269)

[FIGURE REDACTED]

Figure R4. Free energy of different C-C reaction pathway and the optimized reaction intermediates (*Nano Energy* 141, 2025, 111077).

[FIGURE REDACTED]

Figure R5. DFT model of MIL-100(Fe). (a) An extended cluster model taken from the MIL-100(Fe) crystal structure. (b) Fe small cluster model taken from the circle in the large cluster with adsorbed NO molecule (*Adv. Mater.* 2024, 36, 2403053).

[FIGURE REDACTED]

Figure R6. Time-dependent structural evolution of the intermediates during the O₂ adsorption process (*J. Am. Chem. Soc.* 2025,147, 5, 4595).

“Supplementary information, Line 14-16: The Ti-MOF model framework has been built on the basis of experiment results and literature including: XRD patterns (Figure 2b),⁵ FTIR (Figure 2c)⁶ and X-ray absorption spectroscopy (XAS, Figure 2d-e; Supplementary Tables 1–2).”

Ref 5. Wang, S. et al. Toward a Rational Design of Titanium Metal-Organic Frameworks. *Matter* **2**, 440–450 (2020).

Ref 6. Feng, X. et al. Rational Construction of an Artificial Binuclear Copper Monooxygenase in a Metal-Organic Framework. *J. Am. Chem. Soc.* **143**, 1107–1118 (2021).

Original Comment 6. *The authors have claimed Cl₂ migration to gas chamber by pressure difference according to Bernoulli's principle. However, the description is too simple. More in-depth discussion should be given, both from theory and experiments prospect.*

Response: Thanks for your kind suggestion to improve the quality of this work. As pointed out by the astute reviewer, the pressure difference is the key to Cl₂ migration to gas chamber. Therefore, we would like to give more detailed discussion on the presence, function and stability of pressure difference as follows:

i) The presence/function of pressure difference can be confirmed by both theoretical simulations and experimental phenomenon of Cl₂ evolution. Theoretically, our COMSOL simulation based on Bernoulli's principle (Figure 5c-d) shows the pressure difference in the range of 0 ~ 116.3 mPa with the carrier flow of 0~80 mL min⁻¹, which will drive as-generated Cl₂ migration to gas chamber. This prediction is well aligned with experimental results: by tuning the carrier flow rate from 0 to 80 mL min⁻¹, the Faradaic efficiencies of Cl₂ gas products increases from 18.5% to 82.1% (Figure 3c).

Moreover, mechanism study has been conducted by *operando* Raman spectra for chlorine evolution reaction (CER) with/without interfacial pressure difference (Figure 5a and supplementary Figure 57). The decreased *OCl peak signal unambiguously confirm the role of interfacial pressure difference, *i.e.*, Cl₂ produced at the electrode/electrolyte interfaces intermediately taken away by gas flow following Bernoulli's principle.

ii) Next, the stability of pressure difference can be revealed by stability test of chlorine evolution reaction (CER) with pressure difference. Ti-MOF electrode exhibits strong stability in universal pH conditions (pH=1, 7, 13) with little morphology and structure decay after stability test (Figure 2h; Supplementary Figures 37–39). The prototype device shows long-term stability for Cl₂ electrosynthesis with little fluctuation for 200 hrs at a current density of 100 mA cm⁻² (Figure 3g). Even at extremely high current density of 1.14 A cm⁻², our system can still demonstrate excellent stability for 40 hrs (Supplementary Figure 52). All of above results confirm the pressure difference stable for Cl₂ electrosynthesis.

Based on above discussions, we have revised the manuscript and figures:

Figure 2. Catalyst and system design for Cl₂ electrosynthesis beyond Pourbaix diagram. h, Stability test at different pHs for Ti-MOF at 100 mA cm⁻².

Figure 3. Prototype device for Cl₂ electroynthesis. **c**, The Cl₂ Faradaic efficiencies change with gas flow rates. **g**, Stability test for Ti-MOF for 200 hours (at 100 mA cm⁻²). The arrows below the pompadour dots refer to flushing the electrodes every 20 hours.

Figure 5. Mechanism study. **a**, The operando Raman spectra. **c-d**, Gas pressure distribution from COMSOL simulations.

Supplementary Figure 37. Characterization of Ti-MOF on GDE after 50-hrs CER test in 5 M NaCl at pH=1. **a**, XRD. **b**, FTIR.

Supplementary Figure 38. Characterization of Ti-MOF on GDE after 50-hrs test in 5 M NaCl at pH=7. **a**, XRD. **b**, FTIR.

Supplementary Figure 39. Characterization of Ti-MOF on GDE after 50-hrs test in 5 M NaCl at pH=13. **a**, XRD. **b**, FTIR.

Supplementary Figure 52. Ti-MOF stability experiments at the current density of 1.14 A cm^{-2} .

Supplementary Figure 57. The *operando* Raman spectra without pressure difference. a, *operando* Raman spectra. b, Ti-*OCl Raman peak intensity.

Original Comment 7. *Some typos and errors have been noted, particularly in the Reference section and the fourth paragraph of Introduction section. A careful recheck of the whole manuscript is recommended*

Response: Thank you for the kind reminding. We have rechecked the whole manuscript, and fixed the typos/errors as follows:

“Page 2, Line 18: ... saving 6.75% (1.17 million dollar per year) as comparison to conventional chlor-alkali design.”

“Page 10, Line 210-211: ... while the use of membrane has accounted for additional production cost

of 6.75% (1.17 million dollar per year) in the membrane-based systems.”

“Page 13, Line 296-297: The Ti-MOF can still maintain its structure in pH=1~13 (Please see the blue area).”

“Page 14, Line 298-299: ... and the Ti-MOF can still maintain its durability in all pH ranges (as shown in the pompadour area).”

We would like to thank the Editor's invitation to revise the manuscript. We are also appreciated of the reviewer #4 for his/her constructive comments to improve the quality of this work.

After receiving the reviewer's comments, we have spent ten intensive weeks in order to answer the concerns raised by the astute reviewer. We believe we have shifted the work to the next level of proof and gained a much deeper understanding of the system. Please see the following point-by-point response to the reviewer #4's comments. Thank you and best wishes,

Kind regards

The authors

Response to Reviewer #4

Original Comment: *In this manuscript, the authors reported a membrane-free electrochemical system for Cl₂ synthesis with a titanium-based metal-organic framework (Ti-MOF) anode. Under the guidance of Bernoulli's principle, Cl₂ could permeate through the gas diffusion electrode and be directly separated from the anode by the carrier gas on the other side during the reaction. Although the Cl₂ separation strategy is of significance, the manuscript lacks sufficient evidences to support the conclusions. Meanwhile, the comparison of the activity data was not scientific. Therefore, this manuscript is not recommended for publication in Nature Communications.*

Original Comment 1. *The author's identification of the signal at 1.92 Å as the Ti-O scattering path was unconvincing without reference samples with Ti-O and Ti-C scattering pathways (such as TiO₂, TiC₂, etc.). In addition, the authors are required to clarify why the Ti-Ti coordination in Ti-MOF (2.89 Å) was significantly smaller than that in the reference Ti foil (3.11 Å).*

Response: Thanks for your kind reminding. We have added the data of XANES/EXAFS/XAFS for TiO₂/TiC₂/Ti-foil reference samples (Figure R1, Figures 2d-e, Supplementary Figure 9 and Supplementary Table 2), and given the response as follows:

- (1) Firstly, we confirm the signal at 1.92 Å as the Ti-O scattering path according to the following evidence: *i)* Based on the literature (*Coord Chem Rev* 2018, 359, 80), the length of the Ti-O bond is typically within the range of 1.8-2.0 Å angstroms., where our Ti-MOF system (1.92 Å) locates within this range; *ii)* The Ti-MOF EXAFS data in Figure 2f shows a Ti-O bond distance (1.92 Å) similar to TiO₂ (1.95 Å), indicating identical coordination environments. This result is consistent to Ti K-edge absorption energy of Ti-MOF (4962.95 eV, Figure 2d) lies between Ti-foil (4961.22 eV) and TiO₂ (4963.46 eV), and closer to the later, thus indicating the intermediate Ti valence state close Ti-O from TiO₂; *iii)* the TiC reference data reveals a Ti-C bond distance of 2.14 Å (Supplementary Table 2). This length is significantly longer than Ti-O bonds (1.92 Å), thereby precluding the Ti-C configuration within our Ti-MOF system.
- (2) Following the reviewer's comment, we have carefully rechecked the Ti-Ti distances for Ti-foil and Ti-MOF in the original manuscript. We sincerely apologize for the typos that cause confusion. Now, we have corrected the distance of Ti-Ti bond from Ti-foil as 2.89 Å, while Ti-MOF as 3.12 Å (Supplementary Table 2).

[FIGURE REDACTED]

Figure R1 Fourier transform obtained from the k^2 -weighted EXAFS oscillations $\chi(k)$ of Ti_2CT_x , Ti_2AlC , $Ti_3C_2T_x$, and Ti_3AlC_2 . (Coord. Chem. Rev. 2018, 359, 80; 2D Materials 2023, 10, 035024)

Figure 2. Catalyst and system design for Cl₂ electrosynthesis beyond Pourbaix diagram. d, X-ray absorption near edge structure (XANES) spectra of Ti-MOF, Ti-foil and TiO₂. e, Extended X-ray absorption fine structure (EXAFS) and fitting curves spectra of Ti-MOF, Ti-foil and TiO₂.

Supplementary Figure 9. Synchrotron radiation of TiC. a, XANES spectra of reference TiC. b, d EXAFS spectra of reference TiC and data fitting. c, the k-space XAFS data for edges of reference TiC.

Supplementary Table 2. Bond length and coordination of Ti-MOF-before and after calcination.

Material	shell	CN	R (Å)	ΔE_0 (eV)	σ^2 (10^{-3} \AA^2)	R factor
Ti-foil	Ti-Ti	6.24 ± 0.21	2.89 ± 0.002	-7.05 ± 0.18	0.006 ± 0.0005	0.0084 ± 0.00224
TiO ₂	Ti-C	4.15 ± 0.05	1.95 ± 0.003	-1.55 ± 0.26	0.006 ± 0.0001	0.014 ± 0.0012
	Ti-Ti	2.05 ± 0.05	3.06 ± 0.0004		0.0040 ± 0.00025	
TiC	Ti-O	6.03 ± 0.08	2.14 ± 0.00039	-3.47 ± 0.01	0.0002 ± 0.0001	0.0095 ± 0.00035
	Ti-Ti	11.93 ± 0.14	3.05 ± 0.0002		0.0009 ± 0.0001	
Ti-MOF	Ti-O	3.98 ± 0.04	1.90 ± 0.0005	-7.04 ± 0.10	0.016 ± 0.0001	$0.0034 \pm 4.1 \times 10^{-5}$

(before)	Ti-Ti	4.05±0.002	3.10±10 ⁻⁷		0.019±1.5×10 ⁻⁵	
Ti-MOF	Ti-O	3.36±0.11	1.92±0.002	-4.21±0.21	0.011±0.0009	0.013±0.0043
	Ti-Ti	4.05±0.09	3.12±0.00008		0.018±0.00064	

“Supplementary information, Line 719-724: **Supplementary note:** CN: coordination number; R: bond lengths between central atoms and surrounding coordination atoms; ΔE_0 : the difference of the zero kinetic energy value between the sample and theoretical model; σ^2 : Debye-Waller factor to account for both thermal and structural disorders; R factor is used to measure the goodness of the fitting. All the data in the figure are the average results after five repeated fittings, and the standard deviation of each average value is added to indicate the relative deviation of the value.”

“Page 6, Line 98-104: The Ti-MOF has presented XANES absorption edge energy between Ti-foil and TiO₂ (4962.95 vs. 4961.22 and 4963.46 eV), indicating its valence state of Ti in the range of 0 ~ +4. Different from Ti foil bearing only Ti-Ti coordination (distance: 2.89 Å), the Fourier transform EXAFS spectrum of Ti-MOF shows featured shells of Ti-Ti (3.12 Å) and Ti-O (1.92 Å) coordination. The Ti-Ti distance (3.12 Å) corresponds to interlayer interaction between Ti atoms in the form of multi-nuclear titanium clusters.³¹ The Ti-O distance (1.92 Å) is significantly shorter as comparison to Ti-C bond distance (2.14 Å, Supplementary Figures 9–10).”

“Supplementary information, Line 228-231: The additional TiC reference data, as presented in Supplementary Figure 9 and Supplementary Table 2, reveal a Ti-C bond distance of 2.14 Å. This length is significantly longer than Ti-O bonds (1.92 Å), thereby precluding the inclusion of Ti-C configurations within the structural framework of our material system.²¹”

Manuscript Ref 31. Sun, Y. et al. Large Titanium-Oxo Clusters as Precursors to Synthesize the Single Crystals of Ti-MOFs. *ACS Mater. Lett.* 3, 64-68 (2020).

Supplementary information Ref 21. Näslund, L.-Å. Magnuson, M. The origin of Ti 1s XANES main edge shifts and EXAFS oscillations in the energy storage materials Ti₂CT_x and Ti₃C₂T_x MXenes. *2D Mater* 10, 035024 (2023).

Original Comment 2. Line 90, the authors mentioned the well-defined crystallinity of Ti-MOF nanosheet, which was inconsistent with an amorphous structure revealed by the TEM image in Figure 2a. Moreover, the authors are suggested to index the sharp diffraction peaks (25-30 degree) in the XRD pattern (Figure 2b).

Response: Thanks for your constructive comment. We have given the response as follows:

(1) Firstly, we have consulted with TEM expert in Analytic Centre about the as-observed “amorphous structure” of Ti-MOF, and understand it as a common phenomenon for MOF-based materials. *Nat Mater* 2017, 16, 532-536; *Commun Chem* 2020, 3, 99⁹⁹ Owing to its organic ligand structure, MOF materials can be easily damaged by electron beams of TEM at high resolution with applied high voltage (up to 200 kV). Therefore, it is very difficult to get well-defined crystal lattice fringes in MOF materials.

With above understanding in mind, we have repeated TEM characterization for Ti-MOF and reprocessed the TEM image. As shown in Figure 2a, the inset (upper right corner) now displays lattice fringes with improved clarity, and the as-measured lattice spacing is 0.21 nm. Further, in order to enhance the credibility of the good crystallinity of Ti-MOF, the analysis results of dislocations after FTT processing and the specific data of the measured crystal plane spacings are also presented (Supplementary Figure 3).

Commonly, the crystallinity of MOF materials has been determined by XRD. *Heliyon* 2024, 10, e23840. *Nat Protoc* 2022, 17, 2389. For our Ti-MOF sample, it demonstrated well-defined diffraction peaks in the range of 5~60 °C, indicating it good crystallinity.

(2) Secondly, following the reviewer’s comment, we plan to identify each XRD peak by comparison with theoretical simulation ones. However, Ti-MOF is a polycrystalline material with multiple crystal planes, and some simulated diffraction peaks show weak intensities. *ACS Mater Lett* 2021, 3, 64; *Crystal Growth Des* 2023, 23, 3778⁷⁸ To more clearly illustrate these weak peaks, we have amplified relevant areas (as shown in Figure 2b). Now in the simulated XRD profile, all of the peak positions are clearly visible.

Subsequently, we have compared each experimental XRD peak with theoretically simulation patterns. As shown in the Supplementary Table 1, all of the indexed values match well, providing strong evidence for the successful synthesis of desired Ti-MOF phase.

Based on above discussions, the following sections have been added/revised in updated

manuscript:

Figure 2. Catalyst and system design for Cl₂ electro-synthesis beyond Pourbaix diagram. a, TEM and enlarge the image of Ti-MOF catalyst. The blue wavy line form is the lattice spacing after FFT. **b,** X-ray diffractometer (XRD) of Ti-MOF (In order to make the peaks ranging from 25 to 60° in the theoretical simulation more clearly, they were enlarged by a factor of 10).

Supplementary Figure 3. The dislocation analyses for measuring crystal plane spacing of Ti-MOF.

Supplementary Table 1. The crystal plane indices of Ti-MOF and the simulated value.

(h k l) (Simulation)	2θ (Simulation)	2θ (Experiment)
(1 0 0)	7.434	7.49
(1 0 1)	10.826	10.6
(2 0 1)	14.665	15
(2 1 0)	16.171	16
(0-1 1)	19.759	19.3

(0 1 2)	25.79	25.9
(3 2 0)	29.104	29.5
(4 2 0)	32.676	32.5
(0 2 3)	43.289	43.3

“Page 5, Line 87-91: Close examination of a Ti-MOF nanosheet reveals its well-defined crystallinity with the adjacent lattice spacing of 0.21 nm (Figure 2a, Supplementary Figure 3), which agrees well with X-ray diffraction (XRD) containing a series of characteristic peaks corresponding to metal-organic framework architectures with dominant (2 1 0) crystal face (Figure 2b, Supplementary Table 1).”

Original Comment 3. *The FTIR identification of organic ligands in Figure 2c should be confirmed by reference samples.*

Response: Thanks for your kind reminding. According to experimental synthesis procedure, Ti-MOF is derived from 2,6 naphthoic acid (H₂NDC) ligand and 2,6 naphthoic 2Na, so we have performed comparison FTIR for these organic ligands (Supplementary Figure 4).

The characteristic vibrational peaks of H₂NDC at 1668 and 1415 cm⁻¹ have presented in Ti-MOF. No detectable peaks appear in the fingerprint region (500-1000 cm⁻²) resembling the intense bands in Figure 2c (attributed to Ti-O vibrations). *J Am Chem Soc* 2021, 143, 1107 Further, the FTIR spectrum of 2,6 naphthoic 2Na further exhibits a sharp, intense peak at 786 cm⁻¹, providing spectroscopic evidence for successful deprotonation *via* Na-mediated hydrogen abstraction. *J Am Chem Soc* 2009, 131, 10857 These results collectively confirm effective coordination between Ti centers and carboxylate groups in the synthesized Ti-MOF.

Based on above discussions, relevant Figures/sentences have been added into updated manuscript:

Supplementary Figure 4. Fourier Transform infrared spectroscopy (FTIR) of 2,6 naphthoic acid and 2,6 naphthoic 2Na.

Page 5, Line 92-94: ...showing characteristic vibrations of 1400-1630 cm^{-1} of naphthalene ring from the organic ligand (Supplementary Figures 4–6) and the vibrational peak at 785 cm^{-1} from Ti-O bonding.^{26,27}

Supplementary information, Line 183-188: The characteristic vibrational peaks of H₂NDC at 1668 and 1415 cm^{-1} remain present in Ti-MOF. Notably, no detectable peaks appear in the fingerprint region (500-1000 cm^{-1}) resembling the intense bands in Figure 2c of the main text (attributed to Ti-O vibrations).⁶ The FTIR spectrum of disodium 2,6 naphthoic 2Na further exhibits a sharp, intense peak at 786 cm^{-1} , providing spectroscopic evidence for sodium-mediated deprotonation *via* hydrogen atom transfer.¹⁸ Collectively, these results confirm effective coordination between Ti centers and carboxylate groups in the synthesized MOF.”

Manuscript Ref 27. Feng, X. et al. Rational Construction of an Artificial Binuclear Copper Monooxygenase in a Metal-Organic Framework. *J. Am. Chem. Soc.* **143**, 1107–1118 (2021).

Supplementary information Ref 6. Feng, X. et al. Rational Construction of an Artificial Binuclear Copper Monooxygenase in a Metal-Organic Framework. *J. Am. Chem. Soc.* **143**, 1107–1118 (2021).

Supplementary information Ref 18. Dan-Hardi, M. et al. A New Photoactive Crystalline Highly Porous Titanium (IV) Dicarboxylate. *J. Am. Chem. Soc.* **131**, 10857–10859 (2009).

Original Comment 4. *Line 108-113, Page 6, the final structure of Ti-MOF did not align with the provided evidence. The coordination number of Ti-O was 3.183, inconsistent with the metal-oxygen-layers of a hexacoordinated TiO₆ units.*

Response: We are sorry for not explaining it clearly. “The metal-oxygen-layers of a hexacoordinated TiO₆ units” illustrates one Ti atom can theoretically coordinated with six other atoms in the form of hexacoordinated architecture, which represents the global symmetry constraints for Ti atoms. On the other hand, the coordination number of Ti-O in Ti-MOF has been experimentally determined by XAS (Figures 2d, e, Supplementary Figures 7–8), where the average number is 3.98 (before low-temperature calcination) or 3.36 (final structure after calcination). To clarify the relationship between the two, we have built a structural model for Ti-MOF as follows (Figure R2):

As shown in Figure R2, the spatial configuration within the Ti-O cluster is in the form of octahedra containing of 16 Ti atoms and 96 O atoms with a ratio of 1/6 (Figure R2). However, when counting the actual coordination number of Ti-O from XAS, the contribution of each atom has to be taken into account, for example, the coordination of O atoms would be considered according to their local environment, which is 1 (internal O atom) or 0.5 (surface O atom). We can see that 48 O atoms are internal and 48 O atoms are on the surface. Accordingly, the number of O atoms coordinated to Ti is calculated as: $48 + 48 \times 0.5 = 72$, resulting in a Ti/O ratio of 1:4. This result is close to our experimentally XAS measured value (3.98).

Further, to improve its crystallinity, the Ti-MOF has been subject to calcination at 450 °C for 3 hrs. Such a calcination treatment has not altered Ti-MOF architecture, which is still in the form of octahedra, as confirmed by TEM (Figure 2a), XRD (Figure 2b), FT-IR (Figure 2c). While X-ray absorption near-side structure (XANES) confirms the coordination number of Ti-O has decreased to 3.36 (Supplementary table 2). This phenomenon indicates the evaporation of some organic ligands during calcination process, which can expose more Ti sites for electrochemical reaction.

Figure R2. The periodic structure of Ti-MOF-before.

Supplementary Table 2. Bond length and coordination of Ti-MOF-before and after calcination.

Material	shell	CN	R (Å)	ΔE_0 (eV)	σ^2 (10^{-3} \AA^2)	R factor
Ti-foil	Ti-Ti	6.24 ± 0.21	2.89 ± 0.002	-7.05 ± 0.18	0.006 ± 0.0005	0.0084 ± 0.00224
TiO ₂	Ti-C	4.15 ± 0.05	1.95 ± 0.003	-1.55 ± 0.26	0.006 ± 0.0001	0.014 ± 0.0012
	Ti-Ti	2.05 ± 0.05	3.06 ± 0.0004		0.0040 ± 0.00025	
TiC	Ti-O	6.03 ± 0.08	2.14 ± 0.00039	-3.47 ± 0.01	0.0002 ± 0.0001	0.0095 ± 0.00035
	Ti-Ti	11.93 ± 0.14	3.05 ± 0.0002		0.0009 ± 0.0001	
Ti-MOF (before)	Ti-O	3.98 ± 0.04	1.90 ± 0.0005	-7.04 ± 0.10	0.016 ± 0.0001	$0.0034 \pm 4.1 \times 10^{-5}$
	Ti-Ti	4.05 ± 0.002	3.10 ± 10^{-7}		$0.019 \pm 1.5 \times 10^{-5}$	
Ti-MOF	Ti-O	3.36 ± 0.11	1.92 ± 0.002	-4.21 ± 0.21	0.011 ± 0.0009	0.013 ± 0.0043

	Ti-Ti	4.05±0.09	3.12±0.00008		0.018±0.00064	
--	-------	-----------	--------------	--	---------------	--

“Page 6-7, Line 111-117: Ti atom can theoretically coordinate with six other atoms in the form of hexacoordinated architecture, which represents the global symmetry constraints for Ti atoms. On the other hand, the coordination number of Ti-O in Ti-MOF has been experimentally determined by XAS (Figures 2d, e, Supplementary Figures 7–8), where the average number is 3.98 (before low-temperature calcination) or 3.36 (final structure after calcination). This phenomenon indicates the evaporation of some organic ligands during calcination process, which can expose more Ti active sites for electrochemical reaction.”

Original Comment 5. *The authors were required to provide the complete calculation process of the pressure difference at the triple-phase interface in the main text, as well as the revised triple-phase Bernoulli formula.*

Response: Following the reviewer’s comment, the following calculation formula and details have been added into updated manuscript.

Page 18-19, line 388-409:

Computational details of COMSOL

Based on Bernoulli’s principle for fluid dynamics, the local pressure of gas flowing through a conduit decreases as its flow velocity increases.⁴¹ Consequently, the gas pressure within the backside region of the electrode in the gas chamber diminishes with elevated flow rates. This hydrodynamic relationship is further operationalized in our simulations through Darcy law:

$$u = -\frac{\kappa}{\mu} \nabla p$$

which provides a concrete framework for modeling such systems. Given that the gas-liquid flow within the chambers can be regarded as laminar, Darcy law offers an appropriate theoretical foundation for these calculations.

Theoretically, Bernoulli principle a basic rule in fluid dynamics illustrating the relationship between the surface velocity and pressure. In the present work, two fluids have presented at the three-

phase boundary, carrier gas flow and electrolyte flow, and according to Bernoulli principle that can be illustrated as:

$$P_1 + \frac{1}{2}\rho_1V_1^2 + \rho_1gh_1 = P_2 + \frac{1}{2}\rho_2V_2^2 + \rho_2gh_2$$

where 1 represents carrier gas flow and 2 represents electrolyte flow. So, the interfacial pressure difference (∇p) can be illustrated as:

$$\nabla p = \frac{1}{2}\rho_1V_1^2 + \rho_1gh_1 - \frac{1}{2}\rho_2V_2^2 - \rho_2gh_2$$

Quantitatively, the pressure difference has been calculated by COMSOL software (Figures 5c-d, Supplementary Video S1), showing the pressure difference in the range of $0 \sim 116.3 \times 10^{-3}$ Pa with the carrier gas flow of $0 \sim 80$ mL min^{-1} and electrolyte flow of 8.4 mL min^{-1} . Therefore, when high-velocity carrier gas flow and low electrolyte flow have been applied to three-phase boundary, it will generate a large localized pressure gradient. This can promote Cl_2 bubbles migration away from electrolyte flow, and into carrier gas (Please see Supplementary Video S2).”

Ref 41. Singh, M. R., Clark, E. L., Bell, A. T. Effects of electrolyte, catalyst, and membrane composition and operating conditions on the performance of solar-driven electrochemical reduction of carbon dioxide. *Phys. Chem. Chem. Phys.* **17**, 18924–18936 (2015).

Original Comment 6. *The applied potentials for all the activity experiments in the three-electrode system were referenced against the Ag/AgCl and had not been converted to the potentials relative to the more commonly used reversible hydrogen electrode (RHE). Therefore, it was unreasonable to investigate the effect of pH variation on activity at the same potential (vs. Ag/AgCl), as the potential (vs. RHE) would change with pH under these conditions.*

Response: Thanks for your kind reminding. We have converted reference potentials against reversible hydrogen electrode (RHE) based on the following equation: ^{Langmuir 2024, 40, 7632}

$$E_{RHE} = E_{Ag/AgCl} + 0.059 \times pH + E_{Ag/AgCl}^{\circ}$$

Where $E_{Ag/AgCl}^{\circ}$ is the standard electrode potential of Ag/AgCl, which is approximately 0.197 V in a saturated KCl solution.

Further, we have conducted linear sweep voltammetry (LSV), and plotted these data versus RHE

(Supplementary Figures 14,32). We have further determined the overpotentials at different pHs (148, 156 and 404 mV for pH=1, 7 and 13) according to LSV. *ChemElectroChem* 7, 1448-1455 (2020), *Nature* 617, 519-523 (2023) When compared with literature (Supplementary Table 3), our system demonstrates competitive overpotentials among all non-noble metals which is even comparable to some noble metal benchmarks.

Supplementary Figure 32. Electrochemical performance data of Ti-MOF by potential conversion to reversible hydrogen electrode (RHE). a-c, LSV. d-f, Faradaic efficiency and gas production rate.

Supplementary Figure 47. Electrochemical data of Ti-MOF and DSA after potential conversion (vs. RHE). a, Faradaic efficiency. b, yield rate.

“Page 7, Line 135-136: To facilitate comparison across diverse pH conditions, all of the potentials have been converted to RHE (Supplementary Figure 32 and Supplementary Table 3).”

“Page 9, Line 179-180: ...which shows minor alternations in Cl₂ Faradaic efficiencies (94% vs. 92%) and yield rates (0.93 vs. 0.9 mmol h⁻¹ cm⁻² Figures 3d, 3e and Supplementary Figure 47).”

Original Comment 7. *The authors compared the Cl₂ yield rate of Ti-MOF with that of the most recently reported catalysts. However, when achieving nearly similar Cl₂ generation rates, Ti-MOF required a higher current density and exhibited a lower Faraday efficiency, which did not meet the claimed “record activities”. Note that the overpotential at 10 mA cm⁻² or 100 mA cm⁻² is a common descriptor for chlorine evolution activity. The authors should provide the activity data of the electrocatalysts in the text and conduct a systematic comparison with the state-of-the-art materials, especially under different pH conditions.*

Response: Thanks for your constructive comment. We have given the response as follows:

(1) Firstly, by using “record activities” in the original manuscript, we refer to the Ti-MOF demonstrating the highest Cl₂ yield rate (18.675 mmol h⁻¹ cm⁻²) ever reported, which surpasses all of the previous studies such as Nature 2023, 617, 519; Nat. Commun. 2023, 14, 2475; Angew. Chem., Int. Ed. 2024, 63, e202406273.

Indeed, as pointed out by the reviewer, we noted the system exhibits relatively low Faraday efficiency (87.6%) at high current density (1.14 A cm⁻²). This is because of its non-precious metal nature, which cannot promote chlorine evolution reaction as efficiently as noble metal catalysts (like Ru-O₄ SAM from Nat. Commun. 2023, 14, 2475).

Following the reviewer’s comment, we have removed the term “record activities” from title and other sections

(2) Following the reviewer’s suggestion, we have determined the overpotentials at 10 mA cm⁻² for our Ti-MOF based systems (Supplementary Figure 32), which are 148 mV (pH=1), 156 mV (pH=7) and 404 mV (pH=13), respectively.

By comparing with the state-of-the-art literature in Supplementary Table 3, we can see almost all of the previous studies have operated in strong acid electrolyte (pH <3), and used membrane-

based systems and noble-metal catalysts. Under such harsh testing condition, the overpotentials for most previous studies (at 10 mA cm^{-2}) are still in the range of 30~220 mV, which is comparable to our system with membrane-free configuration and non-precious metal catalyst (148 mV at pH=1; 156 mV at pH=7).

To the best of our knowledge, there is no study operating in strong alkaline condition (pH = 13) because of the Pourbaix diagram limitation. It is common concept that no Cl_2 can evolve from alkaline electrochemical system because of chemical reaction between Cl_2 and alkaline ($\text{Cl}_2 + 2\text{OH}^- \rightarrow 2\text{Cl}^- + \text{H}_2\text{O}$). In this work, our system can operate at strong alkaline condition (pH = 13) for chlorine evolution with overpotentials comparable to some previously reported studied in acidic condition (Energy Environ. Sci. 2019, 12, 1241; Chem. Eng. J. 2021, 421, 127785).

Based on above discussions, the relevant figures and tables have been added/revised in the updated manuscript:

Supplementary Figure 32. Electrochemical performance data of Ti-MOF by potential conversion to reversible hydrogen electrode (RHE). **a-c**, LSV. **d-f**, Faradaic efficiency and gas production rate.

Supplementary Table 3. The comparison of the overpotential (10 mA cm^{-2}) of CER.

Electrocatalysts	Overpotential (mV)	Reaction conditions	References
------------------	--------------------	---------------------	------------

RuO _x /2D TiO _x	90	1 M pH=1	29
Ru-O ₄	30	1 M pH=1	30
CoO _x Cl _y	100	0.5 M pH=2	31
RCON-H	89 (1 A cm ⁻²)	5 M pH=2	11
MoO _x @IrO ₂ -Ta ₂ O ₅	30	4 M pH=2	32
Pt ₁ /p-NC@CNTs	40	1M 0.1 HClO ₄	33
Ir ₁ O ₆	71.8	4 M pH=2	7
RuO ₂ /TiO ₂	220	4 M pH=3	34
Ni(Co/Mn)Sb ₂ O _x	450	4 M pH=2	24
Co ₃ O ₄ /FTO	200	Saturated pH=3	35
Ti _{0.35} V _{0.35} Sn _{0.25} Sb _{0.05} -oxide	987	5 M NaCl pH≈2	25
Ti-MOF	148	5 M NaCl pH=1	This work
	156	5 M NaCl pH=7	
	404	5 M NaCl pH=13	

“Title (Page 1 line 1): Bernoulli’s principle-mediated Cl₂ electrosynthesis”

“Page 7, Line 135-136: To facilitate comparison across diverse pH conditions, all of the potentials have been converted to RHE (Supplementary Figure 32 and Supplementary Table 3).”

Ref 7. Wang, J. et al. Engineering the Coordination Environment of Ir Single Atoms with Surface Titanium Oxide Amorphization for Superior Chlorine Evolution Reaction. *J. Am. Chem. Soc.* **146**, 11152–11163 (2024).

Ref 24. Moreno-Hernandez, I. A., Brunschwig, B. S., Lewis, N. S. Crystalline nickel, cobalt, and manganese antimonates as electrocatalysts for the chlorine evolution reaction. *Energy Environ. Sci.* **12**, 1241–1248 (2019).

Ref 25. Mirzaei Alavijeh, M. et al. A selective and efficient precious metal-free electrocatalyst for

chlorine evolution reaction: An experimental and computational study. *Chem. Eng. J.* **421**, 127785 (2021).

Ref 29. Ji, J. et al. Ruthenium Oxide Clusters Immobilized in Cationic Vacancies of 2D Titanium Oxide for Chlorine Evolution Reaction. *Small Struct.* **5**, 2300240 (2023).

Ref 30. Liu, Y. et al. Electrosynthesis of chlorine from seawater-like solution through single-atom catalysts. *Nat. Commun.* **14**, 2475 (2023).

Ref 31. Xiao, M. et al. Self-adaptive amorphous CoO_xCl_y electrocatalyst for sustainable chlorine evolution in acidic brine. *Nat. Commun.* **14**, 5356 (2023).

Ref 32. Zhan, Y. et al. Efficient electrocatalytic chlorine evolution of MoO_x modified $\text{IrO}_2\text{-Ta}_2\text{O}_5$ and its application in the seawater electrolysis. *J. Electroanal. Chem.* **992**, 119283 (2025).

Ref 33. Quan, L. et al. Atomic Pt-N₄ Sites in Porous N-Doped Nanocarbons for Enhanced On-Site Chlorination Coupled with H₂ Evolution in Acidic Water. *Adv. Funct. Mater.* **33**, 2307643 (2023).

Ref 34. Menzel, N., Ortel, E., Mette, K., Kraehnert, R., Strasser, P. Dimensionally Stable Ru/Ir/TiO₂-Anodes with Tailored Mesoporosity for Efficient Electrochemical Chlorine Evolution. *ACS Catal.* **3**, 1324–1333 (2013).

Ref 35. Zhu, X. et al. Co₃O₄ nanobelt arrays assembled with ultrathin nanosheets as highly efficient and stable electrocatalysts for the chlorine evolution reaction. *J. Mater. Chem. A* **6**, 12718–12723 (2018).

Supplementary information, line 365-399:

According to the standard electrochemical principle, the conversion relationship between the potential of Ag/AgCl electrode (vs Ag/AgCl) and RHE is (at 25 °C):

$$E_{RHE} = E_{Ag/AgCl} + 0.059 \times pH + E_{Ag/AgCl}^0$$

Here, $E_{Ag/AgCl}^0$ is the standard electrode potential of Ag/AgCl, which is approximately 0.197 V in a saturated KCl solution. From this equation, the potential at each pH can be calculated.

Equation $2\text{Cl}^- - 2e^- \rightarrow \text{Cl}_2$, $E^0=1.36$ V vs. RHE is the standard electrode potential measured at

pH = 0. However, in the actual situation of this work, further calculations are required:

$$E_{eq} = E^o + \frac{RT}{nF} \ln \frac{[p_{Cl_2}^\theta]}{[Cl^-]^2} + E_{Ag/AgCl}^o$$

In the equation, R is gas constant ($8.314 \text{ J mol}^{-1} \text{ K}^{-1}$), T : absolute temperature (K), n : number of transferred electrons, F : Faraday constant (96485 C mol^{-1}), $[p_{Cl_2}^\theta]$: partial pressure of chlorine gas (is the constant 1 bar), $[Cl^-]^2$: chloride ion concentration (5 mol L^{-1}) and the $E_{Ag/AgCl}^o$ is the standard electrode potential of the reference electrode (is the constant 0.197 V) Substituting into the formula and calculating, we obtain $E_{eq}=1.3186 \text{ V vs. SHE}$. And by converting the SHE and RHE potentials, we can obtain:

$$E_{RHE} = E_{SHE} + 0.059 \times pH$$

The standard electrode potentials at pH values of 1, 7 and 13 can be calculated:

$$E_{RHE} = 1.3186 + 0.059 \times 1 = 1.3777 \text{ V vs RHE (pH = 1)};$$

$$E_{RHE} = 1.3186 + 0.059 \times 7 = 1.7316 \text{ V vs RHE (pH = 7)}$$

$$E_{RHE} = 1.3186 + 0.059 \times 13 = 2.0856 \text{ V vs RHE (pH = 13)}.$$

From above equations, we can calculate the overpotential of Ti-MOF at each pH value. According to the formula:

$$\eta = E_{measured} - E_{theoretical}$$

It can be calculated that: $\eta_{pH=1} = 1.5259 - 1.3777 = 0.1482 \text{ V vs RHE}$; $\eta_{pH=7} = 1.888 - 1.7316 = 0.1482 \text{ V vs RHE}$, $\eta_{pH=13} = 2.49 - 2.0856 = 0.404 \text{ V vs RHE}$.

By comparing with the state-of-the-art literature in Supplementary Table 3, we can see almost all of the previous studies have operated in strong acid electrolyte (pH < 3), and used membrane-based systems and noble-metal catalysts. Under such harsh testing condition, the overpotentials for most previous studies (at 10 mA cm^{-2}) are still in the range of 30~220 mV, which is comparable to our system with membrane-free configuration and non-precious metal catalyst (148 mV at pH=1; 156 mV at pH=7).

To the best of our knowledge, there is no study operating in strong alkaline condition (pH = 13) because of the Pourbaix diagram limitation. It is common concept that no Cl_2 can evolve from alkaline electrochemical system because of chemical reaction between Cl_2 and alkaline ($Cl_2 + 2OH^- \rightarrow 2Cl^-$

+ H₂O). In this work, our system can operate at strong alkaline condition (pH = 13) for chlorine evolution with overpotentials comparable to some previously reported studied in acidic condition.^{24,25}

Ref 24. Moreno-Hernandez, I. A., Brunschwig, B. S., Lewis, N. S. Crystalline nickel, cobalt, and manganese antimonates as electrocatalysts for the chlorine evolution reaction. *Energy Environ. Sci.* **12**, 1241–1248 (2019).

Ref 25. Mirzaei Alavijeh, M. et al. A selective and efficient precious metal-free electrocatalyst for chlorine evolution reaction: An experimental and computational study. *Chem. Eng. J.* **421**, 127785 (2021).

Original Comment 8. *The authors claimed to observe Raman signals of Ti-*OCl in the operando Raman spectra, but the signal-to-noise ratio of the provided Raman spectra is too low to substantiate their conclusions.*

Response: Thanks for your kind reminding. We have re-tested the *operando* Raman data for five times and provide the responses as follows:

Generally, Raman spectra is a scattering signal with intrinsic weak intensity. When used to study the structural characteristics of a bulk material, obvious signal peaks can be seen. This is due to the structural crystal lattice that contributes to overall vibration of the bulk material, leading to enhanced scattered signals.

While in this work, the *operando* Raman is used to probe the transient surface-adsorbed species (*OCl intermediates) during electrocatalysis. The scattered Raman signals are mainly focused on bond vibrations of adsorbed species on catalyst surfaces, which are known to be very weak as comparison to structural crystal lattices in bulk materials. Consequently, the *operando* Raman signals for catalytic reactions often demonstrate characteristically low signal-to-noise ratios in the literatures (like ORR, NRR, CRR). *Nature Commun* 2024 15, 4157. *Angew Chem Int Ed* 2025 64, e202414202. *J Am Chem Soc* 2024 146, 11152-

¹¹¹⁶³ Our *operando* Raman signals are comparable to the above literature.

To confirm the accuracy of the experimental results, we performed five *operando* Raman tests. We directly determined the positions and intensities of the peaks using Raman spectrometer software, and updated the *operando* Raman data with error bars (Figure 5a-b and Supplementary Figure 57 in

updated manuscript). By comparing the Raman vibration at different electrolysis potentials, we conclude the characteristic *OCl Raman peak at approximately 1093 cm^{-1} (inside the blue transparent frame), and the intensity of *OCl peak changes consistently with applied potentials, reaching maximum intensity at 2.0 V vs Ag/AgCl (Figure 5b). Further, the data demonstrate significantly greater accumulation of the *OCl intermediate under no pressure condition. This observation suggests that the Bernoulli principle leveraged in our reaction system generates an immediate pressure gradient, enabling continuous removal of Cl_2 product and thereby enhancing chlorine evolution kinetics.

Based on above results, relevant figure/sentence has been revised/added into updated manuscript:

Figure 5. Mechanism study. a, The *operando* Raman spectra. b, Ti-*OCl Raman peak intensity.

Supplementary Figure 57. The *operando* Raman spectra without pressure difference. a,

operando Raman spectra. **b**, Ti-*OCl Raman peak intensity.

“Page 11, line 236-237): The *operando* Raman spectra has displayed a subset of vibrational data: the band at 1601 cm^{-1} is assigned to the C=C stretching vibration of the benzene ring, while the band at 1374 cm^{-1} corresponds to the symmetric stretching vibration of the carboxylate group ($-\text{COO}^-$) in the MOF substrate.; the band at 3400 cm^{-1} from H_2O ; the potential-dependent band at 1093 cm^{-1} from the Cl-O stretching mode of *OCl during CER process.³³”

“Supplementary information, line 530-550):

Generally, Raman spectra is a scattering signal with intrinsic weak intensity. When used to study the structural characteristics of a bulk material, obvious signal peaks can be seen. This is due to the structural crystal lattice that contributes to overall vibration of the bulk material, leading to enhanced scattered signals.

While in this work, the *operando* Raman is used to probe the transient surface-adsorbed species (*OCl intermediates) during electrocatalysis. The scattered Raman signals are mainly focused on bond vibrations of adsorbed species on catalyst surfaces, which are known to be very weak as comparison to structural crystal lattices in bulk materials. Consequently, the *operando* Raman signals for catalytic reactions often demonstrate characteristically low signal-to-noise ratios in the literatures (like ORR, NRR, CRR).^{7,19,28} Our *operando* Raman signals are comparable to the above literature.

To confirm the accuracy of the experimental results, we performed five *operando* Raman tests. We directly determined the positions and intensities of the peaks using Raman spectrometer software, and updated the *operando* Raman data with error bars (Figures 5a-b and Supplementary Figure 57). By comparing the Raman vibration at different electrolysis potentials, we conclude the characteristic *OCl Raman peak at approximately 1093 cm^{-1} (inside the blue transparent frame), and the intensity of *OCl peak changes consistently with applied potentials, reaching maximum intensity at 2.0 V vs Ag/AgCl (Figure 5b). Further, the data demonstrate significantly greater accumulation of the *OCl intermediate under zero pressure differential conditions. This observation suggests that the Bernoulli principle leveraged in our reaction system generates an immediate pressure gradient, enabling continuous removal of Cl_2 product and thereby enhancing chlorine evolution kinetics. All of these results provide the validity of our *operando* synthesis condition.”

Ref 7. Huang, Q. et al. Single-zinc vacancy unlocks high-rate H₂O₂ electrosynthesis from mixed dioxygen beyond Le Chatelier principle. *Nat. Commun.* **15**, 4157 (2024).

Ref 19. Quan, L., Zhao, X., Yang, L. M., You, B., Xia, B. Y. Intrinsic Activity Identification of Noble Metal Single-Sites for Electrocatalytic Chlorine Evolution. *Angew. Chem., Int. Ed.* **64**, 202414202 (2024).

Ref 28. Wang, J. et al. Engineering the Coordination Environment of Ir Single Atoms with Surface Titanium Oxide Amorphization for Superior Chlorine Evolution Reaction. *J. Am. Chem. Soc.* **146**, 11152–11163 (2024).

Original Comment 9. *The model constructed in DFT calculations demonstrated the atomically dispersed Ti atoms. The authors need to further clarify whether Ti-MOF was a single-atom catalyst, as there was no relevant evidence or description in the main text. Furthermore, the author constructed a cluster model terminated by sodium ions without considering periodic boundary conditions, which seems to be an unreasonable approach, as both the XRD and HRTEM results demonstrated the periodic crystalline structure of Ti-MOF.*

Response: Thanks for your constructive comment. We have given the response as follows:

- (1) Whether Ti-MOF is a single-atom catalyst. We consider Ti-MOF not a single-atom catalyst. Yes, as pointed out by the reviewer, Ti-MOF is composed of Ti nodes coordinated with organic ligands (2,6 naphthoic), where Ti atoms have been dispersed atomically. However, Ti-MOF has shown many different characteristics from traditional single-atom catalysts, including, periodic crystalline structure (XRD, Figure 2b), ordered TiO₆ unit cell (HRTEM and XAS, Figure 2a, d-e), and rich organic ligands (FTIR, Figure 2c). Therefore, it has usually classified as MOF material rather than single-atom catalyst.
- (2) Reasons for using cluster model.

Firstly, and most importantly, the aim of DFT calculations is to explain the chlorine evolution reaction (CER) mechanism on Ti-MOF electrode. Based on our calculation results from cluster model (Figures 5e-i), the detailed CER mechanism has been successfully unravelled, *i.e.*, CER pathway by calculating Gibbs free energy change of reaction intermediates, as well as

Pourbaix diagrams showing the surface adsorption changes with applied potentials at different pHs. Further, we have unraveled the high CER selectivity by calculating Gibbs free energy change of side OER process. Therefore, a cluster model is sufficient for our study.

Secondly, as with many other reports, another reason for using cluster model is limited computational resource. In this study, we have focused on a representative portion of Ti-MOF material, and aimed to investigate the mechanism, particularly reaction intermediate and pathways. Actually in the literature, there are enormous state-of-the-art studies employing a cluster model for MOF-based materials: *i*) The cluster model of Au₆ combined with Zr₆O₈ was built to study the formation of CH₄ by carbon dioxide reduction (Figure R1) (*Angew Chem Int Ed* 2025, e202500269); *ii*) The cluster model of cMOF combined with Cu₂O was established to conduct a series of theoretical calculations (Figure R2). The combination of the calculation results and the experimental results verified the reaction path and explained the material advantages after the combination of cMOF (100) and Cu₂O (*Nano Energy* 2025,141, 111077); *iii*) Due to the large scale of the modeling system, the author also extracted a part of MOF for study (Figure R3). The structural data such as bond length and bond angle are studied, and the properties of the MIL-100(Fe) small cluster structure and its adsorption capacity for NO are studied more thoroughly (*Adv Mater* 2024, 36, 2403053); *iv*) due to the large size of the MAF-203 system, the authors adopted a smaller cluster model to simulate the molecular dynamics process of MAF-203 (Figure R4). Detailed data were given for the positions of atoms and bond lengths in the system (*J Am Chem Soc* 2025,147, 5, 4595).

[FIGURE REDACTED]

Figure R1 Reaction coordinates in the pathway of CO₂ reduction over Au₆/Zr₆O₈. (*Angew. Chem. Int. Ed.* 2025, e202500269)

[FIGURE REDACTED]

Figure R2 Free energy of different C-C reaction pathway and the optimized reaction intermediates (*Nano Energy* 141, 2025, 111077).

[FIGURE REDACTED]

Figure R3 DFT model of MIL-100(Fe). (a) An extended cluster model taken from the MIL-100(Fe) crystal structure. (b) Fe small cluster model taken from the circle in the large cluster with adsorbed NO molecule (*Adv. Mater.* 2024, 36, 2403053).

[FIGURE REDACTED]

Figure R4 Time-dependent structural evolution of the intermediates during the O₂ adsorption process (*J. Am. Chem. Soc.* 2025,147, 5, 4595).

Supplementary information, line 14-21: The Ti-MOF model framework has been built on the basis of experiment results and literature including: XRD patterns (Figure 2b),⁵ FTIR (Figure 2c)⁶ and X-ray absorption spectroscopy (XAS, Figure 2d-e; Supplementary Tables 1–2).”

Supplementary information, line 26-33: In the literature, there are many studies discussing the *OCl

and/or *Cl intermediate formation during CER process. As a typical example, Zhang and co-workers have detected by *in situ* Raman spectrum both *Cl and *OCl intermediates on Ir-based active sites.⁷ They have discussed the origin of these intermediates from the perspective of engineering coordination environment (Ir₁O₄ and Ir₁O₆). On another study, the authors have attributed the key of forming *OCl intermediate to appropriate adsorption energy caused by potential differences in electrochemical reaction processes.⁸ Further, in this work, we have experimentally observed *OCl intermediate through *in-situ* Raman experiments (Figure 5a-b). Therefore, we start the DFT calculations from *Cl and *OCl in the CER process.”

Ref 5. Wang, S. et al. Toward a Rational Design of Titanium Metal-Organic Frameworks. *Matter* **2**, 440–450 (2020).

Ref 6. Feng, X. et al. Rational Construction of an Artificial Binuclear Copper Monooxygenase in a Metal-Organic Framework. *J. Am. Chem. Soc.* **143**, 1107–1118 (2021).

Ref 7. Wang, J. et al. Engineering the Coordination Environment of Ir Single Atoms with Surface Titanium Oxide Amorphization for Superior Chlorine Evolution Reaction. *J. Am. Chem. Soc.* **146**, 11152–11163 (2024).

Ref 8. Sumaria, V., Krishnamurthy, D., Viswanathan, V. Quantifying Confidence in DFT Predicted Surface Pourbaix Diagrams and Associated Reaction Pathways for Chlorine Evolution. *ACS Catal.* **8**, 9034–9042 (2018).

Original Comment 10. *It was meaningless for the authors to merely compare whether the presence of a membrane affected the production costs of NaOH and Cl₂ with Ti-MOF as the anode. The authors should compare the costs of the products in a membrane-free system with Ti-MOF as the anode versus a traditional membrane system with DSA as the anode.*

Response: Thank you for the constructive comment. We have further calculated the production costs of NaOH/Cl₂ from above-mentioned two systems (Supplementary Tables 7,9-11; Supplementary Figures 53-56). We have given the response as follows:

The incorporation of DSA significantly elevates NaOH/Cl₂ production costs from traditional membrane system, which underperform Ti-MOF-based membrane-free system in the range of 1~14

year's operation. After 14 years, traditional DSA-membrane system begins to gain advantage due to its lower power consumption. Overall, the production cost per kg of Cl₂/NaOH are comparable for DSA-based membrane system and our Ti-MOF-based membrane-free system (\$0.36034 vs \$0.36069).

Based on above discussions, the following tables, figures and sentences have been added/updated in updated manuscript:

Supplementary Table 7. The prices of various raw materials.

Item	Producer/References	Price (\$ Unit price)	Price (\$ total)
2,6 naphthoic acid	Reagent Suppliers: Bide Pharmatech	\$438.36 kg ⁻¹	\$4109.59
NaOH	Industrial supplier: Dezhou Zhongzhiyuan Water Purification Materials	\$452 ton ⁻¹	\$1.57
N, N-Dimethylformamide	Industrial supplier: Shandong Jinshengrun Chemical Industry	\$500 ton ⁻¹	\$52.5
methanol	Industrial supplier: Shandong Mingshui Dahua Co., LTD	\$ 332.7 ton ⁻¹	\$3.74
Tetraisopropyl titanate	Reagent Suppliers: MERYER	\$42.6 kg ⁻¹	\$184.2
GDE	Industrial supplier: Shanghai Jiazhi Materials	\$94.52 m ⁻²	\$47415.95
DSA	42	\$4800 m ⁻²	\$2400000
water	https://www.fbgtx.org/673/Industrial-Water-Rates	\$0.409/ton	/
electricity	12,13,14,15	\$0.03 kWh	/
membrane	Industrial supplier: Asahi Kasei Singapore Branch	\$700 m ⁻²	\$350000
NaCl	43	\$0.0928 kg ⁻¹	/

Supplementary Table 9. FNPV (\$) analysis of Ti-MOF-membrane-free for CER at different current densities.

J (A cm ⁻²) \ Time (year)	1.14 A	0.7 A	0.34 A	0.18 A
0	-11941000	-11941000	-11941000	-11941000
1	-5827126.4	-8397349.173	-10306896.35	-10946665.68
2	-427552.84	-5445606.889	-9173770.417	-10422843.87
3	4697891.33	-2651430.564	-8111609.669	-9940972.77
4	9562259.94	-7317.05775	-7117034.806	-9499054.71
5	14177985.1	2493879.479	-6186827.454	-9095187.171
6	18556907.1	4858964.616	-5317922.493	-8727558.222
7	22710302.2	7094419.849	-4507400.761	-8394442.216
8	26648909.7	9206418.03	-3752482.105	-8094195.679
9	30382957.7	11200838.07	-3050518.76	-7825253.4
10	33922187.1	13083278.92	-2398989.042	-7586124.698
11	37275874.9	14859072.92	-1795491.353	-7375389.881
12	40452856.6	16533298.51	-1237738.451	-7191696.857
13	43461546.6	18110792.26	-723552.0135	-7033757.923
14	46309958.8	19596160.47	-250857.4475	-6900346.693
15	49005725.5	20993790.05	182321.0507	-6790295.181
16	51556115.6	22307859.03	577865.1986	-6702491.021
17	53968051.8	23542346.51	937567.1083	-6635874.813
18	56248127	24701042.06	1263133.553	-6589437.609
19	58402620.5	25787554.83	1556190.031	-6562218.503
20	60437512.2	26805322.09	1818284.635	-6553302.348

Supplementary Table 10. FNPV (\$) analysis of Ti-MOF-membrane for CER at different current densities.

J (A cm ⁻²) \ Time (year)	1A	0.7 A	0.34 A	0.18 A
0	-11941000	-11941000	-11941000	-11941000
1	-6664621.225	-9234844.03	-11144391.2	-11784160.54
2	-2062661.848	-7080715.896	-10808879.42	-12057952.88
3	2303149.805	-5046172.095	-10506351.2	-12335714.3
4	6444058.861	-3125518.134	-10235235.88	-12617255.79
5	10370774.97	-1313330.689	-9994037.622	-12902397.34
6	14093497.79	395555.3135	-9781331.796	-13190967.52
7	17621941.3	2006058.989	-9595761.621	-13482803.08
8	20965356.88	3522865.211	-9436034.924	-13777748.5
9	24132555.4	4950435.762	-9300921.063	-14075655.7
10	27131928.13	6293019.961	-9189247.998	-14376383.65
11	29971466.79	7554664.776	-9099899.501	-14679798.03
12	32658782.52	8739224.464	-9031812.497	-14985770.9
13	35201124.04	9850369.745	-8983974.534	-15294180.44
14	37605394.86	10891596.54	-8955421.372	-15604910.62
15	39878169.8	11866234.31	-8945234.686	-15917850.92
16	42025710.56	12777453.95	-8952539.884	-16232896.1
17	44053980.67	13628275.38	-8976504.018	-16549945.94
18	45968659.68	14421574.7	-9016333.806	-16868904.97
19	47775156.7	15160091.06	-9071273.74	-17189682.27
20	49478623.25	15846433.17	-9140604.289	-17512191.27

Supplementary Table 11. CER simulation electrolysis and FNPV (\$) analysis of DSA-membrane

for CER at 1 A cm⁻².

Current density (A cm ⁻²)	1	J (A cm ⁻²) Time (year)	1
Potential (V)	1.402	0	-26908800
Materials	DSA	1	-18894615
FE (%)	0.995	2	-12194330
Electrolytic area (m ²)	500	3	-5830113
Current (A)	5000000	4	214040
Cl ₂ production (kg h ⁻¹)	6589.6772	5	5953369
NaOH production (kg h ⁻¹)	7424.9883	6	11402389
H ₂ O consumption (kg h ⁻¹)	1679.017464	7	16574926
NaCl consumption (kg h ⁻¹)	10859.04545	8	21484145
Electrolytic power (kW)	7010	9	26142584
Separation power (kW)	27912	10	30562186
/	/	11	34754324
/	/	12	38729830
/	/	13	42499019
/	/	14	46071716
/	/	15	49457278
/	/	16	52664616
/	/	17	55702217
/	/	18	58578163
/	/	19	61300153
/	/	20	63875517

Supplementary Figure 53. Techno-economic analyses for Ti-MOF systems with/without membrane at 0.7 A cm^{-2} . a, FNPV analyses. b, Cost distribution percentages in an operating cycle. c-d, Production cost distribution in an operating cycle.

Supplementary Figure 54. Techno-economic analyses for Ti-MOF systems with/without membrane at 0.7 A cm^{-2} .

membrane at 0.34 A cm^{-2} . **a**, FNPV analyses. **b**, Cost distribution percentages in an operating cycle. **c-d**, Production cost distribution in an operating cycle.

Supplementary Figure 55. Techno-economic analyses for Ti-MOF systems with/without membrane at 0.18 A cm^{-2} . **a**, FNPV analyses. **b**, Cost distribution percentages in an operating cycle. **c-d**, Production cost distribution in an operating cycle.

Supplementary Figure 56. Techno-economic analyses. a, FNPV analyses for Ti-MOF-based membrane-free system and DSA-based membrane system. **b**, The production cost of electrochemical production per kilogram of Cl₂ and NaOH for Ti-MOF-based membrane-free system and DSA-based membrane system. **c-d**, cost distribution percentages and production cost distribution in an operating cycle for DSA-based membrane system.

“Page 11, Line 224-230: Finally, we have made corresponding comparison with DSA system with membranes, which is a benchmark process for current industrial chlor-alkaline process (Supplementary Figure 56, Supplementary Table 11). It shows that the incorporation of DSA significantly elevates NaOH/Cl₂ production costs from industrial chlor-alkaline process, which underperform Ti-MOF-based membrane-free system in the range of 1~14 year’s operation. After 14 tears, traditional DSA-membrane system begins to gain advantage due to its lower power consumption. Overall, the production cost per kg of Cl₂/NaOH are comparable for DSA-based membrane system and our Ti-MOF-based membrane-free system (\$0.36034 vs \$0.36069).”

“Supplementary information Line 103-105: Further, the technical and economic analyses of DSA-based system with membranes have also been conducted based on the electrochemical data from

literature (1 A cm^{-2} , Faradaic efficiency of 99.5%).¹¹ The specific calculation steps have been placed in Supplementary Tables 7 and 11.”

“Supplementary information Line 770-774: The incorporation of DSA significantly elevates NaOH/Cl₂ production costs from traditional membrane system, which underperform Ti-MOF-based membrane-free system in the range of 1~14 year’s operation. After 14 years, traditional DSA-membrane system begins to gain advantage due to its lower power consumption. Overall, the production cost per kg of Cl₂/NaOH are comparable for DSA-based membrane system and our Ti-MOF-based membrane-free system (\$0.36034 vs \$0.36069).”

Ref 11. Yang, J. et al. CO₂-mediated organocatalytic chlorine evolution under industrial conditions. *Nature* **617**, 519–523 (2023).

Ref 12. Chen, F.-Y. et al. Electrochemical nitrate reduction to ammonia with cation shuttling in a solid electrolyte reactor. *Nat. Catal.* **7**, 1032–1043 (2024).

Ref 13. Zhang, L. et al. High-efficiency ammonia electrosynthesis from nitrate on ruthenium-induced trivalent cobalt sites. *Energy Environ. Sci.* **18**, 5622–5631 (2025).

Ref 14. Gao, Y. et al. Membrane-Free Electrosynthesis of Epichlorohydrins Mediated by Bromine Radicals over Nanotips. *J. Am. Chem. Soc.* **146**, 714–722 (2023).

Ref 15. Cheng, C. et al. Selective electrosynthesis of 1,3-butadiene by tailoring the coverage of acetylene and water. *Nat. Commun.* **16**, 5685 (2025).

Ref 42. Yang, J., Zhu, C., Wang, D. A Simple Organo-Electrocatalysis System for the Chlor-Related Industry. *Angew. Chem., Int. Ed.* **63**, e202406883 (2024).

Ref 43. Micari, M. et al. Techno-economic analysis of integrated processes for the treatment and valorisation of neutral coal mine effluents. *J. Cleaner Prod.* **270**, 122472 (2020).

Original Comment 11. *Is pH-universal Cl₂ electrosynthesis attributed to the use of Ti-MOF or the strategy of chlorine gas separation?*

Response: Thanks for your constructive comment. We consider the strategy of chlorine gas separation is the reason of achieving pH-universal Cl₂ electrosynthesis. We have given more detailed explanation as follows:

Theoretically, the key of chlorine gas separation strategy is to generate interfacial pressure difference based on Bernoulli's description, which promotes oriented migration of Cl₂ away from reaction interfaces (*i.e.*, H₂/OH⁻ at cathode/anode). This can not only prevent H₂/Cl₂ crossover but also circumvent the Pourbaix diagram problems, leading to significantly enhanced activities in pH-universal condition.

Experimentally, without chlorine gas separation, Ti-MOF has demonstrated diminished activities with elevated pH values, *i.e.*, Faradaic efficiency of 72.8% (yield rate of 2.28 mmol cm⁻² h⁻¹) at pH =1, 5.8% (yield rate of 0.21 mmol cm⁻² h⁻¹) at pH =7 and 0% (yield rate of 0 mmol cm⁻² h⁻¹) at pH =13. In great contrast, with chlorine gas separation by interfacial pressure difference, the activities of Ti-MOF increased significantly, *i.e.*, Faradaic efficiency of 93.4% (yield rate of 1.68 mmol cm⁻² h⁻¹) at pH =1, 94% (yield rate of 0.93 mmol cm⁻² h⁻¹) at pH =7 and 88.8% (yield rate of 0.78 mmol cm⁻² h⁻¹) at pH =13.

Other than Ti-MOF, we have loaded the main component (RuO₂ and TiO₂) of DSA benchmark on gas diffusion electrode (GDE) for chlorine evolution reaction, which has shown a similar phenomenon. Without chlorine gas separation, it exhibits the Faradaic efficiency of 70.9% (yield rate of 3 mmol cm⁻² h⁻¹) at pH =1, 7.3% (yield rate of 0.3 mmol cm⁻² h⁻¹) at pH =7 and 0% (yield rate of 0 mmol cm⁻² h⁻¹) at pH =13. In great contrast, with chlorine gas separation by interfacial pressure difference, its activities increased significantly, *i.e.*, Faradaic efficiency of 86.3% (yield rate of 3.42 mmol cm⁻² h⁻¹) at pH =1, 88% (yield rate of 3.3 mmol cm⁻² h⁻¹) at pH =7 and 81.4% (yield rate of 3.3 mmol cm⁻² h⁻¹) at pH =13.

Further, mechanism study of *operando* Raman spectra has revealed the key role of chlorine gas separation. By comparing the *operando* Raman spectra for chlorine evolution reaction (CER)

with/without interfacial pressure difference (Figure 5a and supplementary Figure 57), we have unambiguously seen the decreased *OCl peak signal, which confirm the role of interfacial pressure difference, *i.e.*, Cl₂ produced at the electrode/electrolyte interfaces intermediately taken away by gas flow following Bernoulli's principle.

Quantitatively, our COMSOL simulation based on Bernoulli's principle (Figure 5c-d) has shown the key role of chlorine gas separation. With the carrier flow of 0~80 mL min⁻¹ and electrolyte flow of 8.4 mL min⁻¹, a pressure difference can be generated in the range of 0 ~ 116.3 mPa with, which drive as-generated Cl₂ migration to gas chamber.

Based on above discussions, the following sentences/figures have been added into updated manuscript:

“Page 7, Line 131-136: For reliable compare with benchmark DSA, we have further loaded the main components (RuO₂ and TiO₂) of DSA on gas diffusion electrode (GDE, Supplementary Figure 31). By applying interfacial pressure difference, the activities increased significantly, *i.e.*, Faradaic efficiency of 86.3% (yield rate of 3.42 mmol cm⁻² h⁻¹) at pH =1, 88% (yield rate of 3.3 mmol cm⁻² h⁻¹) at pH =7 and 81.4% (yield rate of 3.3 mmol cm⁻² h⁻¹) at pH =13.”

“Page 12, line 264: So, we consider interfacial pressure difference is the key to achieve pH-universal Cl₂ electrosynthesis.”

Supplementary Figure 31. Comparison experiments for different samples. a-d, The Cl₂ Faradic efficiencies and yield rates without pressure difference for GDE loaded (RuO₂/TiO₂) as a surrogate of DSA. **c-d**, Cl₂ Faradic efficiency and yield rate of Ti-MOF materials without pressure difference.

Original Comment 12. *It is still quite confusing whether the pH tolerance is brought by the characteristic of Ti-MOF or the pressure difference (namely the three-phase set up). Obviously, the DSA exhibits very poor CER performance at pH=7 and 13 in a traditional three-electrode set up. However, when testing the CER activity of Ti-MOF, the authors tuned the pressure difference which is not performed for the scenario of DSA. The authors should provide the experimental conditions clearly in the Method Section/Figure Illustration. Furthermore, if Ti-MOF is the main contributor for*

the pH tolerance, what property brings this benefit?

Response: Thanks for your constructive comment. We consider pressure difference as main contributor for the pH tolerance. We give more detailed discussions as follows:

As discussed in the response to Comment 11, the as-generate pressure difference at three-phase boundary can promote oriented migration of Cl_2 away from reaction interfaces (*i.e.*, H_2/OH^- at cathode/anode). This can not only prevent H_2/Cl_2 crossover but also circumvent the Pourbaix diagram problems, as Cl_2 will not contact with $\text{H}_2\text{O}/\text{OH}^-$ in cathode/anode, leading to significantly enhanced activities in pH-universal condition.

Without pressure difference, Ti-MOF has demonstrated diminished activities with elevated pH values, *i.e.*, Faradaic efficiency of 72.8% (yield rate of $2.28 \text{ mmol cm}^{-2} \text{ h}^{-1}$) at pH =1, 5.8% (yield rate of $0.21 \text{ mmol cm}^{-2} \text{ h}^{-1}$) at pH =7 and 0% (yield rate of $0 \text{ mmol cm}^{-2} \text{ h}^{-1}$) at pH =13. In great contrast, with chlorine gas separation by interfacial pressure difference, the activities of Ti-MOF increased significantly, *i.e.*, Faradaic efficiency of 93.4% (yield rate of $1.68 \text{ mmol cm}^{-2} \text{ h}^{-1}$) at pH =1, 94% (yield rate of $0.93 \text{ mmol cm}^{-2} \text{ h}^{-1}$) at pH =7 and 88.8% (yield rate of $0.78 \text{ mmol cm}^{-2} \text{ h}^{-1}$) at pH =13.

Further, following the reviewer's comment, we have tried to tune the pressure difference for DSA-based system, however, cannot achieve this operation. This is because of DSA fabricated by coating bulk Ti substrate with electrocatalytic metal oxide layers (*e.g.*, RuO_2 , IrO_2). *Chem Rev* 2016, 116, 2982 The bulk Ti substrate does not allow gas penetration at three-phase boundary during electrochemical reaction. Consequently, we have replaced Ti substrate of DSA with gas diffusion electrode (GDE), and synthesized the electrode of metal oxide (*e.g.*, RuO_2 , IrO_2)/GDE. Particularly, we have tested the performance of $\text{RuO}_2/\text{TiO}_2@\text{GDE}$ electrode under the same condition. Without pressure difference, the $\text{RuO}_2/\text{TiO}_2@\text{GDE}$ L electrode has demonstrated diminished activities with elevated pH values, *i.e.*, Faradaic efficiency of 70.9% (yield rate of $3 \text{ mmol cm}^{-2} \text{ h}^{-1}$) at pH =1, 7.3% (yield rate of $0.3 \text{ mmol cm}^{-2} \text{ h}^{-1}$) at pH =7 and 0% (yield rate of $0 \text{ mmol cm}^{-2} \text{ h}^{-1}$) at pH =13. In great contrast, with pressure difference, the activities of $\text{RuO}_2/\text{TiO}_2@\text{GDE}$ increased significantly, *i.e.*, Faradaic efficiency of 86.3% (yield rate of $3.42 \text{ mmol cm}^{-2} \text{ h}^{-1}$) at pH =1, 88% (yield rate of $3.3 \text{ mmol cm}^{-2} \text{ h}^{-1}$) at pH =7 and 81.4% (yield rate of $3.3 \text{ mmol cm}^{-2} \text{ h}^{-1}$) at pH =13.

Finally, Ti-MOF is only one of the contributors for pH tolerance as it can work in pH-universal condition. We attribute it to robust Ti-O bonding within the framework, which has been well documented in the literature, like *Matter* 2020, 2, 440; *Chem Sci* 2020, 11, 5339 and *Mater Chem Phy* 2015, 151, 133.

Based on above discussions, the following experimental conditions and results have been added into updated manuscript.

Page 5, Line 94-95: Notably, Ti-MOF can survive in pH-universal condition, owing to the robust Ti-O bonding within the framework.^{28,29,30},

Page 15, Line 331-341:

Electrochemical testing setup. As shown in Supplementary Figure 30e, our test was conducted in a flow-type reaction device. The anode and cathode have faced each other, and the reference electrode is placed between the anode and cathode. The middle chamber is connected by the electrolyte, while the rightmost chamber allows the carrier gas to pass through. Both the electrolyte and the carrier gas enter through the lower liquid (gas) inlet and exit through the upper liquid (gas) outlet.

We have evaluated the performances of prototype device with a fixed NaCl electrolyte rate (8.4 mL min⁻¹) and tunable gas flow rates (Supplementary Video S2). Since DSA is manufactured by coating an electrocatalytic metal oxide layer (such as RuO₂, IrO₂) on a titanium substrate, its structure is a solid titanium plate, so it is impossible to apply pressure difference treatment to the back of the electrode. Therefore, the effective components (RuO₂ and TiO₂) were used as a surrogate for DSA. In details, the RuO₂ and TiO₂ mixture with a mass ratio of 1:1 was prepared as ink and sprayed onto GDE under the same mass loading.

Supplementary information, Line 341-359:

We have tried to tune the pressure difference for DSA-based system, however, cannot achieve this operation. This is because of DSA fabricated by coating bulk Ti substrate with electrocatalytic metal oxide layers (e.g., RuO₂, IrO₂)²³ The bulk Ti substrate does not allow gas penetration at three-phase boundary during electrochemical reaction. Consequently, we have replaced Ti substrate of DSA with gas diffusion electrode (GDE), and synthesized the electrode of metal oxide RuO₂/TiO₂@GDE. Particularly, we have tested the performance of RuO₂/TiO₂@GDE electrode under the same condition.

Without pressure difference, the RuO₂/TiO₂@GDE electrode has demonstrated diminished activities with elevated pH values, *i.e.*, Faradaic efficiency of 70.9% (yield rate of 3 mmol cm⁻² h⁻¹) at pH =1, 7.3% (yield rate of 0.3 mmol cm⁻² h⁻¹) at pH =7 and 0% (yield rate of 0 mmol cm⁻² h⁻¹) at pH =13. In great contrast, with pressure difference, the activities of RuO₂/TiO₂@GDE increased significantly, *i.e.*, Faradaic efficiency of 86.3% (yield rate of 3.42 mmol cm⁻² h⁻¹) at pH =1, 88% (yield rate of 3.3 mmol cm⁻² h⁻¹) at pH =7 and 81.4% (yield rate of 3.3 mmol cm⁻² h⁻¹) at pH =13.

The same phenomenon has also been observed for Ti-MOF. Without pressure difference, Ti-MOF has demonstrated diminished activities with elevated pH values, *i.e.*, Faradaic efficiency of 72.8% (yield rate of 2.28 mmol cm⁻² h⁻¹) at pH =1, 5.8% (yield rate of 0.21 mmol cm⁻² h⁻¹) at pH =7 and 0% (yield rate of 0 mmol cm⁻² h⁻¹) at pH =13. In great contrast, with pressure difference, the activities of Ti-MOF increased significantly, *i.e.*, Faradaic efficiency of 93.4% (yield rate of 1.68 mmol cm⁻² h⁻¹) at pH =1, 94% (yield rate of 0.93 mmol cm⁻² h⁻¹) at pH =7 and 88.8% (yield rate of 0.78 mmol cm⁻² h⁻¹) at pH =13.”

Manuscript Ref 28. Wang, S. et al. Toward a Rational Design of Titanium Metal-Organic Frameworks. *Matter* **2**, 440–450 (2020).

Manuscript Ref 29. Duong, T. D. et al. Observation of binding of carbon dioxide to nitro-decorated metal–organic frameworks. *Chem. Sci.* **11**, 5339–5346 (2020).

Manuscript Ref 30. Rasmi, K. R., Vanithakumari, S. C., George, R. P., Mallika, C., Kamachi Mudali, U. Development and performance evaluation of nano platinum coated titanium electrode for application in nitric acid medium. *Mater. Chem. Phys.* **151**, 133–139 (2015).

Supplementary information Ref 23. Karlsson, R. K. B. Cornell, A. Selectivity between Oxygen and Chlorine Evolution in the Chlor-Alkali and Chlorate Processes. *Chem. Rev.* **116**, 2982–3028 (2016).

Original Comment 13. *The authors are suggested to provide solid data source during technical-economic evaluations. Important parameters include the price of electricity, NaCl, water, membrane, etc. As the American manufacturing industry is mentioned (Page 7, Supporting Information), we hypothesize this Cl₂ production factory to be set up in the U.S. Based on the data on Global-Petrol-Prices., the industrial electricity price in the U.S. is estimated at \$0.15/kWh in 2024, much higher*

than the \$0.03/kWh used in the authors' estimates.

Response: Thanks for your kind reminding. We have given the response as follows:

- (1) We have provided the prices of all important parameters during technical-economic evaluations. Please see the Supplementary Table 7 in updated manuscript.
- (2) Regarding the industrial electricity price, we consider the discrepancy in electricity pricing data stemming from different consuming sectors. Particularly, U.S. electricity rates exhibit significant variations across residential, commercial, and industrial sectors.^{hangyan.co/charts/3539299747577726947} Given that significant role of electricity price for cost estimates, we have retrieved a number of state-of-the-art literatures as follows:

i) In the TEA calculation of *Nat Commun* 2025, 16, 5685. (Supplementary Information Page 78): “The electricity price was considered to be 0.03 \$ kWh⁻¹”;

ii) In OPEX calculations of *Nat Catal* 2024 7, 1032. Supplementary Information (Supplementary Information Page 33): “The electricity cost is calculated from the power requirement and an electricity price of 3 cent kWh⁻¹ based on the target announced by DOE”

iii) In operating costs of *Energy Environ Sci* 2025 18, 5622. Supplementary Information (Supplementary Information S64): “NH₃ air stripping cost is calculated with the reference of 5.6 kWh⁻¹ and the kg(N) electricity price of 3 cent per kWh”

iv) In TEA of *J Am Chem Soc* 2024 146, 714. (Supplementary Information Page 78): “The electricity price was considered to be 0.03 \$ kWh⁻¹”

Based on above literature, we would like to select the electricity price of 0.03 \$ kWh⁻¹. We have updated all of technical-economic data, and the following tables and sentences have been added into updated manuscript.

Supplementary Table 7. The prices of various raw materials.

Item	Producer/References	Price (\$ Unit price)	Price (\$ total)
2,6 naphthoic acid	Reagent Suppliers: Bide Pharmatech	\$438.36 kg ⁻¹	\$4109.59
NaOH	Industrial supplier: Dezhou Zhongzhiyuan	\$452 ton ⁻¹	\$1.57

	Water Purification Materials		
N, N-Dimethylformamide	Industrial supplier: Shandong Jinshengrun Chemical Industry	\$500 ton ⁻¹	\$52.5
methanol	Industrial supplier: Shandong Mingshui Dahua Co., LTD	\$ 332.7 ton ⁻¹	\$3.74
Tetraisopropyl titanate	Reagent Suppliers: MERYER	\$42.6 kg ⁻¹	\$184.2
GDE	Industrial supplier: Shanghai Jiazhi Materials	\$94.52 m ⁻²	\$47415.95
DSA	42	\$4800 m ⁻²	\$2400000
water	https://www.fbgtx.org/673/Industrial-Water-Rates	\$0.409/ton	/
electricity	12,13,14,15	\$0.03 kWh	/
membrane	Industrial supplier: Asahi Kasei Singapore Branch	\$700 m ⁻²	\$350000
NaCl	43	\$0.0928 kg ⁻¹	/

“Supplementary information, Line 120-124: Regarding electricity price, we consider the discrepancy in electricity pricing data stemming from different consuming sectors. Particularly, U.S. electricity rates exhibit significant variations across residential, commercial, and industrial sectors. Given that significant role of electricity price for production cost estimates, we have retrieved a number of state-of-the-art literatures,^{12,13,14,15} and select the electricity price as 0.03 \$ kWh⁻¹.^{14,15}”

Original Comment 14. *The oxygen evolution reaction (OER) chemical equation (in Line 43, Page 3) was incorrect. Generally, OER involves a four-electron transfer rather than a six-electron transfer.*

Response: We are sorry for making the mistake. We have made correction as follows:

“Page 3, Line 43: Side OER: $2\text{H}_2\text{O} \rightarrow \text{O}_2 + 4\text{H}^+ + 4\text{e}^-$, $E^0=1.23 \text{ V vs. RHE}$ (3)”

Original Comment 15. *The term "Cl₂ yield rates" in Line 128-130, Page 7 was incorrect and should be replaced with "the Cl₂ Faradaic efficiencies".*

Response: Thanks for your kind reminding. The typo has been corrected as follows:

“Page 7, Line 141-142: “The Cl₂ Faradaic efficiencies of Ti-MOF increase with calcination temperatures...””

Original Comment 16. *There are many incorrect expressions and figure numbers in the main text. Such as “...the yield rate of Ti-MOF is 2.6 times the one of DSA...” in Line 161, Page 8, which should be change to “...the Cl₂ yield rate of Ti-MOF is 2.6 times the one of DSA...”.*

Response: Thank you for your kind reminding. We have rechecked the whole manuscript, and found a number of typos and errors. We have made corrections as follows:

“Page 2, Line 18: saving 6.75% (1.17 million dollar per year) as comparison to conventional chlor-alkali design.”

“Page 7, Line 129-130: Specifically at the potential of 1.8 V with Ti-MOF catalyst, the Cl₂ Faradaic efficiencies are 93.3, 94, and 88.8% at the pH of 1, 7, 13, respectively.”

“Page 9, Line 175-180: As showing in Figures 3d, 3e, the Cl₂ gas Faradaic efficiencies for Ti-MOF is 94% at the potential of 1.8 V, which is 18 times the one of DSA under the same test conditions and set-up. Analogously, the Cl₂ gas yield rate of Ti-MOF is 3.4 times the one of DSA at a potential of 1.8 V. Further with Ti-MOF, the performances with and without membranes have been compared, which shows minor alternations in Cl₂ Faradaic efficiencies (94% vs. 92%) and yield rates (0.93 vs. 0.9 mmol h⁻¹ cm⁻² Figures 3d, 3e and Supplementary Figure 47).”

“Page 9, Line 187-188: The overall Cl₂ Faradaic efficiencies maintain stable over the whole 200-hr period (Supplementary Figure 48)”

“Page 19, Line 414: Pt foil and Ag/AgCl electrode were used as the counter and reference electrodes,”

Replace all occurrences of “Faradic” with “Faradaic” in the Manuscript. They are respectively found on: Page 2, Line 15; Page 5, Line 73-74; Page 8, Line 162-163; Page 8, Line 168; Page 8, Line 172;

Page 9, Line 186-188; Page 9, Line 195; Page 14, Line 306; Page 24, Line 520; Page 25, Line 529;
Page 26, Line 536.

We would like to thank the Editor's invitation to revise the manuscript. We are also appreciated of the reviewer #5 for his/her constructive comments to improve the quality of this work.

After receiving the reviewer's comments, we have spent ten intensive weeks in order to answer the concerns raised by the astute reviewer. We believe we have shifted the work to the next level of proof and gained a much deeper understanding of the system. Please see the following point-by-point response to the reviewer #5's comments. Thank you and best wishes,

Kind regards

The authors

Response to Reviewer #5

Original Comment: *In this article, the authors designed and prepared an efficient yet cost-effective electrochemical system for Cl₂ electrosynthesis. It modulates gas diffusion layer by Bernoulli's principle, wherein the pressure difference at triple-phase boundary drives oriented Cl₂ migration directly into gas chamber, thus preventing the crossover of anodic/cathodic products and reaching a pH-tolerant CER. This article provides excellent and comprehensive guidance on design and development of electrocatalytic chlorine evolution devices and effectively reduces industrial costs. However, there are still many problems in terms of structural characterization and mechanism exploration. In a whole, this work cannot be accepted for becoming suitable for this journal.*

Original Comment 1. *The author points out that “a Ti-MOF nanosheet reveals its well-defined crystallinity with the adjacent lattice spacing of 0.2 nm, which agrees well with XRD containing a series of characteristic peaks corresponding to metal-organic framework architectures with dominant (210) crystal face” (Line 90-92). However, in fact, the so-called lattice spacing were not observed in Figure. 2a, and XRD (Figure. 2b) can only correspond to the three peaks before 20°, but basically did not match peaks after 20°. Therefore, a Fourier-transform TEM image or SAED image should be exhibited to support this view. At the same time, XRD needed to be verified for correctness.*

Response: Thanks for the kind reminding. We would like to give a response as follows:

- (1) Firstly, we have repeated TEM characterization for Ti-MOF and processed the TEM image. As shown in Figure 2a, the inset (upper right corner) now displays lattice fringes with improved resolution, and the lattice spacing of 0.21 nm. Further, following the reviewer's suggestion, the profile of dislocations after FTT processing and the specific data of measured crystal plane spacings have also been presented for reference (Supplementary Figure 3).
- (2) Secondly, we plan to identify each XRD peak by comparison with theoretical simulation ones. However, Ti-MOF is a polycrystalline material with multiple crystal planes, and some simulated diffraction peaks show weak intensities. *ACS Mater Lett* 2021, 3, 64; *Crystal Growth Des* 2023, 23, 3778 To more clearly illustrate these weak peaks, we have amplified relevant areas (as shown in Figure 2b). Now in the simulated XRD profile, all of the peak positions are clearly visible.
Subsequently, we have compared each experimental XRD peak with theoretically simulation patterns. As shown in the Supplementary Table 1, all of the indexed values match well, providing

strong evidence for the successful synthesis of desired Ti-MOF phase.

Based on above discussions, the following sections have been added/revised in updated manuscript:

Figure 2. Catalyst and system design for Cl₂ electrosynthesis beyond Pourbaix diagram. a, TEM and enlarge the image of Ti-MOF catalyst. The blue wavy line form is the lattice spacing after FFT. **b,** X-ray diffractometer (XRD) of Ti-MOF (In order to make the peaks ranging from 25 to 60° in the theoretical simulation more clearly, they were enlarged by a factor of 10).

Supplementary Figure 3. The dislocation analyses for measuring crystal plane spacing of Ti-MOF.

Supplementary Table 1. The crystal plane indices of Ti-MOF and the simulated value.

(h k l) (Simulation)	2 θ (Simulation)	2 θ (Experiment)
(1 0 0)	7.434	7.49
(1 0 1)	10.826	10.6
(2 0 1)	14.665	15
(2 1 0)	16.171	16
(0-1 1)	19.759	19.3
(0 1 2)	25.79	25.9
(3 2 0)	29.104	29.5
(4 2 0)	32.676	32.5
(0 2 3)	43.289	43.3

“Page 5, Line 87-91: Close examination of a Ti-MOF nanosheet reveals its well-defined crystallinity with the adjacent lattice spacing of 0.21 nm (Figure 2a, Supplementary Figure 3), which agrees well with X-ray diffraction (XRD) containing a series of characteristic peaks corresponding to metal-organic framework architectures with dominant (2 1 0) crystal face (Figure 2b, Supplementary Table 1).²⁵”

Original Comment 2. *The data from XAS in Figure. 2d-e and Supplementary Table 1 contradicts the author’s description of catalysts structure in the article. The author points out that “The Ti node is octahedrally coordinated” (Line 111-112). But the EXAFS-fitting indicated a CN of 3.183 for Ti-O (Line 103 in Supplementary Table 1). It’s necessary to clarify the error values towards each parameter in Supplementary Table 1 and explain this contradictory phenomenon.*

Response: We are sorry for not explaining it clearly. “Ti node is octahedrally coordinated” means one Ti atom can theoretically coordinated with six other atoms in the form of octahedra architecture, which represents the global symmetry constraints for Ti atoms. On the other hand, the average coordination number of Ti-O in Ti-MOF has been experimentally determined by XAS (Figures 2d, e, Supplementary Figures 7–8), where the average number is 3.98 (before low-temperature calcination)

or 3.36 (after calcination). To clarify the relationship between the two, we have built a structural model for Ti-MOF as follows (Figure R1):

As shown in Figure R1, the spatial configuration within the Ti-O cluster is in the form of octahedra containing of 16 Ti atoms and 96 O atoms with a ratio of 1/6 (Figure R1). However, when counting the actual coordination number of Ti-O from XAS, the contribution of each atom has to be taken into account, for example, the coordination of O atoms would be considered according to their local environment, which is 1 (internal O atom) or 0.5 (surface O atom). We can see that 48 O atoms are internal and 48 O atoms are on the surface. Accordingly, the number of O atoms coordinated to Ti is calculated as: $48 + 48 \times 0.5 = 72$, resulting in a Ti/O ratio of 1:4. This result is close to our experimentally XAS measured value (3.98).

Further, to improve its crystallinity, the Ti-MOF has been subject to calcination at 450 °C for 3 hrs. Such a calcination treatment has not altered Ti-MOF architecture, which is still in the form of octahedra, as confirmed by TEM (Figure 2a), XRD (Figure 2b), FT-IR (Figure 2c). While X-ray absorption near-side structure (XANES) confirms the coordination number of Ti-O has decreased to 3.36 (Supplementary Table 2). This phenomenon indicates the evaporation of some organic ligands during calcination process, which can expose more Ti active sites for electrochemical reaction.

Based on above results, the following figure/sentence has been added into updated manuscript.

Figure R1. The periodic structure of Ti-MOF.

Supplementary Table 2. Bond length and coordination of Ti-MOF-before and after calcination.

Material	shell	CN	R (Å)	ΔE_0 (eV)	σ^2 (10^{-3} Å ²)	R factor
Ti-foil	Ti-Ti	6.24±0.21	2.89±0.002	-7.05±0.18	0.006±0.0005	0.0084±0.00224
TiO ₂	Ti-C	4.15±0.05	1.95±0.003	-1.55±0.26	0.006±0.0001	0.014±0.0012
	Ti-Ti	2.05±0.05	3.06±0.0004		0.0040±0.00025	
TiC	Ti-O	6.03±0.08	2.14±0.00039	-3.47±0.01	0.0002±0.0001	0.0095±0.00035
	Ti-Ti	11.93±0.14	3.05±0.0002		0.0009±0.0001	
Ti-MOF (before)	Ti-O	3.98±0.04	1.90±0.0005	-7.04±0.10	0.016±0.0001	0.0034±4.1×10 ⁻⁵
	Ti-Ti	4.05±0.002	3.10±10 ⁻⁷		0.019±1.5×10 ⁻⁵	
Ti-MOF	Ti-O	3.36±0.11	1.92±0.002	-4.21±0.21	0.011±0.0009	0.013±0.0043
	Ti-Ti	4.05±0.09	3.12±0.00008		0.018±0.00064	

“Page 6, Line 111-117: Ti atom can theoretically coordinate with six other atoms in the form of hexacoordinated architecture, which represents the global symmetry constraints for Ti atoms. On the other hand, the coordination number of Ti-O in Ti-MOF has been experimentally determined by XAS (Figures 2d, e, Supplementary Figures 7–8), where the average number is 3.98 (before low-temperature calcination) or 3.36 (final structure after calcination). This phenomenon indicates the evaporation of some organic ligands during calcination process, which can expose more Ti active sites for electrochemical reaction.”

Original Comment 3. *The author used operando Raman spectroscopy to verify the intermediates in the reaction process and observed potential-dependent *OCl signals, but in reality, no corresponding signal was observed at 1077 cm⁻¹ in either Figure 5a or Supplementary Figure 41a-b.*

Response: Thanks for your kind reminding. We have re-tested the *operando* Raman data for five times and provided the response as follows:

Generally, Raman spectra is a scattering signal with intrinsic weak intensity. When used to study the structural characteristics of a bulk material, obvious signal peaks can be seen. This is due to the structural crystal lattice that contributes to overall vibration of the bulk material, leading to enhanced scattered signals.

While in this work, the *operando* Raman is used to probe the transient surface-adsorbed species (*OCl intermediates) during electrocatalysis. The scattered Raman signals are mainly focused on bond vibrations of adsorbed species on catalyst surfaces, which are known to be very weak as comparison to structural crystal lattices in bulk materials. Consequently, the *operando* Raman signals for catalytic reactions often demonstrate characteristically low signal-to-noise ratios in the literatures (like ORR, NRR, CRR).^{Nat Commun 2024 15, 4157; Angew Chem Int Ed 2025, 64, e202414202; J Am Chem Soc 2024, 146, 11152} Our *operando* Raman signals are comparable to the above literature.

To confirm the accuracy of the experimental results, we performed five *operando* Raman tests. We directly determined the positions and intensities of the peaks using Raman spectrometer software, and updated the *operando* Raman data with error bars (Figure 5a-b and Supplementary Figure 57 in updated manuscript). By comparing the Raman vibration at different electrolysis potentials, we conclude the characteristic *OCl Raman peak at approximately 1093 cm⁻¹ (inside the blue transparent frame), and the intensity of *OCl peak changes consistently with applied potentials, reaching maximum intensity at 2.0 V vs Ag/AgCl (Figure 5b). Further, the data demonstrate significantly greater accumulation of the *OCl intermediate under no pressure condition. This observation suggests that the Bernoulli principle leveraged in our reaction system generates an immediate pressure gradient, enabling continuous removal of Cl₂ product and thereby enhancing chlorine evolution kinetics.

Based on above results, relevant figure/sentence has been revised/added into updated manuscript:

Figure 5. Mechanism study. **a**, The *operando* Raman spectra. **b**, Ti-*OCl Raman peak intensity.

Supplementary Figure 57. The *operando* Raman spectra without pressure difference. **a**, *operando* Raman spectra. **b**, Ti-*OCl Raman peak intensity.

Page 11, line 235-250: The applied potentials have been polarized from 1.0 to 2.0 V in 200 mV steps (Figure 5a, Supplementary Figure 57). The *operando* Raman spectra has displayed a subset of vibrational data: the band at 1601 cm^{-1} is assigned to the C=C stretching vibration of the benzene ring, while the band at 1374 cm^{-1} corresponds to the symmetric stretching vibration of the carboxylate group ($-\text{COO}^-$) in the MOF substrate.; the band at 3400 cm^{-1} from H_2O ; the potential-dependent band at 1093 cm^{-1} from the Cl-O stretching mode of *OCl during CER process.³³

Supplementary information, line 530-550: Generally, Raman spectra is a scattering signal with

intrinsic weak intensity. When used to study the structural characteristics of a bulk material, obvious signal peaks can be seen. This is due to the structural crystal lattice that contributes to overall vibration of the bulk material, leading to enhanced scattered signals.

While in this work, the *operando* Raman is used to probe the transient surface-adsorbed species (*OCl intermediates) during electrocatalysis. The scattered Raman signals are mainly focused on bond vibrations of adsorbed species on catalyst surfaces, which are known to be very weak as comparison to structural crystal lattices in bulk materials. Consequently, the *operando* Raman signals for catalytic reactions often demonstrate characteristically low signal-to-noise ratios in the literatures (like ORR, NRR, CRR).^{7,19,28} Our *operando* Raman signals are comparable to the above literature.

To confirm the accuracy of the experimental results, we performed five *operando* Raman tests. We directly determined the positions and intensities of the peaks using Raman spectrometer software, and updated the *operando* Raman data with error bars (Figures 5a-b and Supplementary Figure 57). By comparing the Raman vibration at different electrolysis potentials, we conclude the characteristic *OCl Raman peak at approximately 1093 cm^{-1} (inside the blue transparent frame), and the intensity of *OCl peak changes consistently with applied potentials, reaching maximum intensity at 2.0 V vs Ag/AgCl (Figure 5b). Further, the data demonstrate significantly greater accumulation of the *OCl intermediate under zero pressure differential conditions. This observation suggests that the Bernoulli principle leveraged in our reaction system generates an immediate pressure gradient, enabling continuous removal of Cl₂ product and thereby enhancing chlorine evolution kinetics. All of these results provide the validity of our *operando* synthesis condition.”

Ref 7. Wang, J. et al. Engineering the Coordination Environment of Ir Single Atoms with Surface Titanium Oxide Amorphization for Superior Chlorine Evolution Reaction. *J. Am. Chem. Soc.* **146**, 11152–11163 (2024).

Ref 19. Huang, Q. et al. Single-zinc vacancy unlocks high-rate H₂O₂ electrosynthesis from mixed dioxygen beyond Le Chatelier principle. *Nat. Commun.* **15**, 4157 (2024).

Ref 28. Quan, L., Zhao, X., Yang, L. M., You, B., Xia, B. Y. Intrinsic Activity Identification of Noble Metal Single-Sites for Electrocatalytic Chlorine Evolution. *Angew. Chem., Int. Ed.* **64**, 202414202 (2024).

Original Comment 4. *Could the authors clearly describe the correlation of Bernoulli principle in its reaction process? It seems that there is no connection between them, and the mechanism study part has not been well reflected.*

Response: Thanks for your constructive comment. The key role of Bernoulli principle is to illustrate the interfacial pressure difference at three-phase boundary, which promote oriented Cl₂ migration to gas chamber during reaction process. We would like to give a more detailed explanation of mechanism both theoretically and experimentally as follows:

(1) Theoretically, Bernoulli principle a basic rule in fluid dynamics illustrating the relationship between the surface velocity and pressure. In the present work, two fluids have presented at the three-phase boundary, carrier gas flow and electrolyte flow, and according to Bernoulli principle that can be illustrated as:

$$P_1 + \frac{1}{2}\rho_1V_1^2 + \rho_1gh_1 = P_2 + \frac{1}{2}\rho_2V_2^2 + \rho_2gh_2$$

where 1 represents carrier gas flow and 2 represents electrolyte flow. So, the interfacial pressure difference (∇p) can be illustrated as:

$$\nabla p = \frac{1}{2}\rho_1V_1^2 + \rho_1gh_1 - \frac{1}{2}\rho_2V_2^2 - \rho_2gh_2$$

Therefore, when high-velocity carrier gas flow and low electrolyte flow have been applied to three-phase boundary, it will generate a large localized pressure gradient. This can promote Cl₂ bubbles migration away from electrolyte flow, and into carrier gas. To clearly clarify this mechanism, we have provided a schematic animation in Supplementary Video S2.

(2) Quantitatively, the function of pressure difference has been described by COMSOL simulations (Figures 5c-d, Supplementary Video S1) and experimental phenomenon of Cl₂ evolution (Figure 3c). Our COMSOL simulation based on Bernoulli's principle shows the pressure difference in the range of 0 ~ 116.3 mPa with the carrier flow of 0~80 mL min⁻¹, which will drive as-generated Cl₂ migration to gas chamber. This prediction is well aligned with experimental results: by tuning the carrier flow rate from 0 to 80 mL min⁻¹, the Faradaic efficiencies of Cl₂ gas products increases from 17.6% to 82.6% (Figure 3c). Moreover, mechanism study has been conducted by *operando* Raman spectra for chlorine evolution reaction (CER) with/without interfacial pressure difference (Figure 5a and

supplementary Figure 57). The decreased *OCl peak signal unambiguously confirm the role of interfacial pressure difference, *i.e.*, Cl₂ produced at the electrode/electrolyte interfaces intermediately taken away by gas flow following Bernoulli's principle.

(3) It should be noted that the as-generated pressure difference is very stable during reaction process, which is revealed by excellent stability of chlorine evolution reaction (CER). Ti-MOF electrode exhibits strong stability in universal pH conditions (pH=1, 7, 13) with little morphology and structure decay after stability test after 50 hrs (Figure 2h, Supplementary Figures 37–39). The prototype device shows long-term stability for Cl₂ electrosynthesis with little fluctuation for 200 hrs at a current density of 100 mA cm⁻² (Figure 3g). Even at extremely high current density of 1.14 A cm⁻², our system can still demonstrate excellent stability for 40 hrs (Supplementary Figure 52).

Based on above results, the following Supplementary Video/sentences have been revised/added into updated manuscript:

Supplementary information, line 564-588:

It should be noted that Bernoulli principle played a key role in illustrating the interfacial pressure difference at three-phase boundary, which promote oriented Cl₂ migration to gas chamber during reaction process. Theoretically, Bernoulli principle a basic rule in fluid dynamics illustrating the relationship between the surface velocity and pressure. In the present work, two fluids have presented at the three-phase boundary, carrier gas flow and electrolyte flow, and according to Bernoulli principle that can be illustrated as:

$$P_1 + \frac{1}{2}\rho_1V_1^2 + \rho_1gh_1 = P_2 + \frac{1}{2}\rho_2V_2^2 + \rho_2gh_2$$

where 1 represents carrier gas flow and 2 represents electrolyte flow. So, the interfacial pressure difference (∇p) can be illustrated as:

$$\nabla p = \frac{1}{2}\rho_1V_1^2 + \rho_1gh_1 - \frac{1}{2}\rho_2V_2^2 - \rho_2gh_2$$

Therefore, when high-velocity carrier gas flow and low electrolyte flow have been applied to three-phase boundary, it will generate a large localized pressure gradient. This can promote Cl₂ bubbles migration away from electrolyte flow, and into carrier gas. To clearly clarify this mechanism, we have provided a schematic animation in Supplementary Video S2.

Quantitatively, the function of pressure difference has been described by COMSOL simulations

(Figure 5c-d, Supplementary Video S1) and experimental phenomenon of Cl₂ evolution (Figure 3c). Our COMSOL simulation based on Bernoulli's principle shows the pressure difference in the range of 0 ~ 116.3 mPa with the carrier flow of 0~80 mL min⁻¹, which will drive as-generated Cl₂ migration to gas chamber. This prediction is well aligned with experimental results: by tuning the carrier flow rate from 0 to 80 mL min⁻¹, the Faradaic efficiencies of Cl₂ gas products increases from 17.6% to 82.6% (Figure 3c). Moreover, mechanism study has been conducted by *operando* Raman spectra for chlorine evolution reaction (CER) with/without interfacial pressure difference (Figure 5a and Supplementary Figure 57). The decreased *OCl peak signal unambiguously confirm the role of interfacial pressure difference, *i.e.*, Cl₂ produced at the electrode/electrolyte interfaces intermediately taken away by gas flow following Bernoulli's principle.”

Response to Reviewer #1

Original Comment: *Most of my concerns have been well addressed in this revision. Still, there are two issues to be solved.*

Original Comment 1. *The influence of pH on the solubility of Cl₂ should be considered. An environment with low pH (acidic) favors the existence of Cl₂ in its molecular form (where physical dissolution dominates), while a high pH (alkaline) environment promotes chemical reactions of Cl₂, significantly increasing its apparent total solubility.*

Response: We are sorry for not explaining it clearly. We have further highlighted the description as follows:

i) Pourbaix diagram illustrates the theoretical limitation of pH on the solubility of Cl₂ in common membrane-free alkaline system (Figure 1c, d). In the pH range less than 4.3, the theoretically generated Cl₂ will escape, so gaseous Cl₂ can be produced; in the pH range of 4.3~7.5, the reaction of Cl₂+H₂O→HClO+Cl⁻+H⁺ will occur, generating liquid-phase HClO; In the pH range greater than 7.5, a further reaction of HClO+OH⁻→ClO⁻+H₂O will take place. The series of reactions that occur with the increase of pH makes Cl₂ precipitation reaction more difficult under high pH conditions.

Therefore, in a membrane-free chlor-alkali design, hydroxyl ions (OH⁻) from cathodic HER easily migrates to the anodic CER chamber that elevates pH (pH>7). This would lead to prohibitively low Cl₂ productivities owing to dissolved Cl₂ into electrolyte. Even for the benchmark DSA catalyst (Figure 1d), Cl₂ Faradaic efficiencies from CER are 46.6%, 4.3% and 0% at pH=1, 7 and 13, respectively (Manuscript, Page 4–5, Line 69–81).

ii) Addressed the stumbling problem by “Bernoulli’s principle”. As shown in Manuscript, Page 4, Line 59–66, we have simulated the process of local pressure generated at triple-phase boundary. By manipulating the flow rates of liquids (0~8.4 mL min⁻¹) and carrier gases (0~80 mL min⁻¹), substantial pressure difference can be achieved (0~116.3×10⁻³ Pa). This offers the opportunity to promote oriented migration of Cl₂ away from reaction interfaces (*i.e.*, H₂/OH⁻ at cathode/anode), which can not only prevent H₂/Cl₂ crossover but also circumvent the Pourbaix diagram problems, leading to significantly enhanced activities (Figure 5c-d and Supplementary Video S1-S2).

iii) Taking account of the influence of pH under all experimental conditions (Supplementary Figure

33). Specifically, for CER process, we have investigated the performances under universal pH condition (Figures 2g, h, Supplementary Figures 24, 28). For the prototype device (Figure 3c, Supplementary Tables 4–5), we have manipulated the speed of carrier gases (CO₂ and air), thereby investigated the influence of Bernoulli's principle. We attribute the declined activities at low gas speed to dissolved Cl₂ into electrolyte. For industrial-level electrolysis (Figures 4a, Supplementary Figures 51–52), we have clearly illustrated the gaseous Cl₂ and dissolved chlorine species.

Figure 1c, Pourbaix diagram of Cl₂ evolution reduction.

Based on above results, the following figure/sentence has been added/revised in updated manuscript:

Supplementary Figure 33. Performance comparison of Cl₂ separation and Cl₂ dissolution under different pH conditions.

“Manuscript, Page 7, Line 135–139. This result is in great contrast to theoretical Pourbaix diagram

(Figure 1c, d; Supplementary Figure 33), where an environment with low pH (acidic) favors the existence of Cl₂ in its molecular form (where physical dissolution dominates), while a high pH (alkaline) environment promotes chemical reactions of Cl₂ in solution, leading to dramatically activity decay.”

Original Comment 2. *Why is the difference between Ti-MOF-membrane and Ti-MOF-membrane-free in Figure 3d so negligible? The performance disparity between the membraned and membrane-free configurations appears marginal and requires further clarification. In Fig. 3f, the colors corresponding to the four catalysts are indistinguishable. Please recreate this figure with clearer and more distinct visual differentiation.*

Response: Thanks for your constructive comment. We have given the response as follows:

i) The marginal performance disparity between Ti-MOF-membrane and Ti-MOF-membrane-free highlights the advantage of our system design. Specifically, traditional chlor-alkali systems have involved membranes (like asbestos fleeces or fluorine-containing ion exchange membranes), which is used for separating Cl₂ production at the anode, hydrogen (H₂) production at the cathode and hydroxyl ions (OH⁻) in the electrolyte. However, a major problem associated with these membrane systems includes high operation costs arising from limited lifetime and low tolerance to contaminant ions of membranes. Further, these membranes also increase system maintenance cost because of complicated configuration, as gas pressures in anodic and cathodic chambers must remain in balance, otherwise aggravating membrane degradation.

Consequently, it is highly desirable to develop a membrane-free chlor-alkali design that would allow for simplified system configuration and durable operation, thus potentially delivering production cost savings. However, realizing such a design has met substantial challenges because of anodic/cathodic product crossover (H₂/Cl₂ mixture), leading to significant safety issue. Further, according to Pourbaix diagram, such a design would lead to prohibitively low Cl₂ productivities owing to parasitic oxygen evolution reaction (OER) and dissolved Cl₂ into electrolyte.

In the present study, we have proposed a new class of membrane-free design for chlor-alkali systems. We have addressed the stumbling problems of anodic/cathodic product crossover and

Pourbaix diagram by “Bernoulli’s principle”. Thereby, even removing the separation membrane, the chlorine evolution performance of membrane-free system can show similar behavior to membrane-based counterpart.

ii) We have adjusted the colors and color layers of Figure 3f. Now the colors corresponding to the four situations are presented clearly.

Figure 3f, The Cl₂ Faradaic efficiencies (FE), yield rates and other activity comparisons for Ti-MOF and DSA in membrane-based/-free devices.

Response to Reviewer #4

Original Comment: *The authors have addressed most of my concerns, but there are still some minor issues should be solved before publication.*

Original Comment 1. *The authors attributed the lower coordination number of Ti-O (3.36) in Ti-MOF compared to that of the theoretical TiO₆ units to the difference in contributions from O at various positions (i.e., internal O contributes 1, while surface O contributes 0.5, Figure R2). They should also consider another possibility that the lower coordination number may be related to the presence of abundant oxygen vacancies in the Ti-O framework. Considering the fact of the lower coordination number, the authors are suggested to revise the theoretical model to better reflect the actual situation.*

Response: We are sorry for not explaining it clearly. We have already taken account of oxygen vacancies in building the theoretical model of Ti-MOF. We have further highlighted the description as follows:

i) Firstly, we have built a theoretical model of Ti-MOF-before based on a number of characterizations. Specifically, in X-ray Photoelectron Spectroscopy (XPS, Supplementary Figures 11–12) analyses, Ti-O, C=O, and C-O bonds were detected in O 1s profile, indicating the bonding interaction between Ti and O. In C 1s spectrum, those peaks corresponding to C=C, C-C, C-O bonds, and the π - π^* transition were observed, which confirms the presence of organic ligands. Moreover, Fourier Transform Infrared Spectroscopy (FTIR, Supplementary Figure 6b) have verified the successful synthesis of Ti-MOF, as the characteristic infrared peaks of Ti-O, C=C, and C-O bonds were identified. X-ray Diffraction (XRD, Supplementary Figure 6a) analyses have revealed characteristic peaks in the 5-10° range, confirming the well-defined crystalline structure of Ti-MOF. X-ray Absorption Spectroscopy (XAS) characterizations have showed that the Ti-O coordination number of the uncalcined Ti-MOF model is 3.98 (Supplementary Table 2). Based on above characterizations, we have built the theoretical model of Ti-MOF-before with Ti-O coordination number of 4 as Figure R1, where Ti as the central atom, surrounded by four O atoms derived from four 2,6-naphthoic acid disodium (2,6 naphthoic 2Na) ligands.

ii) Secondly, we have built the theoretical model of Ti-MOF by modifying Ti-MOF-before (Supplementary Figure 64). Specifically, Ti-MOF has been experimentally prepared by the calcination of Ti-MOF-before at 450 °C for 3 hrs to improve its crystallinity. Such a calcination treatment has not altered Ti-MOF architecture, which is still in the form of octahedra, as confirmed by TEM (Figure 2a), XRD (Figure 2b), FT-IR (Figure 2c). While X-ray absorption near-side structure (XANES) confirms the coordination number of Ti-O has decreased to 3.36 (Supplementary Table 2). Based on these data, we have made modifications of Ti-MOF-before theoretical model, where the coordination number of Ti-O in updated model is 3. This has been achieved by deleting one ligand from Ti-MOF-before followed by structural optimization.

Based on the Ti-MOF theoretical model, we have conducted DFT calculations to explain the chlorine evolution reaction (CER) mechanism on Ti-MOF electrode. Based on our calculation results from Ti-MOF model (Figures 5e-i), the detailed CER mechanism has been successfully unravelled, *i.e.*, CER pathway by calculating Gibbs free energy change of reaction intermediates, as well as Pourbaix diagrams showing the surface adsorption changes with applied potentials at different pHs. Further, we have unraveled the high CER selectivity by calculating Gibbs free energy change of side OER process. Therefore, our Ti-MOF model is reliable for the study.

Figure R1. The periodic structure of Ti-MOF-before.

Based on above results, the following sections have been revised in updated manuscript (Supplementary information, Page, Line 722–723):

Supplementary Figure 64. a, Three-dimensional perspective view of Ti-MOF. b-d, Observe the modeling structure from three different perspectives.

Supplementary Table 2. Bond length and coordination of Ti-MOF-before and after calcination

Material	shell	CN	$R(\text{\AA})$	ΔE_0 (eV)	σ^2 (10^{-3}\AA^2)	R factor
Ti-foil	Ti-Ti	6.24 ± 0.21	2.89 ± 0.002	-7.05 ± 0.18	0.006 ± 0.0005	0.0084 ± 0.00224
TiO ₂	Ti-C	4.15 ± 0.05	1.95 ± 0.003	-1.55 ± 0.26	0.006 ± 0.0001	0.014 ± 0.0012
	Ti-Ti	2.05 ± 0.05	3.06 ± 0.0004		0.0040 ± 0.00025	
TiC	Ti-O	6.03 ± 0.08	2.14 ± 0.00039	-3.47 ± 0.01	0.0002 ± 0.0001	0.0095 ± 0.00035
	Ti-Ti	11.93 ± 0.14	3.05 ± 0.0002		0.0009 ± 0.0001	
Ti-MOF (before)	Ti-O	3.98 ± 0.04	1.90 ± 0.0005	-7.04 ± 0.10	0.016 ± 0.0001	$0.0034 \pm 4.1 \times 10^{-5}$
	Ti-Ti	4.05 ± 0.002	3.10 ± 10^{-7}		$0.019 \pm 1.5 \times 10^{-5}$	
Ti-MOF	Ti-O	3.36 ± 0.11	1.92 ± 0.002	-4.21 ± 0.21	0.011 ± 0.0009	0.013 ± 0.0043
	Ti-Ti	4.05 ± 0.09	3.12 ± 0.00008		0.018 ± 0.00064	

Original Comment 2. *Were the performance tests of DSA and Ti-MOF in Figure 3d and e conducted in the same prototype device shown in Figure 3b? Was there a pressure difference in both cases? Why did DSA exhibit such poor chlorine production performance (with a Faraday efficiency for chlorine gas below 10%)?*

Response: Thank you for comment. We have given the response as follows:

- i) Yes, the performance tests of DSA and Ti-MOF were conducted in the same apparatus (Figure 3b).
- ii) Yes, there is a pressure difference for both DSA and Ti-MOF.
- iii) The reason for the poor Cl₂ generation performance of DSA is its bulk plate structure. Generally, DSA is fabricated by coating bulk Ti substrate with electrocatalytic metal oxide layers (e.g., RuO₂, IrO₂). *Chem Rev* 2016, 116, 2982 Despite pressure difference, the bulk Ti substrate does not allow Cl₂ migration from three-phase boundary to gas chamber. Therefore, during the electrolysis process, DSA has still been limited by the theoretical constraints of Pourbaix diagram; and most of the generated Cl₂ dissolves in the electrolyte, leading to poor chlorine production activities.

To provide more evidence for above hypothesis, we have replaced Ti substrate of DSA with gas diffusion electrode (GDE), and synthesized the electrode of metal oxide (e.g., RuO₂, IrO₂)/GDE. Particularly, we have tested the performance of RuO₂/TiO₂@GDE electrode under the same condition (Supplementary Figure 31). Without pressure difference, the RuO₂/TiO₂@GDE electrode has demonstrated diminished activities with elevated pH values, *i.e.*, Faradaic efficiency of 70.9% (yield rate of 3 mmol cm⁻² h⁻¹) at pH =1, 7.3% (yield rate of 0.3 mmol cm⁻² h⁻¹) at pH =7 and 0% (yield rate of 0 mmol cm⁻² h⁻¹) at pH =13. In great contrast, with pressure difference, the activities of RuO₂/TiO₂@GDE increased significantly, *i.e.*, Faradaic efficiency of 86.3% (yield rate of 3.42 mmol cm⁻² h⁻¹) at pH =1, 88% (yield rate of 3.3 mmol cm⁻² h⁻¹) at pH =7 and 81.4% (yield rate of 3.3 mmol cm⁻² h⁻¹) at pH =13. Therefore, after introducing a porous GDE substrate, the performance of chlorine gas generation is significantly improved.

Based on above discussion, the following sentences have been added into updated Supplementary information:

Supplementary Figure 31. Comparison experiments for different samples. a-d, The Cl₂ Faradic efficiencies and yield rates without pressure difference for GDE loaded (RuO₂/TiO₂) as a surrogate of DSA. c-d, Cl₂ Faradic efficiency and yield rate of Ti-MOF materials without pressure difference.

“Supplementary information, Page 45–46, Line 432–450, We have tried to tune the pressure difference for DSA-based system, however, cannot achieve this operation. This is because of DSA fabricated by coating bulk Ti substrate with electrocatalytic metal oxide layers (e.g., RuO₂, IrO₂)²³ The bulk Ti substrate does not allow gas penetration at three-phase boundary during electrochemical reaction. Consequently, we have replaced Ti substrate of DSA with gas diffusion electrode (GDE), and synthesized the electrode of metal oxide RuO₂/TiO₂@GDE. Particularly, we have tested the performance of RuO₂/TiO₂@GDE electrode under the same condition. Without pressure difference,

the RuO₂/TiO₂@GDE electrode has demonstrated diminished activities with elevated pH values, *i.e.*, Faradaic efficiency of 70.9% (yield rate of 3 mmol cm⁻² h⁻¹) at pH=1, 7.3% (yield rate of 0.3 mmol cm⁻² h⁻¹) at pH=7 and 0% (yield rate of 0 mmol cm⁻² h⁻¹) at pH=13. In great contrast, with pressure difference, the activities of RuO₂/TiO₂@GDE increased significantly, *i.e.*, Faradaic efficiency of 86.3% (yield rate of 3.42 mmol cm⁻² h⁻¹) at pH=1, 88% (yield rate of 3.3 mmol cm⁻² h⁻¹) at pH=7 and 81.4% (yield rate of 3.3 mmol cm⁻² h⁻¹) at pH=13.

The same phenomenon has also been observed for Ti-MOF. Without pressure difference, Ti-MOF has demonstrated diminished activities with elevated pH values, *i.e.*, Faradaic efficiency of 72.8% (yield rate of 2.28 mmol cm⁻² h⁻¹) at pH=1, 5.8% (yield rate of 0.21 mmol cm⁻² h⁻¹) at pH=7 and 0% (yield rate of 0 mmol cm⁻² h⁻¹) at pH=13. In great contrast, with pressure difference, the activities of Ti-MOF increased significantly, *i.e.*, Faradaic efficiency of 93.4% (yield rate of 1.68 mmol cm⁻² h⁻¹) at pH=1, 94% (yield rate of 0.93 mmol cm⁻² h⁻¹) at pH=7 and 88.8% (yield rate of 0.78 mmol cm⁻² h⁻¹) at pH=13.”

Original Comment 3. *The authors should assign the vibrational peaks of H₂NDC at 1668 and 1415 cm⁻¹ and of 2,6 naphthoic 2Na at 786 cm⁻¹.*

Response: Thanks for your kind reminding. We have labelled the vibrational peaks as follows:

Supplementary Figure 4. Fourier Transform infrared spectroscopy (FTIR) of 2,6 naphthoic acid and 2,6 naphthoic 2Na.

Original Comment 4. *The author needs to provide the formulas used for calculating the economic costs, rather than only presenting the final calculation results.*

Response: Thanks for your kind reminding. The formulas used for calculating the economic costs are given as follows (Supplementary information, Page 12–13, Line 233–245):

i) The cost price of product (NaOH+Cl₂) (without membrane)

The cost price of product

$$\begin{aligned}
 &= \text{Raw material cost} + \text{Separation cost} + \text{Equipment cost} + \text{Electrolysis cost} \\
 &= \left[\left(\frac{\$597050}{\text{year}} + \frac{\$3283200}{\text{year}} + \frac{\$6465608.8}{\text{year}} + \frac{\$6262.9}{\text{year}} + \frac{\$13636}{\text{year}} + \frac{\$5024160}{\text{year}} \right) \right. \\
 &\quad \times 20 \text{ year} + \$16532977.05 \left. \right] / \left(\frac{5223.61 + 52584.99 \text{ kg}}{h} / 2 \times \frac{8000h}{\text{year}} \right) \\
 &\quad \times 20 \text{ year} = \$0.36 \text{ kg}^{-1}
 \end{aligned}$$

ii) The cost price of product (NaOH+Cl₂) (with membrane):

The cost price of product

$$\begin{aligned}
 &= \text{Raw material cost} + \text{Separation cost} + \text{Equipment cost} + \text{Electrolysis cost} \\
 &\quad + \text{membrane cost} \\
 &= \left[\left(\frac{\$597050}{\text{year}} + \frac{\$3283200}{\text{year}} + \frac{\$6465608.8}{\text{year}} + \frac{\$6262.9}{\text{year}} + \frac{\$13636}{\text{year}} + \frac{\$5024160}{\text{year}} \right. \right. \\
 &\quad \left. \left. + \frac{\$1172492.8}{\text{year}} \right) \times 20 \text{ year} \right] / \left(\frac{5223.61 + 52584.99 \text{ kg}}{h} / 2 \times \frac{8000h}{\text{year}} \right) \\
 &\quad \times 20 \text{ year} = \$0.39 \text{ kg}^{-1}
 \end{aligned}$$

Original Comment 5. *Why the current densities used in the three tables, namely Supplementary Table 9-11: Ti-MOF-membrane-free for CER, Ti-MOF-membrane for CER, and DSA-membrane for CER, were different? Would the difference in current density affect the calculation of economic analysis?*

Response: Thanks for your kind reminding. We have given the response as follows:

i) We have carefully rechecked the data for Supplementary Table 9–11. We are sorry for making a typo/mistake, the value of 1 A cm⁻² in Supplementary Table 10 should be 1.14 A cm⁻². Therefore, the current densities are the same for Ti-MOF-membrane-free and Ti-MOF-membrane.

ii) The current densities have an impact on the results of the technical and economic analysis. For example, for Ti-MOF-membrane-free, the costs and benefits corresponding to different currents ranging from 0.18 to 1.14 A cm⁻² are all different (Supplementary Figures 54–56). Similarly, the technical and economic analysis calculated for other counterparts have also been affected.

Therefore, we tested the potential and chlorine activity of DSA at an electric current density of 1.14 A cm⁻², and recalculated the economic analysis of DSA at this current density. Based on above discussions, we have made the following changes in updated manuscript:

Supplementary Table 10. FNPV (\$) analysis of Ti-MOF-membrane for CER at different current densities.

J (A cm ⁻²) \ Time (year)	1.14	0.7	0.34	0.18
0	-11941000	-11941000	-11941000	-11941000
1	-6664621.225	-9234844.03	-11144391.2	-11784160.54
2	-2062661.848	-7080715.896	-10808879.42	-12057952.88
3	2303149.805	-5046172.095	-10506351.2	-12335714.3
4	6444058.861	-3125518.134	-10235235.88	-12617255.79
5	10370774.97	-1313330.689	-9994037.622	-12902397.34
6	14093497.79	395555.3135	-9781331.796	-13190967.52
7	17621941.3	2006058.989	-9595761.621	-13482803.08
8	20965356.88	3522865.211	-9436034.924	-13777748.5
9	24132555.4	4950435.762	-9300921.063	-14075655.7

10	27131928.13	6293019.961	-9189247.998	-14376383.65
11	29971466.79	7554664.776	-9099899.501	-14679798.03
12	32658782.52	8739224.464	-9031812.497	-14985770.9
13	35201124.04	9850369.745	-8983974.534	-15294180.44
14	37605394.86	10891596.54	-8955421.372	-15604910.62
15	39878169.8	11866234.31	-8945234.686	-15917850.92
16	42025710.56	12777453.95	-8952539.884	-16232896.1
17	44053980.67	13628275.38	-8976504.018	-16549945.94
18	45968659.68	14421574.7	-9016333.806	-16868904.97
19	47775156.7	15160091.06	-9071273.74	-17189682.27
20	49478623.25	15846433.17	-9140604.289	-17512191.27

Supplementary Table 11. CER simulation electrolysis and FNPV (\$) analysis of DSA-membrane for CER at 1.14 A cm⁻².

Current density (A cm ⁻²)	1.14	J (A cm ⁻²)	
		Time (year)	1.14
Potential (V)	1.42	0	-22908800
Materials	DSA	1	-15127547.82
FE (%)	0.9	2	-8513049.138
Electrolytic area (m ²)	500	3	-2230533.39
Current (A)	5700000	4	3735808.139
Cl ₂ production (kg h ⁻¹)	6589.6772	5	9401031.364
NaOH production (kg h ⁻¹)	7655.7665	6	14779475.25
H ₂ O consumption (kg h ⁻¹)	3639.839025	7	19884795.96
NaCl consumption (kg h ⁻¹)	11807.81997	8	24729999.36
Electrolytic power (kW)	8094	9	29327471.98
Separation power (kW)	29075	10	33689010.53

/	/	11	37825849.97
/	/	12	41748690.25
/	/	13	45467721.81
/	/	14	48992649.83
/	/	15	52332717.32
/	/	16	55496727.19
/	/	17	58493063.11
/	/	18	61329709.57
/	/	19	64014270.82
/	/	20	66553989.02

Supplementary Figure 57. Techno-economic analyses. a, FNPV analyses for Ti-MOF-based membrane-free system and DSA-based membrane system. **b,** The production cost of electrochemical production per kilogram of Cl₂ and NaOH for Ti-MOF-based membrane-free system and DSA-based membrane system. **c-d,** cost distribution percentages and production cost distribution in an operating cycle for DSA-based membrane system.

Response to Reviewer #5

Original Comment: *The data presentation in the manuscript has several shortcomings that need to be addressed before the results can be considered reliable:*

Original Comment 1. *The description of Cl₂ measurement is not clear or sufficiently detailed. Critical information such as reagents details including companies, purity, and prepared concentration, full titration procedure, and standardization of the Na₂S₂O₃ solution is missing. Without these details, the reproducibility and reliability of the results cannot be assessed.*

Response: Thanks for your kind reminding. We have given the experimental details as follows (Supplementary information, Page 1–2 Line 2–14, 23–46):

1.1 Reagents

Sodium thiosulfate (Na₂S₂O₃, 99%, Shanghai Yi'en Chemical Technology Co., Ltd), potassium iodide (KI, 99%, Meryer Technologies Co., Ltd.), potassium dichromate (K₂Cr₂O₇, AR, Sinopharm Chemical Reagent Co., Ltd), starch ((C₆H₁₀O₅)_n, RG, Shanghai Adamas Reagent Co., Ltd), Ammonium ferrous sulfate (FeH₈N₂O₈S₂, 99%, Shanghai Yi'en Chemical Technology Co., Ltd), N,N-Diethyltoluidine (C₁₀H₁₆N₂, 99%, Shanghai Adamas Reagent Co., Ltd), 2,6 naphthoic acid (C₁₂H₈O₄, 98%, Meryer Technologies Co., Ltd.), N,N-Dimethylformamide (C₃H₇NO, 99.5%, Shanghai Aladdin Biochemical Technology Co., Ltd) Tetraisopropyl Titanate (C₃H₇NO, 99.9%, Shanghai Adamas Reagent Co., Ltd), Sodium hydroxide (NaOH, 99.9%, Shanghai Aladdin Biochemical Technology Co., Ltd), Hydrochloric acid (HCl, AR, Sinopharm Chemical Reagent Co., Ltd), Sulfuric acid (H₂SO₄, AR, Sinopharm Chemical Reagent Co., Ltd), Sodium chloride (NaCl, AR, Sinopharm Chemical Reagent Co., Ltd). All aqueous solutions were prepared with high-purity de-ionized water (DI-water, resistance 18 MΩ. Proton exchange membrane is using Nafion 117.

1.3 Calibration of the product by KI-Na₂S₂O₃

i) The prepared concentrations of calibration reagents:

Na₂S₂O₃ solution (0.1 M), KI solution (25g L⁻¹), Starch paste (1%)

ii) Full KI titration procedure:

Cl₂ is determined by the traditional KI titration method, which can change colorless KI to brown I₂. Next, the above mixture was mixed with starch solution (1 mL, 1 wt%), and titrated with 0.1 M Na₂S₂O₃ solution. The end point of titration is determined by the observation of blue-black color disappearing after the last drop of 0.1 M Na₂S₂O₃ into the absorption solution with no longer color

change within half a minute.

iii) The standardization of Na₂S₂O₃ solution:

The accurate concentration of Na₂S₂O₃ was calibrated using the potassium dichromate method: step 1, weigh a certain mass (accurate weighing) of dried K₂Cr₂O₇, and dissolve it in a specific volume of deionized water; step 2, add a KI solution acidified with dilute sulfuric acid, shake well, then place the mixture in a sealed dark environment and let it stand for 3–5 minutes; step 3, titrate with the Na₂S₂O₃ solution to be calibrated until the solution turns pale yellow, then add 2 mL of starch indicator (the solution will turn dark blue at this point). Continue titrating until the blue color of the solution fades and does not revert within 30 seconds—this is the endpoint. Record the volume of Na₂S₂O₃ solution consumed. Repeat the operation 3 times in parallel and take the average concentration; step 4, the accurate concentration of the Na₂S₂O₃ solution can be calculated from the titration results:

$$c(\text{Na}_2\text{S}_2\text{O}_3) = \frac{6 \times m(\text{K}_2\text{Cr}_2\text{O}_7) \times 1000}{M(\text{K}_2\text{Cr}_2\text{O}_7) \times V(\text{Na}_2\text{S}_2\text{O}_3)}$$

Among them, 6 represents the reaction molar ratio, $m(\text{K}_2\text{Cr}_2\text{O}_7)$ is the mass of K₂Cr₂O₇ weighed, $M(\text{K}_2\text{Cr}_2\text{O}_7)=294.18 \text{ g mol}^{-1}$, and $V(\text{Na}_2\text{S}_2\text{O}_3)$ is the volume of the Na₂S₂O₃ solution used for the titration.

Original Comment 2. *The KI–thiosulfate titration method is known to introduce significant errors, especially due to interferences and endpoint subjectivity. The manuscript does not provide any indication of experimental errors, replicate measurements, or data variability. The use of the FAS-DPD titration method would have been more suitable for this study, as it is widely regarded as more accurate and specific for chlorine analysis.*

Response: Thanks for your constructive comment. Accordingly, we have given the response as follows (Supplementary information, Page 2–3 Line 47–56; Supplementary Figure 82a):

1.4 Calibration of products by FAS-DPD titration method

Cl₂ can change DPD indicator into pink color. Next, FAS (0.1 M) was added drop by drop into above solution until the pink color completely disappears, which marks the endpoint:

P
A
G
E

Where the formula for calculating the production of Cl_2 can be derived:

$$c_{Cl_2} = \frac{c_{FAS} \times V_{FAS} \times 1000}{2 \times V_{sample}}$$

c_{FAS} represents the concentration of the FAS, mol L⁻¹; V_{FAS} is the volume of the FAS taken, L; V_{sample} is the volume of the absorbent solution, L

Supplementary Figure 82. a, The Faraday efficiency of Cl_2 at 1.8 V and the errors observed in repeated tests using the KI titration method and the FAS-DPD titration method were represented as error bars.

“Supplementary information, Page 101–102 Line 818–828,

To determine the experimental errors/data variability, we have conducted ten repetitive measurements for KI-thiosulfate titration method. Specifically, we have tested the electrochemical performances at a potential of 1.8 V and pH = 13 by titrating 10 times using the KI-thiosulfate titration method. We have calculated and added error bars into titration results (Faradaic efficiency: 85.2±5.6%, Supplementary Figure 82a).

We have quantitatively titrated five times by using FAS-DPD titration method (Faradaic efficiency: 83.7 ± 5.3%, Supplementary Figure 82a). Notably, the results are comparable for KI-thiosulfate titration method and FAS-DPD titration method, thereby confirming the reliability of our work (Supplementary Figure 82a).”

Original Comment 3. *There are several well-established, high-accuracy techniques for chlorine*

quantification (e.g., amperometric titration, ion chromatography, or membrane-covered amperometric analyzers). The manuscript should validate the reported chlorine yields with at least one of these methods to strengthen confidence in the results.

Response: Thanks for your kind reminding. We have conducted additional amperometric titration method as follows:

Firstly, KI is used to absorb the separated Cl_2 , and then all the absorption solution is taken out and placed in a single-cell electrolytic cell. Two clean Pt electrodes are used as redox electrodes, with the working electrode and counter electrode clamped to the two Pt electrodes respectively.

Next, a potential of $E=0.3\text{ V}$ is applied (Supplementary Figure 82b). In the first stage, namely the stable platform, no $\text{Na}_2\text{S}_2\text{O}_3$ is added, and it can be observed that the current stabilizes at approximately 1.4 mA. With continuous dropwise addition of $\text{Na}_2\text{S}_2\text{O}_3$ (Rapid titration stage), the current drops sharply. When it drops to 0.5 mA, the titration speed is slowed down (entering the Slow titration stage), and at this point, the current decreases slowly. Titration is stopped when the current drops to approximately 0.04 mA. After waiting for half a minute, the current shows almost no change, and the yellow color of I_2 in the electrolytic cell fades completely without recurrence. The consumption of $\text{Na}_2\text{S}_2\text{O}_3$ at the endpoint is recorded and plotted as a typical amperometric I_2 - $\text{Na}_2\text{S}_2\text{O}_3$ titration curve (Supplementary Figure 82c).

We have conducted the amperometric titration for five times. The Faraday efficiency of Cl_2 is calculated to be $83\pm 2.5\%$, which is close to the result of direct KI-thiosulfate titration method (Faradaic efficiency: $85.2 \pm 5.6\%$).

Based on above results, the following figure/sentence has been added in updated Supplementary information:

Supplementary Figure 82b, The measured current-time relationship of the ampere titration method. **c**, The typical I_2 - $Na_2S_2O_3$ titration curve and the calculated Faraday efficiency of the Cl_2 method.

“Supplementary information, Page 3, Line 58–69:

Firstly, KI is used to absorb the separated Cl_2 , and then all the absorption solution is taken out and placed in a single-cell electrolytic cell. Two clean Pt electrodes are used as redox electrodes, with the working electrode and counter electrode clamped to the two Pt electrodes respectively.

Next, a potential of $E=0.3$ V is applied (Supplementary Figure 82b). In the first stage, namely the stable platform, no $Na_2S_2O_3$ is added, and it can be observed that the current stabilizes at approximately 1.4 mA. With continuous dropwise addition of $Na_2S_2O_3$ (Rapid titration stage), the current drops sharply. When it drops to 0.5 mA, the titration speed is slowed down (entering the Slow titration stage), and at this point, the current decreases slowly. Titration is stopped when the current drops to approximately 0.04 mA. After waiting for half a minute, the current shows almost no change, and the yellow color of I_2 in the electrolytic cell fades completely without recurrence. The consumption of $Na_2S_2O_3$ at the endpoint is recorded and plotted as a typical amperometric I_2 - $Na_2S_2O_3$ titration curve (Supplementary Figure 82c).”

“Supplementary information, Page 102 Line 829–832:

Additionally, we conducted five ampere titration experiments using the ampere titration method described in 1.5. The Faraday efficiency of Cl_2 is calculated to be $83 \pm 2.5\%$ (Supplementary Figure 82b-c), which is close to the result of direct KI-thiosulfate titration method (Faradaic efficiency: $85.2 \pm 5.6\%$).”

To ensure the reliable of our results, FAS-DPD titration and amperometric titration method were also employed. The specific methods are described in Supplementary information. The as-obtained test results are shown in Supplementary Figure 82, which is comparable to KI titration method.”

Original Comment 4. *The description of GC analysis for H₂, O₂, and N₂ is incomplete. Essential details such as the instrument type, detector used, column specifications, calibration procedures, and standards are not reported. Furthermore, the manuscript does not include calculations or data for the molar amounts or yields of each gas, which are crucial for evaluating the performance of the system.*

Response: Thanks for your constructive comment. We have given the details as follows:

i) Instrument type, detector used, column specifications: The PANNA A91plus gas chromatograph was used to detect H₂, O₂, and N₂. This gas chromatograph was equipped with a chromatographic column produced by ABELBONDED, which has an operating temperature range of –60 to 200 °C. The column model is AB-PLOTA12O3, and the specifications of this capillary column are 30 m×0.53 mm×15 μm. The detectors include a Flame Ionization Detector (FID) and a Thermal Conductivity Detector (TCD). When detecting H₂, O₂, and N₂, we only need to focus on the detection results from the TCD.

ii) Calibration procedure: mixed standard gases were used to calibrate the relationship between concentration and gas phase signal. Particularly for H₂ calibration, we have used standard gases with H₂ contents of 1.98, 0.0044, 5.9167, 3.843, and 0.02 mol mol⁻¹. The peak areas measured by the instrument were 1078.87, 2.38, 3223.48, 2105.32, and 12.23 μV*s, respectively. From these data, the relationship between H₂ concentration and peak area in the gas phase was calculated as: $c=0.00183*s-0.0024$ (Supplementary Figure 46a).

iii) Standard and the calculations of products: when conducting H₂ measurement and creating the standard curve, the key criterion is that the standard curve of H₂ must consist of five standard gas calibration curves, and the R² value of this curve should be greater than or equal to 0.9999. When measuring H₂ in the gas bag (since H₂ is lighter), it is necessary to thoroughly shake the gas bag and

maintain an intake time of more than 30 seconds. This will yield more accurate data.

The calculations or data for the molar amounts or yields of each gas is as follows:

$$FE_{H_2} = \frac{Q_g}{Q_{total}} \times 100\% = \frac{nzF}{Q_{total}} \times 100\% = \frac{(0.0018s - 0.0024) \times v_{carrier\ gas} \times T \times zF}{Q_{total} \times V_m}$$

Where s represents the measured area of H₂ in the gas phase (μV s), $v_{carrier\ gas}$ is the gas flow rate during the reaction process (mL min⁻¹), T is the reaction time (s), z is the number of transferred electrons, F is the Faraday constant, F=96485 C mol⁻¹, V_m is the molar volume of the gas, which is 24.5 L mol⁻¹ at normal temperature and pressure.

Based on above results, the following figure/sentence has been added in updated Supplementary information:

Supplementary Figure 46a. Gas-phase testing of hydrogen. a, Standard curve of hydrogen gas.

“Supplementary information, Page 3–5 Line 70–93:

1.6 Calibration of products by gas chromatography

The PANNA A91plus gas chromatograph was used to detect H₂, O₂, and N₂. This gas chromatograph was equipped with a chromatographic column produced by ABELBONDED, which has an operating temperature range of –60 to 200 °C. The column model is AB-PLOTA12O3, and the specifications of this capillary column are 30 m×0.53 mm×15 μm. The detectors include a Flame Ionization Detector (FID) and a Thermal Conductivity Detector (TCD). When detecting H₂, O₂, and N₂, we only need to focus on the detection results from the TCD.

For the calibration procedure, mixed standard gases were used to calibrate the relationship between concentration and gas phase signal. Particularly for H₂ calibration, we have used standard gases with H₂ contents of 1.98, 0.0044, 5.9167, 3.843, and 0.02 mol mol⁻¹. The peak areas measured by the instrument were 1078.87, 2.38, 3223.48, 2105.32, and 12.23 μV*s, respectively. From these data, the relationship between H₂ concentration and peak area in the gas phase was calculated as: $c=0.00183*s-0.0024$ (Supplementary Figure 46a).

For the standard and the calculations of products, when conducting H₂ measurement and creating the standard curve, the key criterion is that the standard curve of H₂ must consist of five standard gas calibration curves, and the R² value of this curve should be greater than or equal to 0.9999. When measuring H₂ in the gas bag (since H₂ is lighter), it is necessary to thoroughly shake the gas bag and maintain an intake time of more than 30 seconds. This will yield more accurate data.

The calculations or data for the molar amounts or yields of each gas is as follows:

$$FE_{H_2} = \frac{Q_g}{Q_{total}} \times 100\% = \frac{nzF}{Q_{total}} \times 100\% = \frac{(0.0018s - 0.0024) \times v_{carrier\ gas} \times T \times zF}{Q_{total} \times V_m}$$

Where s represents the measured area of H₂ in the gas phase (μV s), $v_{carrier\ gas}$ is the gas flow rate during the reaction process (mL min⁻¹), T is the reaction time (s), z is the number of transferred electrons, F is the Faraday constant, F=96485 C mol⁻¹, V_m is the molar volume of the gas, which is 24.5 L mol⁻¹ at normal temperature and pressure.”